# Precipitation legacy effects on soil microbiota facilitate adaptive drought responses in plants

**Nichole A. Ginnan** [1,2,3,7], **Valéria Custódio**[4,7], **David Gopaulchan**[4,7], **Natalie Ford**[1,3], **Isai Salas-González** [5], **Dylan H. Jones** [4], **Darren M. Wells** [4], **Ângela Moreno**[6], **Gabriel Castrillo** [4] ✉ & **Maggie R. Wagner** [1,3] ✉

Drought alters the soil microbiota by selecting for functional traits that preserve fitness in dry conditions. Legacy effects or ecological memory refers to how past stress exposure influences microbiota responses to future environmental challenges. How precipitation legacy effects impact soil microorganisms and plants is unclear, especially in the context of subsequent drought. Here we characterized the metagenomes of six prairie soils spanning a precipitation gradient in Kansas, United States. A microbial precipitation legacy, which persisted over a 5-month-long experimental drought, mitigated the negative physiological effects of acute drought for a native wild grass species, but not for the domesticated crop species maize. RNA sequencing of roots revealed that soil microbiota with a low precipitation legacy altered expression of plant genes that mediate transpiration and intrinsic water-use efficiency during drought. Our results show how historical exposure to water stress alters soil microbiota, with consequences for future drought responses of some plant species.

The increasing frequency and intensity of droughts associated with climate change threaten plant productivity in both natural and agricultural ecosystems. However, the ability of soil microbial communities to quickly adapt to environmental shifts[1] may bolster the resilience of plants and ecosystems[2]. In addition, the cumulative effects of past stress exposure can influence microbial communities' responses to future challenges, a phenomenon referred to as legacy effects or ecological memory[3]. Despite growing recognition of microbial legacy effects, little is known about the mechanisms driving them, their long-term persistence and whether they affect different plant species uniformly.

To investigate microbial legacy effects and isolate the drivers and impacts of microbial adaptations to water limitation, we (1) evaluate natural soil metagenome variation across a steep precipitation gradient, (2) test whether legacy effects can persist through experimental perturbation, and (3) evaluate the impacts of low-precipitation microbial legacy effects on plant responses to acute drought at the molecular and physiological levels. Finally, we assess the extent to which microbiota legacy effects are transferable across plant species. Our results show that precipitation legacy effects are more salient at the metatranscriptomic level than the taxonomic or metagenomic levels, and that they trigger transcriptional changes in roots that improve resistance to subsequent acute droughts, at least in some plant species.

## Results

### Nutrients and precipitation jointly shape soil microbiota

To identify microbial markers of precipitation legacy effects, we sequenced the metagenomes of soils from 6 never-irrigated remnant prairies spanning ~568 km of a steep precipitation gradient in Kansas

[1]Kansas Biological Survey and Center for Ecological Research, University of Kansas, Lawrence, KS, USA. [2]Department of Microbiology and Plant Pathology, University of California, Riverside, Riverside, CA, USA. [3]Department of Ecology and Evolutionary Biology, University of Kansas, Lawrence, KS, USA. [4]School of Biosciences, University of Nottingham, Sutton Bonington Campus, Nottingham, UK. [5]Centre for Genomics Sciences, Universidad Nacional Autónoma de México, Cuernavaca, Mexico. [6]Ministério da Agricultura e Ambiente – Empresa Pública Água de Rega, SA, Praia, Cabo Verde. [7]These authors contributed equally: Nichole A. Ginnan, Valéria Custódio, David Gopaulchan. ✉e-mail: gabriel.castrillo@nottingham.ac.uk; maggie.r.wagner@ku.edu

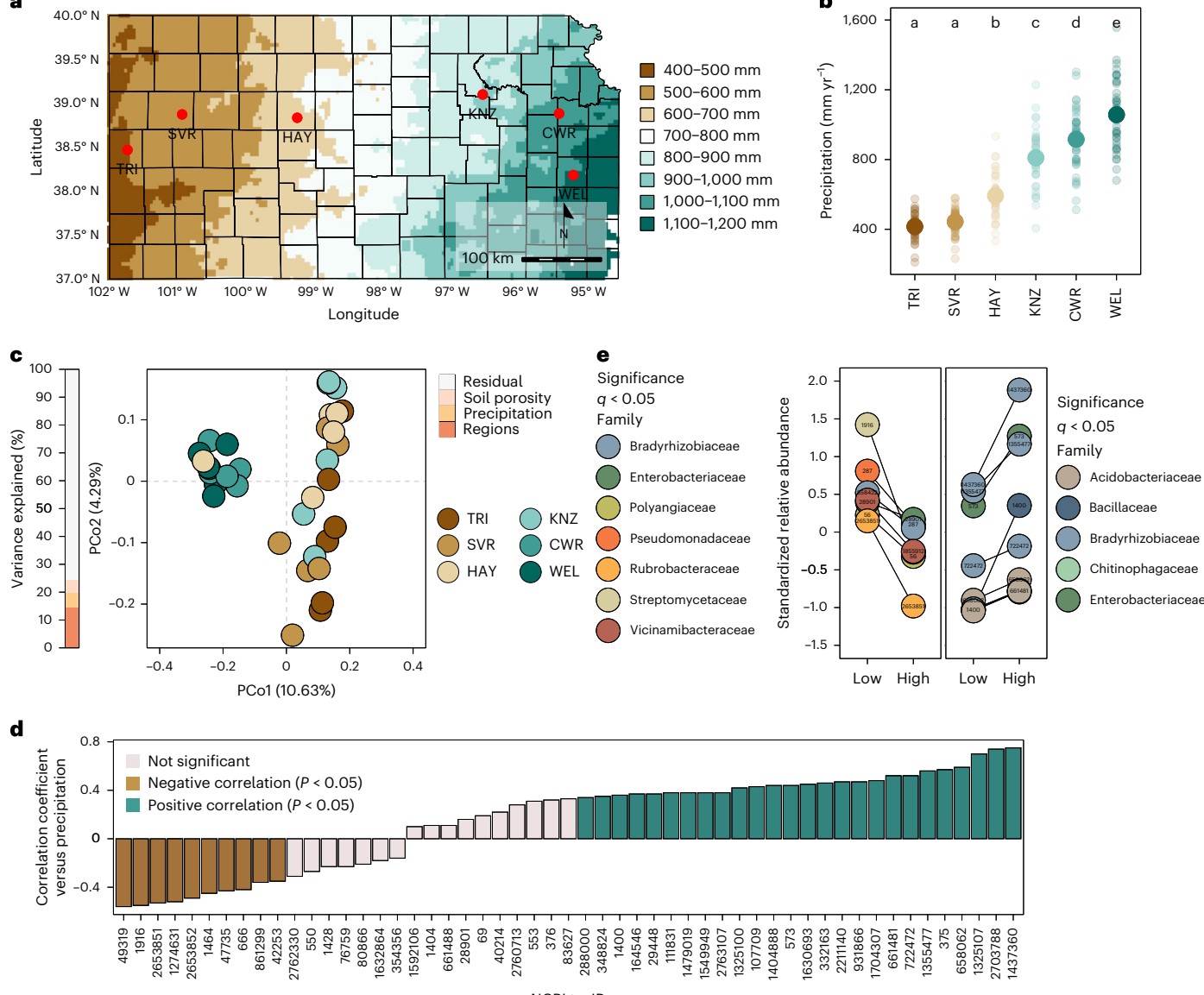

**Fig. 1 | Bacterial markers of precipitation across a regional gradient. a**, Map of Kansas, USA, showing the collection locations of the soils used in this work. **b**, Precipitation (mm yr$^{-1}$) at each collection site from 1981 to 2021. Large points represent the EMM; error bars represent ±95% confidence intervals. The statistical difference between soils was determined via ANOVA ($F_{5,240} = 140.3$, $P = 3.79 \times 10^{-69}$); the letters indicate pairwise group differences based on a two-tailed Tukey post hoc test with multiple comparison correction ($\alpha = 0.05$). **c**, Principal coordinate analysis of the soil microbiota across the precipitation gradient. The bar on the left denotes the percentage of the overall variance explained by the independent variables. **d**, Pearson correlation coefficients between each bacterial taxon (NCBI taxID) and mean annual precipitation.

The significance was assessed using a two-tailed Pearson correlation test. **e**, Relative abundances of bacterial taxa identified in soils exposed to low or high precipitation levels after statistically controlling for soil porosity and mineral nutrient content. The coloured points containing the relevant NCBI taxIDs represent the mean standardized relative abundance for each bacterial taxon. The line connecting both points is the difference between low- and high-precipitation soils. A black line indicates statistical significance ($q < 0.05$). CWR, Clinton Wildlife Reserve; HAY, Hays Prairie; KNZ, Konza Wildlife Reserve; PCo, principal coordinate; SVR, Smoky Valley Ranch; TRI, Kansas State University's Tribune Southwest Research Center; WEL, Welda Prairie.

in the United States (Fig. 1a,b and Supplementary Table 1). Although bacterial α-diversity was similar among soils (Extended Data Fig. 1a), community composition showed a strong biogeographic signature (permutational multivariate analysis of variance (PERMANOVA), proportion of variation explained ($R^2$) = 0.11, $P = 0.001$) and precipitation explained 5.3% of the variation. The first principal coordinate axis, which explained 10.6% of the total variation, separated the communities of the two highest-precipitation sites from the others (Fig. 1c). In line with previous findings[4], Actinomycetota and Bacillota were enriched in low-precipitation soils, whereas Pseudomonadota and Acidobacteriota were enriched in high-precipitation soils (Extended Data Fig. 1b).

To investigate covariation between precipitation and edaphic properties[5], we examined 24 trace element profiles using inductively coupled plasma mass spectrometry (ICP-MS). Nutrients are known drivers of taxonomic composition and functional capacity in soil microbiota, particularly bacterial communities[6,7]. Mineral nutrient content differed among the 6 soils, with precipitation explaining 28.6% of the variation. The first principal coordinate axis of the nutrient profiles, which explained 39.6% of the total variation, separated the 3 lower-precipitation sites from the rest (Extended Data Fig. 1c). Concentrations of K, Mg, Ca, Li and P were negatively correlated with mean annual precipitation, whereas Cd, Mn, Se, As, Zn, Co, Pb, Rb, Fe and Cr

were positively correlated (Extended Data Fig. 1d,e). The mineral nutrient dissimilarities among soils were correlated with the corresponding microbiota composition dissimilarities (Extended Data Fig. 1f), suggesting that precipitation influences mineral nutrient accumulation in these soils, and both precipitation and nutrients impact microbial communities. For example, precipitation drives mineral weathering and solute production in soils[8], although this process also depends on many other geochemical and biological factors[9].

Next, we used X-ray computed tomography to quantify the porosity of intact soil cores. Porosity is directly related to soil hydraulic properties; generally, lower porosity results in lower water retention and infiltration[10]. Therefore, it is a good indicator of how precipitation affects the actual soil water content. Consistently, porosity decreased with depth to ~3.5 cm before stabilizing and increased with precipitation (Extended Data Fig. 2a,b). Notably, precipitation might affect porosity in surface soil layers[11], potentially affecting soil niche properties and, consequently, microbial communities. However, the porosity dissimilarities were uncorrelated with microbiota composition dissimilarities in these soils (Extended Data Fig. 2c), suggesting either that porosity does not control microbial community composition in these soils or that its influence is masked by precipitation legacy.

Finally, to identify taxonomic biomarkers of precipitation legacy, we modelled the relative abundances of bacterial taxa in relation to precipitation while controlling for porosity and elemental composition. This analysis revealed distinct clusters of taxa whose relative abundances varied significantly along the precipitation gradient (Extended Data Fig. 2d–f). Across the 6 soils, 19 taxa (defined and tracked using National Center for Biotechonlogy Information (NCBI) taxonomy identifiers (taxIDs)) were positively correlated and 9 were negatively correlated with precipitation (Fig. 1d). In addition, 15 of the most abundant (>0.1%) and prevalent (>20%) taxa were differentially abundant between the 3 lower-precipitation soils and the 3 higher-precipitation soils (Fig. 1e). These results indicate that water availability shapes soil bacterial communities, possibly by selecting for functions necessary to adapt to dry conditions and/or subsequent re-wetting[12].

## Metagenomic analyses suggest mechanisms of legacy effects

We used assembled metagenomic contigs to explore the functional potential of the soil communities spanning the precipitation gradient. To focus on functions that are associated with water availability rather than site-specific variation, we collapsed the soils into two groups representing sites with low-precipitation versus high-precipitation histories. These groupings preserved the observed similarities in taxonomic composition, nutrient content and porosity (Fig. 1c and Extended Data Figs. 1c and 2a). Overall, 62 Gene Ontology (GO) categories and 3,396 Kyoto Encyclopedia of Genes and Genomes (KEGG) reactions were differentially abundant between groups (Extended Data Fig. 2g and Supplementary Table 2). Processes enriched in low-precipitation soils included nitrogen cycling, fatty acid biosynthesis, DNA repair and glucan metabolism, which have all been linked to drought or stress tolerance[13–15]. However, other stress-related processes were depleted in low-precipitation soils, including ion transport (involved in osmotic adjustment), lipid catabolism (relevant for membrane integrity), and metabolism of cellular aldehydes and ketones (involved in oxidative stress)[16,17] (Extended Data Fig. 2g). These differences in functional potential suggest that these microbiomes are functionally adapted to local precipitation levels, making them excellent candidates for exploring how microbial precipitation legacy affects host plants.

Next, we investigated precipitation-associated genetic variation within 33 focal bacterial species (the previously identified biomarkers and other abundant and prevalent taxa). We mapped shotgun metagenomic reads to reference genomes from the NCBI database and identified sequence variants. Analysis of genetic distances showed variations in strain-level microbiome structure across the precipitation gradient (Extended Data Fig. 3a,b).

A subsequent genotype–environment association analysis identified genetic variants associated with precipitation in *Streptomyces*, *Luteitalea*, *Rubrobacter*, *Lacibacter*, *Rhizobium* and three *Bradyrhizobium* lineages (Extended Data Fig. 3c and Supplementary Table 3). Most precipitation-associated variants were located within or near protein-coding regions. Notably, some of the corresponding genes have known adaptive functions such as the phenolic acid stress response (PadR family transcriptional regulator)[18,19], maintenance of cellular functions under iron starvation and oxidative stress (Fe–S cluster assembly protein SufD and SufB)[20] and fatty acid synthesis (acetyl-CoA carboxylase biotin carboxylase subunit)[21], which impacts membrane composition and stress tolerance[17] (Extended Data Fig. 3c and Supplementary Table 3). These results indicate that precipitation legacy effects manifest through genetic differentiation within bacterial species, not just variation in community composition. Further study of these variants could reveal mechanisms by which precipitation shapes soil microbiota via adaptive evolution.

## Molecular precipitation legacies are robust to perturbation

To assess the effects of short-term perturbations on soil microbiota legacy, we exposed the 6 focal soils to a 5-month-long conditioning experiment. Replicate pots of each soil were either left unplanted or planted with *Tripsacum dactyloides* (eastern gamagrass, which is native to Kansas) and either drought-challenged or well-watered in a factorial design (Extended Data Fig. 4a). Compared with well-watered plants, the drought-treated plants were shorter and had more root aerenchyma (Fig. 2a and Supplementary Fig. 1), confirming that the conditioning-phase drought treatment was severe enough to induce a stress response in the plants and presumably the microorganisms[22].

Next, we explored how precipitation legacy affected the bacterial communities' responses to the conditioning-phase drought treatment. Congruent with observations from the field-collected soils, water availability did not affect α-diversity, regardless of whether or not a plant was present (Extended Data Fig. 4b). Although phylum-level taxonomic profiles were similar across treatments, the conditioning-phase watering and host treatments explained 4.3% and 1.1%, respectively, of the variation in metagenome content (Extended Data Fig. 4c,d). In contrast, precipitation legacy explained 14.1% of the metagenome variation, confirming that the conditioning-phase treatments did not erase the soils' ecological memory (Fig. 2b). In addition, taxonomic differences between high- and low-precipitation soils still reflected the patterns observed in field-collected soils, regardless of conditioning-phase treatment (Supplementary Fig. 2a,b and Supplementary Table 4). This resilience was particularly evident in soils with low precipitation legacies (Fig. 2b and Supplementary Fig. 2b).

To assess whether precipitation legacy effects on strain-level genetic variation were also robust to experimental perturbation, we investigated genetic variants within the previously identified bacterial markers and other prevalent species. Principal coordinate analysis of pairwise genetic distances based on those variants revealed that samples clearly separated along the first axis based on precipitation legacy rather than conditioning-phase treatments (Fig. 2c). Furthermore, conditioning-phase treatments did not affect allele frequencies, whereas a genome–environment association analysis identified precipitation-associated genes in *Luteitalea* and two *Bradyrhizobium* lineages, recapitulating results from the field-collected soils (Supplementary Fig. 3a–c and Supplementary Table 5).

Next, we tested whether functional potential still differed between low- and high-precipitation soils after the conditioning phase. GO terms related to the nitrogen cycle metabolic process were enriched in dry-legacy soils, whereas GO terms related to ion transport and amino acid catabolic processes were depleted in the dry-legacy soils, regardless of conditioning treatment (Supplementary Fig. 4a and Supplementary Table 6). These enrichment patterns mirrored the original field soil observations: of the

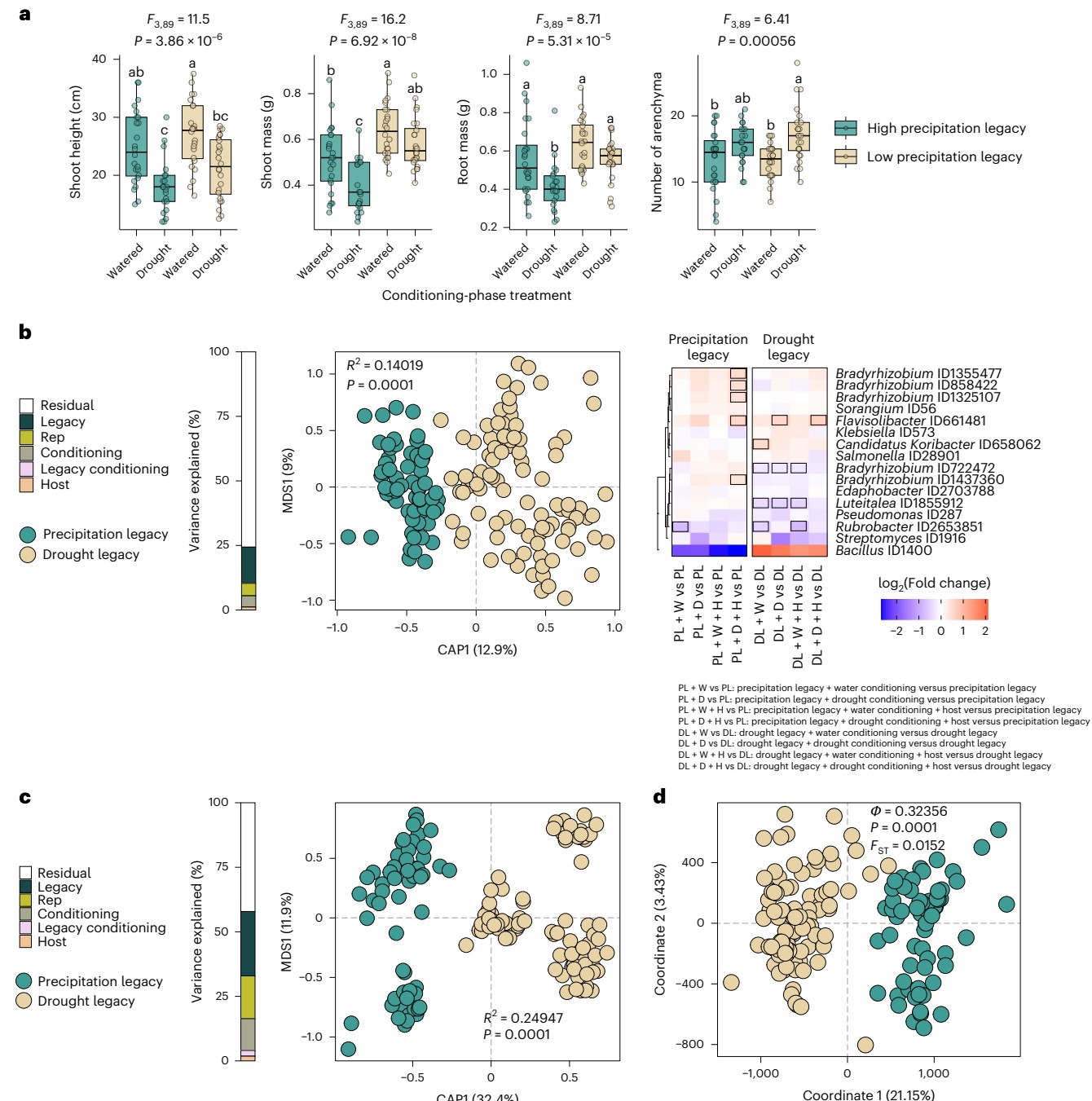

**Fig. 2 | Precipitation legacy effects on the soil microbiota are resilient to short-term perturbations. a**, Box plots showing phenotype distributions of *T. dactyloides* plants grown during the conditioning phase, in which they experienced 5 months of drought (+D) or well-watered (+W) conditions and were grown in soils with either low-precipitation legacies (drought legacy (DL); brown) or high-precipitation legacies (PL; blue) (for additional phenotypes, see Extended Data Fig. 4b,d). PL and DL indicate the baseline soils before the initiation of conditioning-phase treatments. Box edges represent the first and third quartiles, whiskers indicate the range of data points that fall within 1.5× the interquartile range of the first and third quartiles, and the centre lines indicate the medians. The letters indicate pairwise group differences based on ANOVA (N = 93 plants) followed by a two-tailed Tukey post hoc test with FDR correction (q < 0.05). **b**, Left: constrained ordination of soil metagenome taxonomic composition following the conditioning-phase treatments. Differences in β-diversity are shown across soil precipitation legacy groups. Group differences were assessed using PERMANOVA with 9,999 permutations, with $R^2$ and corresponding P values indicated on the plot. The bar on the left represents the percentage of variance explained by the experimental variables.

Right: enrichment patterns of precipitation biomarker taxa in response to the different treatments, relative to the pre-conditioning baseline. Rectangles outlined in black indicate bacterial markers that were significantly enriched (red) or depleted (blue) (q < 0.1). **c**, Constrained ordination of soil metatranscriptome profiles after the conditioning-phase treatments. Bray–Curtis dissimilarity matrices were calculated from RNA-based bacterial counts to assess transcriptional differences across soil precipitation legacy groups. Differences among groups were tested using PERMANOVA with 9,999 permutations; $R^2$ and P values are shown on the plot. The bar on the left indicates the proportion of variance explained by the experimental variables. **d**, Principal coordinates analysis of standardized pairwise genetic distances calculated from SNPs in the genomes of the precipitation biomarker taxa. Genetic variation between groups was assessed using analysis of molecular variance, with significance tested by permutation ($Φ$ and P values are shown on the plot). To quantify genetic differentiation among groups, fixation index ($F_{ST}$) values were also calculated and are displayed in the plot. For **b**–**d**, note that even after 5 months of experimental perturbation, there is a clear separation of the samples on the first axis based on the precipitation legacy. H, host; Rep, biological replicate.

62 GO categories that were associated with precipitation legacy in the pre-conditioning soils, 49 or 50 retained the same pattern after 5 months of experimental drought or ample watering, respectively (Extended Data Fig. 2g and Supplementary Table 7). These results show that the legacy effect of precipitation on soil functional capacity was robust to the conditioning-phase perturbations.

Metagenome data often include unexpressed genes and sequences from dormant or dead organisms, which could exaggerate the robustness of soil legacy effects. Therefore, we also quantified metatranscriptomes from the same samples to focus on biologically active processes. Precipitation legacy explained 24.9% of the variation in microbial gene expression, while conditioning-phase drought and host treatments explained only 12.3% and 1.8% of the variation, respectively (Fig. 2d and Extended Data Fig. 5a). Furthermore, metatranscriptome-based taxonomic patterns confirmed the metagenome-based results: even after 5 months of experimental perturbation, transcriptionally active Actinomycetota and Bacillota remained enriched in low-precipitation soils, while Acidobacteriota, Planctomycetota and Pseudomonadota remained enriched in high-precipitation soils (Extended Data Fig. 5b). Overall, patterns in transcriptionally active bacterial taxa mirrored our previous metagenome-based observations, regardless of the conditioning treatments (Supplementary Fig. 2a and Extended Data Figs. 5c and 6a).

Notably, the metatranscriptome analysis revealed GO categories and KEGG reactions that remained enriched in low-precipitation soils after experimental perturbation, such as the tetrapyrrole metabolic process, response to osmotic stress, liposaccharide metabolic process, haem metabolic process, and trehalose catabolic process (Extended Data Fig. 6b and Supplementary Tables 8 and 9). These results confirm that precipitation legacy strongly shapes gene expression in soil microbiota and remains robust to perturbation (for example, a 5-month-long drought). This functional resilience creates the potential for microbial legacy effects to influence host responses to environmental changes, including plant resilience to future droughts.

**Microbial precipitation legacy alters host drought responses**

During the conditioning phase, drought-treated gamagrass plants grown in low-precipitation-legacy soils were larger and produced more aerenchyma than plants grown in high-precipitation-legacy soils (Fig. 2a). To confirm that these effects of precipitation legacy were conferred by the microbiota rather than co-varying soil properties, we inoculated a new generation of gamagrass seedlings with the microbial communities from the conditioning-phase pots (Extended Data Fig. 4a). These 'test-phase' plants were divided between well-watered conditions ($N = 100$) and a drought treatment ($N = 200$), which was severe enough to impair plant growth (Supplementary Fig. 5a). We phenotyped 5-week-old plants and sampled crown roots, which are highly active in water acquisition[23,24], for RNA-sequencing (RNA-seq) analysis and *16S rRNA* gene microbiota profiling. We focused on bacterial microbiota because bacterial sequences accounted for >89.9% of metagenomic reads in these soils (Supplementary Fig. 6) and because previous work indicated that fungi from these soils are insensitive to drought[25]. To best capture the precipitation gradient extremes, we conducted RNA-seq only in plants inoculated with microbiota from the two lowest-precipitation and two highest-precipitation soils.

Precipitation legacy affected neither α-diversity nor taxonomic composition of the gamagrass root microbiota. Taxonomic composition was impacted by the test-phase drought treatment but not inoculum precipitation legacy (Fig. 3a,b); together these factors explained 2.2% of microbiota variation ($F_{2,139} = 1.74$, $R^2 = 0.022$, $P = 0.01$). β-Dispersion was equal among treatment groups (Fig. 3c). Together, these results suggest that gamagrass exerts a strong homogenizing influence during root microbiota formation, resulting in a stable and drought-resistant microbiota. As gamagrass shares a long evolutionary history with Kansas soil microorganisms[26], this result supports previous findings that co-evolution promotes stable community assembly[27,28].

Although precipitation legacy did not shape *16S rRNA* gene diversity within gamagrass roots, its robust effect on the metatranscriptome (Fig. 2c,d) suggested high potential to influence plant phenotype. Indeed, the root gene expression profiles of plants inoculated with dry-legacy microbiota were distinct from those of plants inoculated with wet-legacy microbiota. Fifteen gamagrass genes were differentially expressed in plants receiving a high-precipitation versus low-precipitation inoculum (Fig. 4a and Supplementary Table 10). Furthermore, inoculum precipitation legacy influenced the responses of 183 gamagrass genes to the drought treatment. Of these, 55% were unresponsive to drought in plants grown with a dry-legacy microbiota but were downregulated or upregulated in plants grown with a wet-legacy microbiota (Fig. 4b, gene sets I and II, respectively, and Supplementary Table 10). This strongly suggests that low-precipitation-legacy soil microbiota tend to dampen the transcriptional response of gamagrass to acute drought. For instance, 50 gamagrass genes were downregulated due to the drought treatment but only in plants that had been inoculated with high-precipitation-legacy microbiota (Fig. 4b, gene set I). These included five orthologues of maize genes with predicted involvement in ethylene- or ABA-mediated signalling of water stress (*Td00002ba004498*, *Td00002ba024351*, *Td00002ba011993*, *Td00002ba005402* and *Td00002ba000033*), and a heat shock protein linked to temperature stress (*Td00002ba042486*). Notably, the latter three genes were not identified as drought-sensitive genes when averaging across inocula, showing that microbial context is necessary for a complete understanding of plant drought responses. We note that these signatures of microbiota precipitation legacy on host gene expression are averaged across all treatments applied during the conditioning phase, again confirming that microbiota legacy effects are robust to short-term perturbations.

Next, we assessed whether microbial precipitation legacy altered host phenotypic drought responses. For 63 plant traits, we calculated a drought susceptibility index (*S*-index) that measures trait stability under drought relative to control plants that received the same inoculum type (Supplementary Table 11). A random forest model identified the ten most important plant traits for describing microbiota precipitation legacy effects (Supplementary Fig. 7a,b). Microbiota precipitation legacy explained 5% of the total variation in host phenotypic response to acute drought (redundancy analysis $F_{1,187} = 10.1$, $P = 0.001$) and impacted 6 of the top 8 non-collinear traits: transpiration, chlorophyll content, maximum root width, metaxylem area, median root diameter, Cu, K and intrinsic water-use efficiency (iWUE)−an important plant drought-response trait[29,30] (Extended Data Fig. 7).

To solidify the mechanisms linking microbiota legacy to plant phenotype, we used mediation analysis to determine whether phenotypic drought responses were mediated by expression patterns of the 198 genes that responded to inoculum precipitation legacy (Fig. 4a–c), summarized in 2 multidimensional scaling (MDS) dimensions (MDS1 and MDS2). MDS1 mediated 18% of the drought-induced decrease in iWUE ($P < 0.001$) and 8.9% of the decrease in transpiration ($P = 0.018$; Fig. 4d). In contrast, MDS2 mediated 11% of the iWUE response to drought ($P = 0.03$), but in the opposite direction, meaning that the activity of these genes counteracted the negative effects of acute drought on iWUE. Four of the top 8 genes with the strongest positive loadings on MDS2 were orthologues of the *Zea mays* (Zm) maize gene Zm*NAS7* (nicotianamine synthase); 2 of these were also in the top 5% of genes with the strongest negative loadings on MDS1 (Extended Data Table 1), indicating that they stabilize iWUE during drought. Drought increased transcription of all four Zm*NAS7* orthologues but only in plants that were inoculated with dry-legacy microbiota (Fig. 4b, gene set IV), suggesting a possible mechanism by which low-precipitation-legacy microbiota confer drought tolerance. Although nicotianamine is best known for its role in metal transport[31], the overexpression of *NAS* genes has

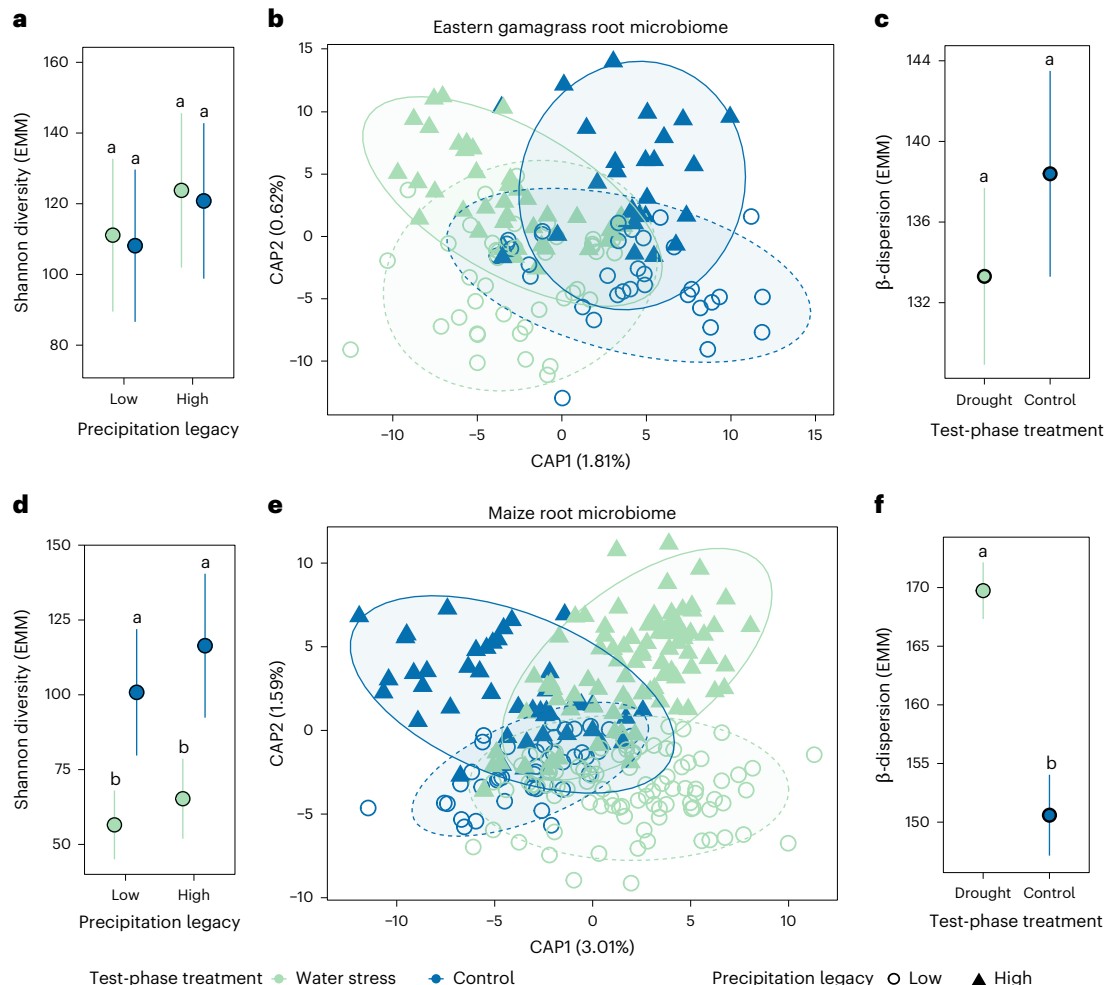

**Fig. 3 | Root bacterial microbiota composition is more stable in *T. dactyloides* than *Z. mays* under varying inocula and water treatments.** For **a**, **c**, **d** and **f**, points are EMMs of biological replicates, error bars represent ±s.e.m., and letters indicate pairwise group differences based on ANOVA followed by a two-tailed Tukey post hoc test and FDR correction ($q < 0.05$). Units of study were individual plants grown in independent pots. Each sample represents one independent biological replicate. These analyses include $n = 146$ gamagrass root microbiota samples and $n = 264$ maize root microbiota samples. **a**, Shannon diversity ($e^{Shannon index}$) was not significantly affected by precipitation legacy (ANOVA, $F_{1,141} = 0.72$, $P = 0.39$) or test-phase water treatment (ANOVA, $F_{1,143} = 0.04$, $P = 0.85$). **b**, Constrained ordination by test-phase water treatment and inoculum precipitation legacy (ANOVA-like permutation test, full model, $F_{2,139} = 1.74$, $R^2 = 0.022$, $P = 0.01$) indicated that test-phase water treatment (by term, $F_{1,139} = 2.58$, $P = 0.003$) but not legacy (by term, $F_{1,139} = 0.89$, $P = 0.55$)

significantly impacted gamagrass root bacterial composition. Ellipses indicate 95% confidence intervals; solid lines for high precipitation legacy and dashed lines for low precipitation legacy. **c**, Drought treatment did not significantly affect within-group variation (β-dispersion) of gamagrass root microbiomes (ANOVA, $F_{1,144} = 0.57$, $P = 0.45$). **d**, Maize root microbiota Shannon diversity was significantly affected by test-phase water treatment (ANOVA, $F_{1,249} = 20.97$, $P = 0.000007$), but not by inoculum precipitation legacy (ANOVA, $F_{1,259} = 1.96$, $P = 0.16$). **e**, Constrained ordination (ANOVA-like permutation test, full model, $F_{2,255} = 6.156$, $R^2 = 0.041$, $P = 0.001$) indicated significant effects of legacy (by term, $F_{1,255} = 4.254$, $P = 0.001$) and test-phase water treatment (by term, $F_{1,255} = 8.058$, $P = 0.001$) on maize root bacterial composition. Ellipses indicate 95% confidence intervals. **f**, Acute drought increased maize root microbiota within-group variation (ANOVA, $F_{1,262} = 20.65$, $P = 0.000008$). All ANOVAs used type III sums of squares.

conferred drought tolerance, including maintenance of photochemical efficiency at pre-drought levels, in rice[32] and in the grass *Lolium perenne*[33]. Together, these results show that low-precipitation-legacy microbiota improve the drought response of gamagrass.

### The implications of legacy effects vary among plant species

To explore whether the observed microbial legacy effects were transferable across hosts, we simultaneously applied the same inocula to seedlings of maize (*Z. mays*), a relative of gamagrass that is non-native to Kansas (Extended Data Fig. 4a). Unlike gamagrass, the maize root microbiota retained taxonomic signatures of both the inoculum precipitation legacy and the drought treatment, which together explained 4.1% of microbiota variation ($F_{2,255} = 6.16$, $P = 0.001$). Also unlike gamagrass, drought treatment increased β-dispersion and decreased α-diversity, indicating lower microbiota stability (Fig. 3d–f).

Next, we identified amplicon sequence variants (ASVs) that were differentially abundant between droughted versus control plants in a manner that depended on the precipitation legacy of the inoculum. These included only 2 ASVs in gamagrass roots (*Azospirillum* sp. and Enterobacteriaceae) but 100 ASVs in maize roots (Supplementary Fig. 8 and Supplementary Table 12). Again, this result indicates that relative to maize, gamagrass root microbiota are more stable—that is, they experience less change in taxonomic composition in response to drought treatment and inoculation with different starting communities.

RNA-seq analysis identified four sets of orthologous genes that were sensitive to microbiota legacy in both plant species, but none showed congruent drought responses (Supplementary Note 1). In maize, 23 genes were upregulated in plants inoculated with dry-legacy microbiota, relative to those that received wet-legacy inocula (Extended Data Fig. 8a). These included six defence-related

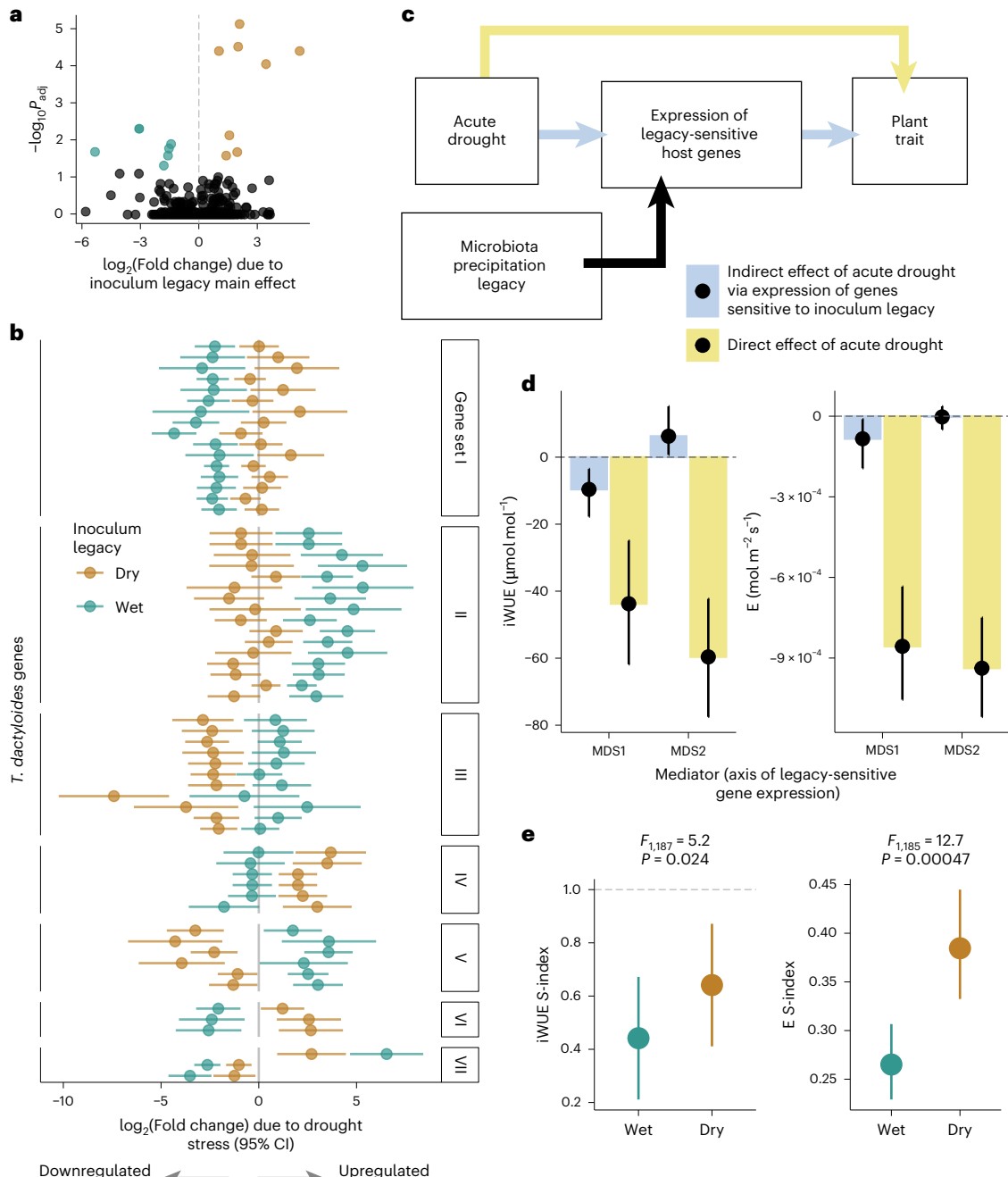

**Fig. 4 | The precipitation legacy of the microbial inoculum mediates transcriptional and physiological responses of *T. dactyloides* to acute drought. a**, Fifteen genes were differentially expressed between plants inoculated with low-precipitation-legacy microbiota (brown) versus high-precipitation-legacy microbiota (turquoise). **b**. In total, 183 gamagrass genes responded to drought in a manner that was dependent on the drought legacy of the soil microbiota (the inoculum legacy × drought treatment interaction; Supplementary Table 10). Only genes with a drought response of ≥4-fold and with annotated maize orthologue(s) are shown for illustration purposes. Each pair of points represents one gene; the position of each point illustrates how the gene's transcription level responded to drought treatment in plants inoculated with low-precipitation-legacy (brown) or high-precipitation-legacy microbiota (turquoise). Gene sets correspond to patterns of how inoculum legacy altered their drought responses. **c**, The model used for mediation analysis to test whether the expression of the 198 legacy-sensitive genes (in **a** and **b**) contributed to the overall effect of acute drought on plant phenotype. **d**, Mediation analysis confirmed that the expression of legacy-sensitive genes (summarized in two

dimensions, MDS1 and MDS2) is involved in drought-induced decreases in iWUE (left) and transpiration (E; right). Yellow bars indicate the 'direct' effect of the drought treatment on trait values, that is, the portion of the trait response that is independent of the transcription levels of microbiota legacy-sensitive genes. Blue bars show the portion of the trait response that is mediated by microbiota legacy-sensitive genes. Points show the mean mediation effect and the error bars show 95% confidence intervals. **e**, Low-precipitation-legacy microbiota (brown) stabilized iWUE and E during acute drought. The *S*-index describes trait values scaled by the mean value of well-watered control plants, such that an *S*-index of 1 indicates that drought-treated plants were phenotypically identical to non-drought-treated plants. Points are EMMs from linear mixed-effects models and error bars indicate 95% confidence intervals; statistical support is from ANOVA with type III sums of squares. For **a** and **b**, points indicate the estimated mean $\log_2$-transformed fold change with 95% confidence intervals (95% CIs), and statistical support is derived from a two-sided Wald test of the null hypothesis that the $\log_2$(fold change) = 0, with FDR adjustment of *P* values ($P_{adj}$).

genes, particularly linked to jasmonic acid signalling, suggesting that low-precipitation-legacy microbiota were perceived as pathogens by maize but not gamagrass. Other notable functions showing this pattern include root development and iron acquisition (Supplementary Table 10). In addition, 109 maize genes responded to drought in a manner that was dependent on inoculum precipitation legacy. Only 30 of these were identified as drought-responsive genes when averaging across inocula (Extended Data Fig. 8b), reinforcing the importance of microbial context for understanding plant drought responses. As in gamagrass, most of these genes were drought responsive only in wet-legacy-inoculated or only in dry-legacy-inoculated plants (Extended Data Fig. 8c). For example, 30 genes were upregulated in response to drought only in plants grown with wet-legacy microbiota (Extended Data Fig. 8c, gene set I), including several related to pathogen defence (*Zm00001eb116230*, *Zm00001eb150050* and *Zm00001eb222540*) and response to symbiotic fungi (*Zm00001eb033580*). This suggests that high-precipitation-legacy soils contain both harmful and beneficial microorganisms and that water deprivation activates interactions with both groups in maize. Notably, one of the relatively few genes that reversed its drought response depending on the inoculum's precipitation legacy was *TIP3*, which encodes an aquaporin that regulates vacuolar water storage[34]. Drought caused a seven-fold increase in *TIP3* expression in wet-legacy-inoculated maize, but a nearly three-fold decrease in dry-legacy-inoculated maize (Extended Data Fig. 8c, gene set VI). Averaged across the inocula, however, *TIP3* expression was stable; thus, its role in drought response was apparent only when accounting for microbiota precipitation legacy.

Finally, we investigated whether the beneficial effects of dry-legacy microbiota on gamagrass phenotype were also conferred on maize. Overall, microbiome precipitation legacy explained only 2.0% of the phenotypic variation in maize (Extended Data Fig. 9a), a weaker effect than that observed in gamagrass. In general, microbiota legacy had little impact on maize phenotype (Extended Data Fig. 9). Thus, the capacity to integrate the beneficial effect of soil precipitation legacy is host specific and may be related to the host's ability to maintain a stable microbiota during drought.

## Discussion

Microbially mediated precipitation legacy effects can affect plant performance during subsequent droughts[35,36]. Our results reveal how these effects manifest in terms of microbiota taxonomic composition, functional potential, gene expression, and strain-level genetic variation, and how they affect host transcriptional and phenotypic drought responses. We identified bacterial taxa, genes and functional pathways that are associated with precipitation, suggesting increased nitrogen-metabolizing and DNA repair capacity in dry-legacy microbiota. Thus, we provide evidence for the ecological and molecular mechanisms of precipitation legacy effects and show their robustness to a 5-month-long perturbation, which mimics a particularly dry or wet season.

We have shown that metagenomic precipitation legacy has measurable implications for plant drought response. In the bunch-grass *T. dactyloides*, dry- and wet-legacy soil microbiota gave rise to taxonomically similar root bacterial communities yet had strikingly different effects on plant gene expression and phenotype during a subsequent acute drought. Inoculation with dry-legacy microbiota altered transcription of key genes that mediated iWUE and transpiration during drought. These benefits did not extend to maize, which also had a relatively unstable root microbiota and lower phenotypic sensitivity to microbial legacy during drought. However, further research is needed to confirm whether microbiota stability is a mechanism of adaptive plant drought responses. Importantly, these differences between gamagrass and maize responses to microbiota legacy indicate that crops may not reap the same benefits as native plant species from

potentially beneficial microbial communities. Therefore, our findings contribute to our mechanistic understanding of precipitation legacy effects, their robustness and their role in plant drought responses, with implications for agricultural and natural ecosystem management.

## Methods

No statistical methods were applied to predetermine sample size. The experiments were randomized and investigators were blinded to treatment allocation during experiments and outcome assessment (except for visually obvious categories, such as plant species).

### Legacy-phase soil characterization across precipitation gradients

**Region selection and soil collection.** Soils were sampled from six never-irrigated native prairies across Kansas in the United States in October 2020. This selection includes the following eastern Kansas tallgrass prairies: Welda Prairie (WEL), Clinton Wildlife Reserve (CWR) and Konza Wildlife Reserve (KNZ). It also includes the following western Kansas shortgrass prairies: Hays Prairie (HAY), Smoky Valley Ranch (SVR) and Kansas State University's Tribune Southwest Research Center (TRI). The GPS coordinates of each collection site are available in Supplementary Table 1. All soil collections were made in accordance with local laws and permitting requirements, and using methods that minimized disturbance to the local ecosystem. For soil collection, each site was split into three subplots. In each subplot, the surface soil (~10 cm) containing thick plant root masses was removed using a bleach-sterilized metal shovel. Then, approximately 2.5 l of soil was collected from each subplot and pooled into a clean plastic bag, for a total of ~7.5 l of soil collected per site. Soil was held at room temperature for transport back to the laboratory, where it was then stored at 4 °C until use in the growth chamber experiments (~1 month).

Subsamples for downstream metagenomic sequencing and nutrient mineral content analyses were air dried in sterile plastic trays at room temperature for 1 week and then sieved using a 2-mm sieve to remove rocks, big soil particles and vegetable debris. The subsamples were shipped to the University of Nottingham for further processing, and all subsamples were stored at 4 °C until use (~3 days).

**Precipitation data collection.** Daily precipitation from 1981 to 2021 was extracted from the NASA POWER database based on the latitude and longitude of each site[37].

**Soil elemental content analysis.** The soil mineral nutrient and trace element profiles were determined using ICP-MS. The soil samples were dried using plastic weighing boats in the fume hood for approximately 72 h at ambient temperature. For each sample, 5 g of soil was weighed into a 50 ml conical tube using a 4-decimal balance and treated with 20 ml of 1 M $NH_4HCO_3$, 5 mM diamine-triamine-penta-acetic acid (DTPA) and 5 ml of 18.2 MΩcm Milli-Q Direct Water (Merck Millipore) for 1 h at 150 rpm in a rotary shaker (adapted from ref. 38) to extract available elements. Each treated sample was gravity filtered through a quantitative filter paper (grade 42, WHA1442070; Whatman) to obtain approximately 5 ml of filtrate. Before the digestion, 20 µg l$^{-1}$ of indium was added to the nitric acid Primar Plus (Fisher Chemicals) as an internal standard for assessing error in dilution, variations in sample introduction and plasma stability in the ICP-MS instrument. Next, 0.5 ml of the soil filtrates were open-air digested in glass Pyrex tubes using 1 ml of concentrated trace metal grade nitric acid-spiked indium internal standard for 2 h at 115 °C in a dry block heater (SCP Science DigiPREP MS; QMX Laboratories). After cooling, the digests were diluted to 10 ml with 18.2 MΩcm Milli-Q Direct Water and elemental analysis was performed using a PerkinElmer NexION 2000 ICP-MS equipped with Elemental Scientific 4DXX FAST Dual Rinse autosampler, FAST valve and peristaltic pump. The instrument was fitted with PFA-ST3 MicroFlow nebulizer, baffled cyclonic C3 high-sensitivity

glass spray chamber cooled to 2 °C with a PC3X Peltier heated–cooled inlet system, 2.0 mm internal diameter quartz injector torch and a set of nickel cones. A total of 23 elements were monitored, including the following stable isotopes: [7]Li, [11]B, [23]Na, [24]Mg, [31]P, [34]S, [39]K, [43]Ca, [52]Cr, [55]Mn, [56]Fe, [59]Co, [60]Ni, [63]Cu, [66]Zn, [75]As, [82]Se, [85]Rb, [88]Sr, [98]Mo, [111]Cd, [208]Pb and [115]In. Helium was used as a collision gas in kinetic energy discrimination (KED) mode at a flow rate of 4.5 ml min[−1] while measuring Na, Mg, P, S, K, Ca, Cr, Mn, Fe, Ni, Cu, Zn, As, Se and Pb to exclude possible polyatomic interferences. The remaining elements were measured in standard mode. Any isobaric interferences were automatically corrected by the instrument Syngistix software for ICP-MS v.2.3 (Perkin Elmer). The ICP-MS measurements were performed in peak-hopping scan mode, with dwell times ranging from 25 ms to 50 ms, depending on the element, 20 sweeps per reading and 3 replicates. The ICP-MS conditions were as follows: radio frequency power of 1,600 W and auxiliary gas flow rate of 1.20 l min[−1]. Torch alignment, nebulizer gas flow and quadrupole ion deflector voltages (in standard and KED mode) were optimized before analysis for highest intensities and lowest interferences (oxides and doubly charged ions levels lower than 2.5%) with NexION Setup Solution containing 1 μg l[−1] of Be, Ce, Fe, In, Li, Mg, Pb and U in 1% nitric acid using a standard built-in software procedure. To correct for variation between and within ICP-MS analysis runs, liquid reference material was prepared using pooled digested samples, run after the instrument calibration and then run after every nine samples in all ICP-MS sample sets. Equipment calibration was performed at the beginning of each analytical run using 7 multi-element calibration standards (containing 2 μg l[−1] indium internal standard) prepared by diluting 1000 mg l[−1] Single Element Standard solutions (Inorganic Ventures; Essex Scientific Laboratory Supplies) with 10% nitric acid. As a calibration blank, 10% nitric acid containing 2 μg l[−1] indium internal standard was used and it was run throughout the course of the analysis.

Sample concentrations were calculated using the external calibration method within the instrument software. Further data processing, including the calculation of final element concentrations, was performed in Microsoft Excel. First, sample sets run at different times were connected as an extension of the single-run drift correction. Linear interpolation between each pair of liquid reference material standards was used to generate a theoretical standard for each sample, which was then used to correct the drift by simple proportion to the first liquid reference material standard analysed in the first run. Liquid reference material composed of pooled samples was used instead of a certified reference material to match the chemical matrix of the samples as closely as possible, thereby emulating the sample drift. Second, the blank concentrations were subtracted from the sample concentrations and each final element concentration was obtained by multiplying by the dilution factor and normalizing the element concentrations to the sample's dry weight.

**Soil porosity analysis.** To quantify soil porosity, we used X-ray computed tomography. First, six soil samples were collected in October 2020 from each of the selected locations in Kansas (see 'Region selection and soil collection'). The top layer of soil was removed (10 cm), and the soil samples were collected using polyvinylchloride columns of 5 cm internal diameter and 7 cm length. After soil collection, the bottom of each column was sealed with tape to retain the soil in the column.

The undisturbed soil columns were non-destructively imaged using Phoenix V|tome|x MDT (Waygate Technologies) at the Hounsfield Facility (University of Nottingham). Scans were acquired by collecting 2,695 projection images at 180 kV X-ray energy, 200 μA current and 334 ms detector exposure time in fast mode (15 min total scan time per column). Scan resolution was 55 μm.

**DNA extraction for metagenomic analysis of free-living soil microbiota.** For metagenomic analysis, approximately 5 g of each soil sample was transferred into 50-ml conical tubes containing 20 ml of sterile

distilled water. To remove large plant debris and soil particles, the samples were shaken thoroughly and filtered into new sterile 50-ml tubes using 100-μm nylon mesh cell strainers. The filtered soil solutions were centrifuged at high speed for 20 min in a Eppendorf 5810R centrifuge and most of the supernatants were discarded. The remaining 1–2 ml of supernatant was used to dissolve the soil pellets. The resulting suspensions were transferred to sterile 1.5-ml Eppendorf tubes. Samples were centrifuged again at high speed in a benchtop centrifuge and the supernatants were discarded. The remaining pellets were stored at −80 °C for DNA extraction. For DNA isolation, we used the 96-well-format MoBio PowerSoil Kit (Mo Bio Laboratories; Qiagen) following the manufacturer's instructions. Before starting the extraction, all samples were manually randomized by placing them in a plastic bag and shaking several times. Samples were then taken individually from the bag and loaded onto the DNA extraction plates. This random distribution was maintained throughout library preparation and sequencing.

**Metagenomic library preparation and sequencing.** DNA sequencing libraries were prepared using the Rapid PCR Barcoding Kit (SQK-RPB004) from Oxford Nanopore Technologies. In brief, 1 μl Fragmentation Mix was added to 3 μl DNA (2–10 ng μl[−1]), and the reaction was mixed by gentle finger flicking. The DNA was fragmented using the following conditions: 30 °C for 1 min and then 80 °C for 1 min in an Applied Biosystems Veriti 96-Well Thermal Cycler (Applied Biosystems). The fragmented DNA was cooled, barcoded and amplified in a PCR reaction containing 20 μl nuclease-free water, 25 μl LongAmp Taq 2X Master Mix (New England Biolabs), 4 μl fragmented DNA and 1 μl barcode adaptor. The reaction was gently mixed and amplified using the following conditions: 95 °C for 3 min, 20 cycles of denaturation at 95 °C for 15 s, annealing at 56 °C for 15 s, extension at 65 °C for 6 min, and a final extension of 65 °C for 6 min. The resulting DNA library was purified using 0.6× Agencourt AMPure XP beads (Beckman Coulter) and eluted in 10 μl 10 mM Tris-HCl pH 8.0 and 50 mM NaCl. The library concentration was determined using a Qubit 4 Fluorometer with the Qubit dsDNA HS Assay Kit (Thermo Fisher Scientific). Equimolar quantities of individual barcoded sample libraries were pooled and the volume was adjusted to 10 μl using 10 mM Tris-HCl pH 8.0 and 50 mM NaCl. Subsequently, 1 μl rapid sequencing adaptor was added to the pooled library and the tube was incubated at room temperature for 5 min. Then, 34 μl sequencing buffer, 25.5 μl loading beads and 4.5 μl nuclease-free water were added to the tube and the contents were mixed gently. The prepared pooled library was added to a verified and primed FLO-MIN106 R9.4.1 flow cell in a MinION (both Oxford Nanopore Technologies) following the manufacturer's instructions. DNA sequencing was conducted with default parameters using a MinIT with MinKNOW v.2.1.12 (both Oxford Nanopore Technologies). Fast5 files were base called with Guppy v.4.0.15 using the template_r9.4.1_450bps_hac.jsn high-accuracy model (Oxford Nanopore Technologies).

**Metagenomic sequence processing.** The initial dataset underwent demultiplexing, and primer and barcode sequences were trimmed using qcat v.1.1.0 (Oxford Nanopore Technologies). Reads with ambiguous barcode assignments were excluded from further analysis. The reads were filtered with NanoFilt v.2.8.0 (ref. [39]) to discard low-quality sequences based on Phred quality score (Q-score < 9), which is a logarithmic measure of the probability of an incorrect base call, and sequences <100 bp. We used the Kraken v.2.1.2 pipeline[40] to classify the whole metagenome shotgun sequencing reads. The reads were classified using the Kraken 2 archaea, bacteria, viral, plasmid, human, UniVec_Core, protozoa and fungi reference database (k2_pluspf_20220607). To estimate relative abundances, the Bracken v.2.7 pipeline[41] was applied to the classification results. Subsequently, Pavian v.1.0 (ref. [42]) was used to extract abundance and taxonomic tables.

**Precipitation gradient data analysis.** *Precipitation data analysis.* To determine the differences in precipitation levels across regions, we compared the mean annual precipitation in each region by fitting a linear model with the following formula:

$$\text{Mean annual precipitation} \sim \text{region}$$

Differences between regions were indicated using the confidence letter display derived from Tukey's post hoc test implemented in the package multcomp v.1.4.25 (ref. 43). We inspected the normality and variance homogeneity (here and elsewhere) using Q–Q plots and the Levene test, respectively. We visualized the results via a point range plot using the ggplot2 v.3.5.1 R package[44].

*Soil elemental content analysis.* For the soil elemental profile, we created a matrix in which each cell contained the calculated element concentration in one sample. Then, we applied a *z*-score transformation to each ion across the samples in the matrix. Afterwards, we applied a principal component (PC) analysis using the Euclidean distance between samples and the *z*-score matrix as input to compare the elemental profiles of soils. In addition, we estimated the variance explained by porosity, precipitation, region and the interaction between them by performing a PERMANOVA via the function adonis2 from the vegan v.2.6-4 R package[45]. The significant variables were visualized via a stacked bar plot using the ggplot2 v.3.5.1 R package.

To visualize the mineral content in each region, the *z*-score matrix created above was hierarchically clustered (method ward.D, function hclust), and we visualized the results using a heat map. The rows in the heat map were ordered according to the dendrogram order obtained from the clustered ions, and the regions were ordered according to the precipitation gradient (low precipitation to high). The heat map was coloured based on the *z*-score.

To explore the relationship between ion concentrations and the precipitation gradient, we performed a correlation test using the cor function from the stats v.4.3.1 package in R[46] of the average *z*-score measurement of each ion against the precipitation gradient. Afterwards, we plotted the correlation coefficient for each ion in a bar plot.

*Soil porosity analysis.* The soil core images taken using X-ray computed tomography were analysed using ImageJ v.1.54b[47]. First, XY slice projection images were filtered using a median function to remove any noise from the raw data, and then an automatic threshold (Li method) was applied to produce binary images. In the binary images, pores and solid particles were represented by black and white pixels, respectively. Afterwards, a region of interest was defined in the central part of each projection to remove any potential border effect, cropping to a 600 × 600 pixel area. From the region of interest defined in each image, we extracted soil features, including particle area, perimeter, circularity, roundness, solidity, compactness, percentage of pores and pore size using the measurement function.

To remove the variability in the topsoil due to transportation and handling, we used two strategies. First, we plotted the pore average size (mm) and soil porosity (%) for each sample. Then, we excluded from the analysis all projections with a value of soil porosity >40% and considered the topsoil of those samples as the first projections after the exclusion. The second strategy was applied to samples with an irregular shape (for example, mountain-like shape) at the topsoil level. We discarded all the projections with an irregular shape until we found projections with a regular distribution of soil layers. We created a data frame with the projection number (or slice) and the soil depth (projection number multiplied by the resolution). The soil porosity for each soil type was visualized via a point plot using the ggplot2 v.3.5.1 R package[44].

To compare the soil elemental profiles against soil porosity, we first applied a *z*-score transformation of each ion and soil porosity across the samples. Then, we estimated the distance between samples using Euclidean distance. Afterwards, we contrasted the dissimilarity matrices of each pair of datasets (soil elemental profile versus soil porosity) using the mantel test implemented in the vegan v.2.6-4 R package[45]. Finally, we computed the significance of the correlation between matrices by permuting the matrices 10,000× (ref. 43).

*Taxonomic data analysis.* To compare the α-diversity across regions, we calculated the Shannon diversity index using the diversity function from the vegan v.2.6-4 package in R[45]. We used an analysis of variance (ANOVA) to test the α-diversity differences between regions. Differences between regions were indicated using the confidence letter display derived from Tukey's post hoc test implemented in the R package multcomp v.1.4.25 (ref. 43).

The β-diversity analysis (principal coordinate analysis) was based on Bray–Curtis dissimilarity matrices calculated using the rarefied relative abundance tables. In addition, we estimated the variance explained by soil porosity, precipitation, regions and the interaction between them by performing a PERMANOVA via the function adonis2 from the vegan v.2.6-4 R package[45]. The significant variables were visualized via a stacked bar plot using the ggplot2 v.3.5.1 R package[44].

The relative abundance of bacterial phyla was depicted using a stacked bar plot using the ggplot2 v.3.5.1 package.

To compare the microbiota composition with the elemental profiles and soil porosity, we contrasted the dissimilarity matrices of each pair of datasets (soil microbiota versus soil elemental profile and soil microbiota versus soil porosity) using the mantel test implemented in the vegan v.2.6-4 R package[45]. Briefly, we calculate the microbiota dissimilarity matrix using the Bray–Curtis distance. Next, we applied a *z*-score transformation of each ion and soil porosity across the samples. Afterwards, we calculated the soil elemental dissimilarity matrix and soil porosity dissimilarity matrix using Euclidean distance. Finally, we used the mantel test to compare and test the significance of the correlation between matrices by permuting the matrices 10,000× (ref. 43).

*Heat map and enrichment analysis.* We used the R package DESeq2 v.1.40.2 (ref. 48) to compute the bacterial enrichment profiles in the soils across the precipitation gradient. For each NCBI taxID in the rarefied table and at the species and family levels, we estimated difference in abundance compared with the wettest collection site (Welda Prairie) using a generalized linear model (GLM) with the following design:

$$\text{Abundance} \sim \text{region}$$

We extracted the following comparisons from the fitted model: CWR versus WEL, HAY versus WEL, KNZ versus WEL, SVR versus WEL and TRI versus WEL. A taxID, species or family was considered statistically significant if it had $P < 0.05$ after adjusting for multiple comparisons using the Benjamini–Hochberg method. We visualized the results using a heat map. The rows in the heat map were ordered according to the dendrogram obtained from the taxID, species and family analysis. The relative abundance matrix was standardized across the significant taxID, species and family by using the *z*-score, and the heat map was coloured based on this value.

*Identification of marker taxa associated with precipitation gradients.* To identify the corresponding bacterial isolates considered as 'biomarker' taxa associated with precipitation gradients, we identified the PCs that explain more than 80% of the variance in the data. These identified PCs were used to control the effects of soil elemental profile on the taxa abundances. Then, we fit five models: a Poisson model, a negative binomial model, two zero-inflated models and a multiple regression model.

For the Poisson and negative binomial models, we fitted the following design:

$$\text{Abundance} \sim \text{precipitation} + \text{offset (porosity)}$$
$$+ \text{offset (soil elemental profile: six first PCs)}$$

In parallel, we fitted the zero-inflated models using the following design:

$$\text{Poisson: abundance} \sim \text{precipitation} + \text{offset (porosity)} |$$
$$1 + \text{offset (soil elemental profile)} |1$$

$$\text{Negative binomial: abundance} \sim \text{precipitation} + \text{porosity}$$
$$|\text{porosity} + \text{soil elemental profile}|\text{soil elemental profile}$$

Next, to assess the statistical significance, we applied ANOVA to the best-performing model for each taxon according to the Akaike information criterion. In addition, we applied a multiple regression model with the following design:

$$\text{Abundance} \sim \text{precipitation} + \text{porosity} + \text{soil elemental profile}$$

Then, we applied ANOVA to find taxIDs with a significant partial regression coefficient for precipitation.

A taxID was considered a marker if it had a relative abundance >0.01 and a prevalence >20%. We visualized the average standardized relative abundance ($z$-score) of the significant taxID in a point plot using the ggplot2 v.3.5.1 package[44].

*Enrichment of bacteria biological functions associated with precipitation gradients.* To identify biological processes enriched within the microbial communities, the sequence reads were assembled into contigs for each sample using metaFlye from the Flye v.2.9 package[49] with default mode. The generated contigs were then grouped and de-duplicated using the dedupe.sh tool in BBTools v.38.76 (ref. 50) to eliminate redundancies. Next, we determined the relative abundance of the contigs by mapping the reads from the samples to the contigs using minimap2 v.2.17 (ref. 51) and extracting the relative abundance counts using CoverM v.0.6.1 (ref. 52) in the 'contig' mode and reads_per_base coverage method. Taxonomic classification of the contigs was performed using the CAT v.8.22 taxonomic classification pipeline[53]. Subsequently, the contigs were filtered to retain only bacteria sequences. DESeq2 v.1.40.0 (ref. 48) was used to determine the contig enrichment profiles by fitting a GLM with the following design:

$$\text{Abundance} \sim \text{legacy}$$

The low-precipitation soil versus the high-precipitation soil contrast was extracted from the fitted model. Contigs meeting the criteria of a false discovery rate (FDR)-adjusted $P$ value ($q$ value) <0.05 and a $\log_2$-transformed fold change >2 were selected for further analysis. Open reading frames encoded within the contigs were predicted using FragGeneScanRs v.1.1.0 (ref. 54) with default settings. This was followed by functional annotation of the predicted proteins using the eggNOG-mapper v.2.1.9 (ref. 55) pipeline with the eggNOG v.5.0.2 database[56] with Diamond v.2.0.11 (ref. 57) and MMseqs2 (ref. 58). The genes annotated with GO classifications were subsequently extracted, and a GO enrichment analysis focusing on biological processes was conducted. This involved using adaptive GO clustering in conjunction with Mann–Whitney $U$-testing, using the GO_MWU tool[59], which evaluates the enrichment of each GO category based on whether genes linked to the GO category are significantly clustered at either the top or bottom of a globally ranked gene list. First, genes were ranked based on the signed $\log_2$-transformed fold change values. For each gene, any

missing parental terms for specific GO categories were then automatically added. Next, fully redundant categories (those containing identical sets of genes) were collapsed into the more specific GO term. To further streamline the analysis, highly similar categories were grouped using complete linkage clustering based on the fraction of shared genes. We used default settings, where GO categories were merged if the most dissimilar pair within a group shared more than 75% of the genes in the smaller category. The merged group was named after the largest category. Significantly enriched and depleted GO categories were then determined by an adjusted $P$ value <0.05. This approach simplified the GO hierarchy and addressed multiple testing, which improved the statistical power of the GO enrichment analysis. The most prominent enriched and depleted GO categories shared across comparisons were visualized in ggplot2 v.3.4.2 and coloured based on the square-root-transformed Δ rank values (enrichment score) of the GO categories.

*Analysis of genetic variation among bacterial lineages along the precipitation gradient.* To assess genetic differences between bacteria lineages along the precipitation gradient, we focused on 15 of the identified bacterial markers (*Pseudomonas* ID287, *Salmonella* ID28901, *Sorangium* ID56, *Bradyrhizobium* ID722472, *Luteitalea* ID1855912, *Bradyrhizobium* ID1355477, *Flavisolibacter* ID661481, *Bradyrhizobium* ID858422, *Rubrobacter* ID2653851, *Bradyrhizobium* ID1437360, *Candidatus Koribacter* ID658062, *Streptomyces* ID1916, *Klebsiella* ID573, *Bradyrhizobium* ID1325107 and *Edaphobacter* ID2703788) and 18 additional abundant and prevalent species (*Rubrobacter* ID49319, *Bacillus* ID1428, *Bradyrhizobium* ID1274631, *Priestia* ID1404, *Lacibacter* ID2760713, *Bradyrhizobium* ID1325100, *Candidatus Solibacter* ID332163, *Burkholderia* ID28450, *Flavisolibacter* ID1492898, *Bacillus* ID1396, *Escherichia* ID562, *Rhizobium* ID384, *Rubrobacter* ID2653852, *Microvirga* ID2807101, *Archangium* ID83451, *Pseudomonas* ID303, *Paenibacillus* ID1464 and *Nitrosospira* ID1231) across the samples as proxies for the broader bacterial communities. These taxa were selected for their high genome coverage across samples, enabling more precise allele frequency estimates. Reference genomes for each species were retrieved from the NCBI Genome database, and the filtered shotgun metagenomic reads were aligned to these genomes using minimap2 v.2.17-r941 (ref. 51). Alignments were sorted and indexed with SAMtools v.1.18 (ref. 60), followed by variant calling using BCFtools v.1.18 (ref. 61). This process identified 23,197,278 sequence variants, which were then filtered using VCFtools v.0.1.16 (ref. 62) to retain only biallelic single nucleotide polymorphisms (SNPs) for further analysis. SNP filtering criteria included a variant quality score >20, a minor allele frequency >0.01, <50% missing data and a minimum sequencing depth of 10× in each sample. After filtering, 23,061 high-quality biallelic SNPs were retained. Genetic distances were computed using PLINK v.1.90p[63] and reduced to 2 dimensions through classical multidimensional scaling with the stats v.4.3.0 package. Principal coordinate analysis plots were created with ggplot2 v.3.4.2 (ref. 44), coloured by soil type, precipitation levels and geographical region.

To identify genes potentially under selection across the precipitation gradient, we conducted a genetic–environment association analysis. For this, SNPs were re-filtered using VCFtools v.0.1.16 (ref. 62) with the same criteria, but allowing <50% missing data and a minimum sequencing depth of 5× per sample. After filtering, 93,013 biallelic SNPs were retained. Subsequently, genetic–environment association analysis was performed using a general linear model in the rMVP v.1.1.1 package[64], with native SNP data imputation and average precipitation at each sampling location as the environmental variable. The genetic structure in the data was corrected using the first ten PCs. Manhattan plots were generated using CMplot v.4.5.1 (ref. 65) and significant associations were identified using the permutation method within the rMVP package.

## Conditioning-phase soil drought legacy is resilient to short-term perturbations

**Experimental design.** The six soils collected from across the Kansas precipitation gradient, as described in 'Region selection and soil collection', were used in this experiment. The conditioning perturbations imposed in these experiments took place over approximately 20 weeks at the University of Kansas from 17 December 2020 to 5 May 2021. Each of the six input soils remained independent throughout the experiment. Mesocosms consisted of a 1:1 (v/v) mixture of field-collected soil to sterile Turface MVP (Turface Athletics). A total of 192 sterile 100-ml pots were filled with the 6 soils and randomly assigned to 1 of 4 conditions in a fully factorial design: with or without a host, and either water-stressed or well-watered. Half the pots were planted with seedlings of the native prairie grass *T. dactyloides* (Eastern gamagrass, cultivar 'Pete'); the rest remained unplanted. These 24 treatment groups (6 soils × 2 water-stressed or well-watered × 2 planted or unplanted) each had $N = 8$ replicates for a total of 192 experimental soils in pots. All mesocosms were allowed to adapt to their watering regimes in a growth chamber set to a 12-h day cycle, with daytime temperature at 27 °C and night-time temperature at 23 °C, and ambient humidity. Well-watered control pots were watered every 1–2 days and water-stressed plants were watered every 3–5 days when plants displayed drought symptoms (for example, leaf curling). All pots were fertilized with 35 ml of 1 ml l$^{-1}$ concentration of Bonide 10-10-10 plant food (Bonide Products) on week 8 and week 12.

**Sample collection.** To characterize the baseline microbial communities going into the conditioning treatments, we sampled 4 replicates of each soil and treatment combination 1 week after beginning the experiment. To collect the samples, the top centimetre of soil was discarded and the remaining soil was homogenized to ensure even sampling of the top, middle and bottom of the pot. Homogenized soil samples (2 g) were placed in a 15-ml tube, flash frozen in liquid nitrogen and stored at −80 °C for microbial DNA and RNA extraction. To characterize the effects of the 4 treatments on the microbial communities, we sampled all remaining replicates at the end of the 20-week conditioning phase. Soil samples were collected as described for the baseline communities, but an additional 6 g of homogenized soil was preserved at 4 °C in 50-ml conical tubes for use as inocula in a downstream experiment. In addition, for the planted pots, we measured gamagrass shoot height before uprooting the plants. We collected samples of a crown root from each plant (each 3 cm long, beginning 2 cm from the base of the plant) and stored them in 50% ethanol (EtOH) at 4 °C for downstream laser ablation tomography (LAT) analysis. The remaining roots and shoots were dried in an oven at 225 °F for 12 h and then weighed separately.

**Changes in bacterial community structure associated with drought and well-watered conditioning with and without a host.** *DNA extraction*. Total DNA was extracted from baseline and post-conditioning soil subsamples using the DNA Set for NucleoBond RNA Soil Kit (Macherey-Nagel) according to the manufacturer's instructions.

*Library preparation and sequencing*. DNA sequencing libraries were prepared using the Rapid PCR Barcoding Kit (SQK-RPB004) from Oxford Nanopore Technologies and sequenced on a FLO-MIN106 R9.4.1 flow cell in a MinION, with a MinIT using MinKNOW v.2.1.12 (ref. 66) (all from Oxford Nanopore Technologies) as described in 'Metagenomic library preparation and sequencing'.

*Sequence processing*. Raw sequence data were demultiplexed and primer and barcode sequences were trimmed using qcat v.1.1.0 (Oxford Nanopore Technologies). Reads with ambiguous barcode assignments were excluded from further analysis. The reads were filtered with NanoFilt v.2.8.0 (ref. 39) to discard low-quality sequences (*Q*-score < 9) and sequences <100 bp. We used the Kraken v.2.1.2 pipeline[40] for

classifying the whole metagenome shotgun sequencing reads. The reads were classified using the Kraken 2 archaea, bacteria, viral, plasmid, human, UniVec_Core, protozoa and fungi reference database (k2_pluspf_20220607). To estimate relative abundances, the Bracken v.2.7 pipeline[41] was applied to the classification results. Subsequently, Pavian v.1.0 (ref. 42) facilitated the extraction of abundance and taxonomic tables. Functions in phyloseq v.1.44.0 (ref. 67) with microbiome v.1.22.0 and microbiomeutilities v.1.0.17 (ref. 68) were used to filter the dataset and remove samples with low read depth (<1,000 reads), remove unidentified taxa and singletons, transform abundance values using rarefaction, subset and merge sample and taxonomic groups, and perform other data frame manipulations.

**Plant biomass.** Root and shoots were detached, dried in an oven at 225 °F for 12 h and then weighed.

**Root LAT analysis.** We collected samples of a crown root from each plant (each 3 cm long, beginning 2 cm from the base of the plant) and stored them in 50% EtOH at 4 °C. The samples were shipped to the University of Nottingham for downstream LAT analysis. Briefly, root segments were dehydrated in 100% methanol for 48 h, transferred to 100% EtOH for 48 h and then dried with an automated critical point dryer (CPD; Leica EM CPD 300; Leica Microsystem). Root anatomical images were acquired using a laser ablation tomograph (LATScan; Lasers for Innovative Solutions). This uses a combination of precise positioning stages with a guided pulsed UV (355 nm) laser to thermally vaporize thin sections of the root and then to illuminate the exposed surface. The tomograph was retrofitted with a microscopic imaging system using a machine vision camera unit (model Grasshopper3; FLIR) and infinity-corrected long working distance magnifying objectives (Mitutoyo).

**Changes in gamagrass morphological features under drought and well-watered conditioning.** To identify morphological features in gamagrass that changed with the conditioning (drought and well-watered) treatments, we used ImageJ to quantify the average area aerenchyma, the number of aerenchyma, the number of metaxylem vessels, number of cortical cell layers, total metaxylem area, average metaxylem area, stele minimum diameter, stele maximum diameter, stele area, stele perimeter, total aerenchyma area, adjusted cortex area, root minimum diameter, cortex minimum diameter, root maximum diameter, cortex maximum diameter, total perimeter, cortex perimeter, root total area and cortex area. In addition, we quantified the root:shoot ratio, number of leaves, number of green leaves, root mass, shoot mass and shoot height.

**Metatranscriptome analysis.** *RNA isolation*. Total RNA was extracted from baseline and post-conditioning soil subsamples with the NucleoBond RNA Soil Kit (Macherey-Nagel) using the manufacturer's instructions. Isolated RNA was treated with Turbo DNA-free (Applied Biosystems) to remove contaminating DNA, following the manufacturer's instructions.

*Library preparation and sequencing*. RNA libraries were prepared using 1 µg of total RNA according to established protocols with modifications. Briefly, poly(A)-tail-containing RNA was removed from the RNA samples using Sera-mag oligo(dT) magnetic beads (GE Healthcare Life Sciences) and then the samples were subjected to ribodepletion with the NEBNext rRNA Depletion Kit (New England Biolabs), following the manufacturers' instructions. The purified RNA was resuspended in a fragmentation mix consisting of 6.25 µl Milli-Q water, 5 µl 5× First-Strand Buffer, and 1.25 µl random primers (3 µg µl$^{-1}$) and fragmented at 94 °C for 6 min. First-strand cDNA synthesis was performed using a mixture of 0.8 µl reverse transcriptase, 2 µl 100 mM DTT, 0.4 µl 25 mM dNTP, 0.5 µl RNAseOUT (40 U µl$^{-1}$), 10 µl RNA and

6.3 µl Milli-Q water. The reactions were incubated at 25 °C for 10 min, 42 °C for 50 min and 70 °C for 15 min. Second-strand cDNA synthesis was performed by adding a master mix of 18.4 µl Milli-Q water, 5 µl 10× Second-Strand Buffer, 1.2 µl 25 mM dNTP, 0.4 µl RNAse H (5 U µl⁻¹) and 5 µl DNA PolII (10 U µl⁻¹) to the sample, followed by incubation at 16 °C for 1 h. The samples were then purified using Agencourt AMPure XP beads. Subsequently, the libraries were end repaired with a mixture of 30 µl sample, 2.5 µl of 3 U µl⁻¹ T4 DNA polymerase, 0.5 µl of 5 U µl⁻¹ Klenow DNA polymerase, 2.5 µl of 10 U µl⁻¹ T4 PNK, 5 µl of 10× T4 DNA ligase buffer with 10 mM ATP, 0.8 µl of 25 mM dNTP mix and 8.7 µl Milli-Q water, incubated at 20 °C for 30 min, and purified again using Agencourt AMPure XP beads. Following this, the RNA libraries were adenylated in a mix containing 34 µl of the end-repaired sample, 3 µl of 5 U µl⁻¹ Klenow exo−, 5 µl of 10× Enzymatics Blue Buffer, 1 µl of 10 mM dATP and 9 µl of Milli-Q water. The mixture was incubated at 37 °C for 30 min, followed by 70 °C for 5 min, and then purified using Agencourt AMPure XP beads. Individual samples were indexed through ligation using a mix of 10.25 µl sample, 1 µl of 600 U µl⁻¹ T4 DNA ligase, 12.5 µl of 2× Rapid Ligation Buffer and 1.25 µl of 2.5 µM indexing adaptor from the KAPA Dual-Indexed Adapter Kit (Kapa Biosystems). The samples were incubated at 25 °C for 15 min, followed by the addition of 5 µl of 0.5 M EDTA pH 8. The libraries were purified twice with Agencourt AMPure XP beads. The libraries were enriched in a reaction containing 20 µl sample, 25 µl of 2× KAPA HiFi HS Mix (Kapa Biosystems), 2.5 µl of 5 µM I5 primer and 2.5 µl of 5 µM I7 primer. The reactions were initially heated to 98 °C for 45 s, followed by 14 cycles of 98 °C for 15 s, 60 °C for 30 s and 72 °C for 30 s, with a final extension at 72 °C for 1 min. The resulting RNA libraries were purified using Agencourt AMPure XP beads, quantified on a Qubit 4 Fluorometer (Thermo Fisher Scientific), and the library size was assessed using High Sensitivity D1000 ScreenTape on the Agilent 4200 TapeStation (Agilent Technologies). Equimolar quantities of individual barcoded RNA libraries were pooled in a randomized manner and shipped on dry ice to Beijing Genomics Institute. Each library pool was sequenced on an MGI Tech MGISEQ-2000 sequencing platform to generate a minimum of 10 million 100-bp paired-end reads per sample.

*Taxonomic classification of transcripts.* Cutadapt v.4.6 (ref. [69]) was used to remove primer and barcode sequences and low-quality sequences from the paired-end reads of the sequenced RNA libraries. To identify taxa with enriched gene expression activity, the reads were classified using the Kraken v.2.1.2 pipeline[40] with the archaea, bacteria, viral, plasmid, human, UniVec_Core, protozoa, and fungi reference database (k2_pluspf_20220607), and the Bracken v.2.7 pipeline[41] was applied to the classification results to estimate the relative abundances. This approach enabled a rapid and comprehensive overview of taxonomic profiles based on read-level assignments, and was primarily used for estimating taxonomic abundance and diversity directly from the sequence data. The counts table was generated from Pavian v.1.0 (ref. [42]). Data filtering and statistical analysis were then performed as before using phyloseq v.1.44.0 (ref. [67]) with microbiome v.1.22.0 (ref. [70]) and microbiomeutilities v.1.0.17 (ref. [68]).

**Data analysis.** *Changes in bacterial community structure associated with drought and well-watered conditioning with and without a host.* To assess the α-diversity across the samples, we calculated the Shannon diversity index using phyloseq v.1.44.0 (ref. [67]). We used ANOVA to test for significant differences in Shannon diversity indices between groups, and means were separated using Tukey's honestly significant difference test from the agricolae v.1.3.5 R package[71]. For β-diversity, Bray–Curtis dissimilarity matrices were calculated using phyloseq v.1.44.0 and the variance explained by legacy, conditioning and host were estimated by performing PERMANOVA using the adonis2 function in the vegan v.2.6.4 R package[45]. Constrained ordination of β-diversity was plotted using canonical analysis of principal coordinates (CAP) based on Bray–Curtis dissimilarity matrices calculated with vegan

v.2.6.4. We visualized differences with the CAP analysis using the following models:

$$\sim Legacy + Condition\,(conditioning + host)$$

$$\sim Conditioning + Condition\,(legacy + host)$$

$$\sim Host + Condition\,(legacy + conditioning)$$

The relative abundance of taxa was plotted as a stacked bar representation using phyloseq v.1.44.0. The tax_glom function in phyloseq v.1.44.0 was used to agglomerate taxa, and the aggregate_rare function in microbiome v.1.22.0 was used to aggregate rare groups. We used DESeq2 v.1.40.0 (ref. [48]) to calculate the enrichment profiles by fitting a GLM with the following design:

$$Abundance \sim legacy + conditioning$$

We extracted the following comparisons from the fitted model: wet soil legacy with watered conditioning versus wet soil legacy (baseline), wet soil legacy with drought conditioning versus wet soil legacy (baseline), dry soil legacy (baseline) versus wet soil legacy (baseline), dry soil legacy with watered conditioning versus wet soil legacy (baseline), and dry soil legacy with drought conditioning versus wet soil legacy (baseline). Taxa were considered significant if they had an FDR-adjusted *P* value (*q* value) <0.05. The results of the GLM analysis were rendered in heat maps, coloured based on the $\log_2$-transformed fold change output by the GLM. Significant differences between comparisons with *q* < 0.05 with $\log_2$-transformed fold change >2 were highlighted with black squares.

Relative abundances of the taxonomic markers were extracted and an ANOVA was performed to assess significant differences between treatment groups. Tukey's honestly significant difference test, implemented using the agricolae v.1.3.5 R package[71], was used for post hoc pairwise comparisons. To further explore the effects of watering and drought treatments, with and without a host, on the relative abundances of the taxonomic markers, we subset the data and applied a GLM using DESeq2 v.1.40.0 (ref. [48]). The model was structured as follows:

$$Abundance \sim legacy + conditioning + host$$

We then extracted the following comparisons from the fitted model for each soil legacy: water conditioning versus baseline, drought conditioning versus baseline, water conditioning with host versus baseline, and drought conditioning with host versus baseline. Markers were considered significant if the FDR-adjusted *P* value (*q* value) was <0.05. Results from the GLM analysis were visualized in a heat map, with colours representing $\log_2$-transformed fold changes. Comparisons showing significant differences (*q* < 0.05 and $\log_2$-transformed fold change >2) were highlighted with black squares.

*GO term enrichment analysis.* To identify enriched biological processes within the microbial communities, sequence reads from individual samples were assembled into contigs using metaFlye from the Flye v.2.9 package[49] with default parameters, as described in 'Enrichment of bacteria biological functions associated with precipitation gradients'. Relative abundance counts were then determined, and the resulting contigs were subjected to taxonomic classification and filtering, also as outlined in 'Enrichment of bacteria biological functions associated with precipitation gradients'. We used DESeq2 v.1.40.0 (ref. [48]) to determine the bacterial contig enrichment profiles by fitting a GLM with the following design:

$$Abundance \sim legacy + conditioning + biological\,replica$$

We extracted the following comparisons from the fitted model: wet soil legacy with watered conditioning versus wet soil legacy (baseline), wet soil legacy with drought conditioning versus wet soil legacy (baseline), dry soil legacy (baseline) versus wet soil legacy (baseline), dry soil legacy with watered conditioning versus wet soil legacy (baseline), and dry soil legacy with drought conditioning versus wet soil legacy (baseline). Contigs meeting the criteria of an FDR-adjusted $P$ value ($q$ value) <0.05 and a $\log_2$-transformed fold change >2 were selected. Open reading frames were predicted and functionally annotated, and genes with GO classifications were subjected to GO enrichment analysis with the GO_MWU tool[59].

*Changes in gamagrass morphological features under drought and well-watered condition.* For each root feature identified, we used ANOVA to test for significant differences between groups, and means were separated using Tukey's honestly significant difference test from the agricolae v.1.3.5 R package[71]. Subsequently, the feature values were normalized using the rescale function from the scales v.1.2.1 R package. The mean normalized feature values were then visually represented on a heat map using ggplot2 v.3.4.2. Then, Pearson correlation coefficients between these features and corresponding $P$ values were computed using the rcorr function from the Hmisc v.5.0.1 package[72]. The results of the correlation analysis were graphically presented using ggplot2 v.3.4.2, where the colour of the plots reflected the correlation coefficient values. Significant correlations ($P < 0.05$) were emphasized with black squares on the plots. Furthermore, the coefficient of variation for the feature values was calculated and depicted using ggplot2 v.3.4.2. The three plots were integrated based on the hierarchical clustering of the Pearson correlation coefficients of the features. The clustering used the ward.D2 method within the hclust function in R, using Euclidean distances calculated using the dist function.

*Metatranscriptome sequence analysis.* To assess transcriptional differences in the activity of the bacterial community, Bray–Curtis dissimilarity matrices were calculated. The variance explained by legacy, conditioning and host was estimated using PERMANOVA with the adonis2 function from the vegan v.2.6.4 R package[45]. β-Diversity patterns were visualized through partial constrained ordination using CAP. We applied CAP analysis to visualize differences using the following models:

$$\sim Legacy + Condition(CondWater + CondHost)$$

$$\sim CondWater + Condition(legacy + CondHost)$$

$$\sim CondHost + Condition(legacy + CondWater)$$

In these models and others described later in the text, the variables have the following meanings: legacy refers to the precipitation regime of the inoculum's original collection site (a factor with two levels, dry and wet). CondWater is our shorthand for 'conditioning-phase watering treatment', a factor with two levels (drought or control). CondHost is our shorthand for 'conditioning-phase host treatment', a factor with two levels (gamagrass or none). Note that in the above models, 'Condition()' is not a variable but part of the syntax of the CAP functions in the vegan[45] package in R. These analyses were 'partial' constrained ordinations, meaning that variation attributed to the factors within the Condition() function was removed or 'partialled out' before the constrained ordination being conducted using the residuals from the partialling-out process. This approach is useful for clearer visualizations of the patterns associated with variable(s) of interest (the factors outside of the Condition() function).

Transcriptional activity among taxa was displayed as a stacked bar plot using phyloseq v.1.44.0. We used DESeq2 v.1.40.0 (ref. 48) to calculate enrichment profiles by fitting a GLM with the following design:

$$Abundance \sim legacy + CondWater$$

From the fitted model, we extracted the following comparisons: wet soil legacy with watered conditioning (versus wet soil legacy (baseline), wet soil legacy with drought conditioning versus wet soil legacy (baseline), dry soil legacy (baseline) versus wet soil legacy (baseline), dry soil legacy with watered conditioning versus wet soil legacy (baseline), and dry soil legacy with drought conditioning versus wet soil legacy (baseline). Taxa were considered differentially abundant if the FDR-adjusted $P$ value ($q$ value) was <0.05. Results from the GLM analysis were visualized in heat maps, where colours represent $\log_2$-transformed fold changes. Comparisons showing significant differences ($q < 0.05$ and $\log_2$-transformed fold change >2) were highlighted with black squares.

In addition, high-quality filtered reads of the transcriptome were de novo assembled into a reference metatranscriptome using Trinity v.2.15.1 (ref. 73) with default parameters. Open reading frames in transcripts were predicted with TransDecoder v.5.7.1 (ref. 74) with default settings. Functional annotation of the predicted proteins was performed using the eggNOG-mapper v.2.1.9 (ref. 55) pipeline, using the eggNOG v.5.0.2 database[56] with Diamond v.2.0.11 (ref. 57) and MMseqs2 (ref. 58). The CAT v.8.22 taxonomic classification pipeline[53] was used to assign taxonomy to the predicted protein-coding transcripts following de novo assembly and functional annotation (as opposed to assigning taxonomy to the raw reads, as was done in 'Taxonomic classification of transcripts'). These taxonomic assignments were used in downstream analyses involving gene expression and functional enrichment. Sequence reads were further filtered using SortMeRNA v.4.3.6 (ref. 75) with the smr_v4.3_default_db.fasta database to remove residual amplified rRNA sequences. Transcript quantification analysis was performed using Salmon v.1.10.0 in the mapping-based mode with the de novo assembled reference metatranscriptome. Subsequently, the transcript-level abundance estimates from salmon were extracted for the identified transcripts using the R package tximport v.1.28.044 as raw counts in default setting[76]. DESeq2 v.1.40.0 (ref. 48) was used to determine the bacterial transcript enrichment profiles by fitting a GLM as described earlier. Genes meeting the criteria of an FDR-adjusted $P$ value ($q$ value) <0.05, a $\log_2$-transformed fold change >2 and had GO classifications were subjected to GO enrichment analysis with the GO_MWU tool[59].

*Analysis of genetic variation among bacterial lineages.* Filtered shotgun metagenomic reads were aligned to the reference genomes of 33 selected taxa, including 22 identified bacterial markers and 11 additional abundant and prevalent species (see 'Analysis of genetic variation among bacterial lineages along the precipitation gradient'), using Minimap2 v.2.17-r941 (ref. 51). The resulting alignments were sorted and indexed with SAMtools v.1.18 (ref. 60). Variant calling was performed using BCFtools v.1.18 (ref. 61), and variants were filtered with VCFtools v.0.1.16 (ref. 62). Filtering criteria included a variant quality score >20, a minor allele frequency >0.01, <50% missing data and a minimum sequencing depth of 10× in each sample. After filtering, a total of 8,293 high-quality biallelic SNPs were available for further analysis. To assess genetic variation within and between groups, we analysed molecular variance using poppr v.2.9.6 (ref. 77). The significance of the analysis of molecular variance results was determined with a permutation test, using the randtest function in ade4 v.1.7.22 (ref. 78). To determine the extent of genetic differentiation between groups, we calculated the fixation index values using the hierfstat v.0.5.11 (ref. 79) package. Principal coordinate analysis plots were generated as previously described in 'Analysis of genetic variation among bacterial lineages along the precipitation gradient' and coloured to reflect soil legacy,

drought or well-watered, and host or no-host treatments. To identify genes associated with soil legacy, we conducted a genome-wide association study. SNPs were re-filtered using VCFtools v.0.1.16 (ref. 62), applying the same criteria but allowing for up to 70% missing data and a minimum sequencing depth of 3× in each sample. The genome-wide association study was conducted using a general linear model in the rMVP v.1.1.1 package[64]. Associations were identified by comparing bacterial lineages from dry legacy soil with those from wet legacy soil. Genetic structure was accounted for by incorporating the first ten PCs. Significant associations were identified through permutation testing within the rMVP package, and Manhattan plots were generated using CMplot v.4.5.1 (ref. 65).

## Test-phase soil microbiota legacy effects on plant tolerance to drought

**Experimental design and non-destructive phenotypic measurements.** At the end of the conditioning phase, homogenized soil was collected from each pot by discarding the top 1 cm of soil, mixing the soil in the pot with a clean plastic spatula and placing 6 g in a sterile 50-ml conical tube. For rhizosphere samples (that is, planted pots), plants were gently pulled from the pots and the soil particles adhered to and within the root bundle were shaken into a sterile 50-ml conical tube. The rhizosphere particles were homogenized with a clean plastic spatula and particles were poured out of the tube until 6 g remained. Soil and rhizosphere samples were stored at 4 °C overnight.

Soil microorganisms were extracted from the 6 g soil or rhizosphere sample the following day by adding 25 ml of autoclaved 1× PBS with 0.0001% Tween-89 to the 50 ml tube containing the sample. Tubes were vigorously shaken to mix and break up large soil aggregates. Large particles were allowed to settle to the bottom of the tube; the supernatant was then filtered through autoclaved Miracloth (Sigma-Aldrich) into a new sterile 50-ml conical tube. Filtered samples were then centrifuged at 3,600g for 25 min at 4 °C. The supernatant was discarded and the microbial pellet was resuspended in 6 ml of 1× PBS buffer using a vortex. The resuspended pellets were stored at 4 °C until being used for inoculations a few hours later.

As stated in 'Experimental design', the conditioning phase had 24 treatment groups with 8 replicates of each treatment (192 pots total). For the test phase, microbial extracts from all 8 replicates of each group (plus 8 sterile buffer-only mock inocula) were inoculated into a pot planted with gamagrass ($N$ = 200) and a pot planted with maize ($N$ = 200) that were then maintained under water-stress (drought) conditions. Furthermore, 4 of the 8 replicate extracted microbial inoculants (and 4 sterile buffer-only control inoculums) were each inoculated into additional gamagrass-planted ($N$ = 100) and maize-planted ($N$ = 100) pots, which were then maintained under well-watered control conditions. This makes for a total of $N$ = 600 plants at the start of the test phase. Throughout the experiment, nine water-stressed maize and five well-watered maize were lost (no gamagrass died). Therefore, phenotype measurements were completed on a total of $N$ = 586 plants. We chose this design because resource and space limitations prevented us from testing all 192 inocula under both drought and control conditions, and we were primarily interested in microbial effects on plant function under drought; we therefore opted to maximize our power to test for differences among the inocula under water limitations.

To create the inoculum, the resuspended pellet was inverted 3× to mix and 1 ml of the sample was added to 100 ml of sterile 0.5× MS liquid medium, for a microbial titre equivalent to 0.01 g soil per ml. The 'mock inoculation' controls were created by substituting 1 ml of sterile PBS for the resuspended microbial pellet. Finally, 25 ml of this suspension was inoculated onto the soil surface of each test-phase pot. Thus, each microbial community extracted from 1 of the 192 conditioning-phase pots was used to inoculate either 2 or 4 plants in the test phase. To maintain statistical independence of the experimental replicates from the conditioning phase, no pooling was performed.

Before inoculation, pots were planted with 3–4-day-old gamagrass or maize germinants. Gamagrass seeds were soaked in 3% hydrogen peroxide for 24 h and germinated in seed trays filled with sterile clay. Maize seeds were soaked in 70% EtOH for 3 min, soaked in 5% NaClO on a rotator for 2 min, and then rinsed with sterile deionized water 3×. Treated maize seeds were germinated on sterile damp paper towels inside sealed plastic bags.

Pots were fully randomized and, before inoculation, filled with a homogenized 5:1 (w/w) mixture of all-purpose sand (TechMix All-Purpose 110241) and calcined clay (Pro's Choice Rapid Dry) that had been sterilized by autoclaving on a 1 h liquid cycle. Pots were autoclaved on a 30 min liquid cycle and then filled with the sand–clay mixture, leaving 1 in. of room at the top of the pot. To help keep the mixture from falling out of the drain holes in the bottom of the pots, a sterile filter paper was shaped into a cone, pushed to the bottom of the pot, and a sterile marble was used to weigh the paper down. This effectively blocked the substrate, but still allowed water to exit the drainage holes. Plants were grown under 12-h days, with a daytime temperature at 27 °C and night-time temperature at 23 °C, and ambient humidity, with the light setting set to 1, which is equivalent to 312 μmol m$^{-2}$ s$^{-1}$. Three-day-old gamagrass and maize leaf photosynthetic rates and gas exchange were measured using the LI-6800 (LI-COR) over 3 days for each host. The LI-6800 aperture was 2 cm, warm-up tests were performed at the start of each measurement session, the fluorometry was set to 'on' and the leaf vapor pressure difference (VPD_Leaf) set to 1.5 kPa. The newest fully emerged leaf, which was most commonly the fourth leaf, on each plant was clamped in the chamber and allowed to stabilize until all measurements were stable for at least 30 s before the measurements were recorded, which took approximately 5 min per leaf. Similarly, the leaf chlorophyll content was also measured using the MC-100 Chlorophyll Concentration Meter (Apogee Instruments).

Maize plants were sampled 4 weeks after planting and gamagrass was sampled 5 weeks after planting. In total, we measured 300 gamagrass plants (200 droughted and 100 well-watered plants) and 286 maize plants (191 droughted and 95 well-watered plants).

At the end of the test phase, uprooted plants were gently shaken to remove the soil attached to roots, before the collection of phenotypic, transcriptomic and microbiota data as described later. One crown root was cut off with a ceramic blade and placed in a 1.7-ml tube on dry ice for downstream DNA extraction for *16S rRNA* gene sequencing. Another crown root (0.15–0.2 g) was cut off with a ceramic blade, placed in a 1.7-ml tube, and flash frozen in liquid nitrogen for downstream RNA extraction. In between plants, the ceramic blade, plastic tweezers, plastic cutting board and gloves were cleaned with 30% bleach. All samples were held on dry ice and then transferred to a −80 °C freezer for storage. Next, the root and shoot were separated with a ceramic blade. A length of 3 cm of crown root beginning ~2 cm from the base of the shoot was cut with a ceramic blade and submerged in 50% EtOH for LAT analysis. The rest of the root system was submerged in 70% EtOH in a 50-ml centrifuge tube for downstream root architecture scanning. Shoot height and number of leaves were recorded. Shoots were placed in individual paper bags, dried in an oven at 225 °F for 12 h and the shoot dry weight was recorded.

**Root system architecture and root biomass analyses.** A Perfection V600 flatbed scanner (Epson) was used to scan intact maize and gamagrass root systems collected from test-phase plants. The scanner was set to professional mode, reflective, document, black and white mode, and 600 dpi, with a threshold of 55. A clear plastic tray filled with clean water was placed on the scanning bed. Each root system was placed in the tray with water and the tangled roots were gently pulled apart using plastic tweezers until they were no longer overlapping. A small amount of fine fibrous roots that fell off during this process was pushed to the corner of the tray and not included in the root cluster scan. The scanned root images were then analysed using Rhizovision Explorer software

v.2.0.3 (ref. 80) in 'whole-root' mode and converted to 600 dpi. The region-of-interest tool was used to outline the main root bundle before pressing play to collect trait measurements. Finally, after collection of root system architecture data, the roots were dried in an oven at 225 °F for 12 h and then weighed.

**Root LAT analysis.** We collected 1 crown root from each plant (each 3 cm long, beginning 2 cm from the base of the plant) and stored them in 50% EtOH at 4 °C. These samples were shipped to the University of Nottingham for LAT analysis. Briefly, root segments were dried with an automated CPD (Leica EM CPD 300; Leica Microsystem). Then, samples were ablated by a laser beam (Avia 7000, 355 nm pulsed laser) to vaporize the root tissue at the camera focal plane ahead of an imaging stage, and cross-sectional images were taken using a Canon T3i camera with a 5× micro-lens (MP-E 65 mm) on the laser-illuminated surface. ImageJ software[47] was used to measure root anatomical traits captured in the high-quality LAT images.

**Leaf ionome and shoot biomass analyses.** The elemental profiles of the shoots were measured using ICP-MS. The shoot biomass from all uprooted plants was dried in an oven at 225 °F for 12 h and then weighed. The dried biomass samples were cut into small pieces using a clean ceramic scalpel and placed in 5-ml Eppendorf tubes with 3 zirconium oxide beads. Shoots were pulverized using a Tissue Lyzer II (Qiagen) using 2 cycles of 60 s at a frequency of 30 s$^{-1}$. Next, 5–10 mg of pulverized shoot samples were weighed on a Mettler 5-decimal analytical scale, and 1–3 ml (depending on the sample dry weight) of concentrated trace metal grade nitric acid Primar Plus (Fisher Chemicals) was added to each tube. Before the digestion, 20 µg l$^{-1}$ of indium was added to the nitric acid as an internal standard to assess putative errors in dilution or variations in sample introduction and plasma stability in the ICP-MS instrument. The samples were then digested in DigiPREP MS dry block heaters (SCP Science; QMX Laboratories) for 4 h at 115 °C. After cooling down, the digested samples were diluted to 10–30 ml (depending on the volume of the nitric acid added) with 18.2 MΩcm Milli-Q Direct Water. The elemental analysis was performed using an ICP-MS, PerkinElmer NexION 2000 equipped with Elemental Scientific 4DXX FAST Dual Rinse autosampler, FAST valve and peristaltic pump. The instrument was fitted with a PFA-ST3 MicroFlow nebulizer, baffled cyclonic C3 high-sensitivity glass spray chamber cooled to 2 °C with a PC3X Peltier heated–cooled inlet system, 2.0 mm inner diameter quartz injector torch and a set of nickel cones. A total of 24 elements were monitored, including the following stable isotopes: $^{7}$Li, $^{11}$B, $^{23}$Na, $^{24}$Mg, $^{31}$P, $^{34}$S, $^{39}$K, $^{43}$Ca, $^{48}$Ti, $^{52}$Cr, $^{55}$Mn, $^{56}$Fe, $^{59}$Co, $^{60}$Ni, $^{63}$Cu, $^{66}$Zn, $^{75}$As, $^{82}$Se, $^{85}$Rb, $^{88}$Sr, $^{98}$Mo, $^{111}$Cd, $^{208}$Pb and $^{115}$In. Helium was used as a collision gas in KED mode at a flow rate of 4.5 ml min$^{-1}$ while measuring Na, Mg, P, S, K, Ca, Ti, Cr, Mn, Fe, Ni, Cu, Zn, As, Se and Pb to exclude possible polyatomic interferences.

The remaining elements were measured in standard mode. The instrument's Syngistix software for ICP-MS v.2.3 (Perkin Elmer) automatically corrected any isobaric interferences. The ICP-MS measurements were performed in peak hopping scan mode with dwell times ranging from 25 ms to 50 ms, depending on the element, 20 sweeps per reading and 3 replicates. The ICP-MS conditions were adjusted to a radio-frequency power of 1,600 W and an auxiliary gas flow rate of 1.20 l min$^{-1}$. Torch alignment, nebulizer gas flow and quadrupole ion deflector voltages (in standard and KED mode) were optimized before analysis for highest intensities and lowest interferences (oxides and doubly charged ions levels lower than 2.5%) with NexION Setup Solution containing 1 µg l$^{-1}$ of Be, Ce, Fe, ln, Li, Mg, Pb and U in 1% nitric acid using a standard built-in software procedure. To correct for variation between and within ICP-MS analysis runs, liquid reference material was prepared using pooled digested samples and run after the instrument calibration and then after every nine samples in all ICP-MS sample sets. Equipment calibration was performed at the beginning of each analytical run using 7 multi-element calibration standards (containing 2 µg l$^{-1}$ indium internal standard) prepared by diluting 1,000 mg l$^{-1}$ Single Element Standard solutions (Inorganic Ventures; Essex Scientific Laboratory Supplies) with 10% nitric acid. As a calibration blank, 10% nitric acid containing 2 µg l$^{-1}$ indium internal standard was used, and it was run throughout the analysis. Sample concentrations were calculated using the external calibration method within the instrument software. Further data processing, including the calculation of final element concentrations, was performed in Microsoft Excel.

**Crown root transcriptomics.** *RNA extraction and sequencing.* For RNA extractions and sequencing, flash-frozen crown roots were freeze dried for 48 h and finely ground with pellet pestles. The RNA extraction protocol was carried out according to the NucleoSpin RNA Plant kit (Macherey-Nagel).

For the 132 maize samples, remnant DNA was removed from purified RNA using the DNA-Free kit (Invitrogen). RNA-seq libraries were prepared using the QuantSeq 3′ mRNA-Seq V2 kit with unique dual sequences and the unique molecular identifier (UMI) module (Lexogen) following the manufacturer's recommendations. Libraries were pooled at equimolar concentrations and then sequenced (2 × 150 bp, but reverse reads were not used) on a NovaSeq S4 flow cell (Illumina) along with a 25% PhiX spike-in. Maize RNA-seq library preparations and sequencing were performed by the RTSF Genomics Core at Michigan State University. For the 132 gamagrass samples, RNA-seq libraries were prepared using the NEBNext Ultra II Directional Library Kit with the oligo(dT) magnetic isolation module (New England Biolabs) and sequenced on the Illumina NovaSeq 6000 platform at the Genomic Sciences Laboratory at North Carolina State University to generate a minimum of 40M read pairs (2 × 150 bp) per sample.

*Sequence processing.* For the maize sequence reads, UMIs were removed from all sequences and added to the read headers using UMI-tools[81]. Next, cutadapt v.4.2 (ref. 69) was used to remove the first four bases of each read, remove poly(A) tails (if present), remove spurious poly(G) runs using the --nextseq-trim=10 parameter and remove adaptor sequences. Reads that were <10 bp long or that aligned to maize rRNA gene sequences were removed; the remaining reads were aligned to the maize reference genome B73 RefGen_v5 (ref. 82) using HISAT2 v.2.2.1 (ref. 83). Aligned reads were converted to BAM format, sorted and indexed using SAMtools v.1.9 (ref. 60). We then used UMI-tools[81] to de-duplicate reads that both shared a UMI and had identical mapping coordinates. Finally, we used the FeatureCounts function of the subread package v.2.0.5 (ref. 84) with the maize genome annotation version Zm00001eb.1 and parameters -O --fraction -M --primary -g ID -t gene to generate a table of transcript counts.

For the gamagrass sequence reads, we first used cutadapt to remove NEBNext adaptor sequences, poly(A) tails, spurious poly(G) runs and low-quality tails using the -q 20,20 parameter and other default parameters. The cleaned reads were aligned to the gamagrass reference genome (Td-KS_B6_1-REFERENCE-PanAnd-2 .0a$^{60}$) using HISAT2. The alignments were name-sorted so that mate pairs could be fixed using the fixmate function of SAMtools[60], then re-sorted based on coordinates, de-duplicated and converted to indexed BAM files using the same software. Finally, a table of transcript expression estimates was generated using the FeatureCounts function of the subread package with parameters -p -O --fraction -M --primary -g ID -t gene and the gamagrass genome annotation version Td-KS_B6_1-REFERENCE-PanAnd-2.0a_Td00002ba.2.

*Statistical analyses.* As the maize and gamagrass RNA-seq datasets were generated using different approaches (3′ tag sequencing versus full-length sequencing, respectively) we analysed them in parallel rather than comparing them directly. For each species, we used DESeq2 (ref. 48) to identify genes that were differentially expressed

between plants inoculated with microbiomes from a low-precipitation climate (dry legacy) versus those inoculated with microbiomes from a high-precipitation climate (wet legacy). A single negative binomial model with default parameters was used to estimate $\log_2$-transformed fold changes in gene expression due to inoculum legacy, while also controlling for the other experimental factors, using the model:

$$Counts \sim legacy + CondWater + CondHost + TestWater$$

where TestWater is our shorthand for the variable 'test-phase watering treatment', a factor with two levels (drought or control). Statistical support was obtained using the Wald test with Benjamini–Hochberg FDR correction. All available samples were included in each analysis; thus, these results should be interpreted as the gene expression response to microbiome precipitation legacy, averaged across all levels of the other experimental factors.

In addition, we investigated whether plants' gene expression responses to limited versus ample water during the test phase (that is, the TestWater variable) were affected by inoculum precipitation legacy. To do so, we used DESeq2 to fit a model with the formula:

$$Counts \sim TestWater * legacy$$

Then, we extracted the estimated $\log_2$-transformed fold changes due to acute drought (relative to well-watered conditions) for both dry-legacy-inoculated and wet-legacy-inoculated plants. We inferred a meaningful interaction between these variables when the 95% confidence intervals of the two drought-induced $\log_2$-transformed fold changes did not overlap at all. These interacting genes are candidates for linking real-time plant drought response to the microbiome's historical environmental conditions.

Finally, we conducted a mediation analysis to determine whether the gamagrass genes that were sensitive to inoculum legacy were implicated in phenotypic responses to subsequent acute drought. A gene was considered legacy sensitive if its expression was significantly affected by the main effect of inoculum legacy, or if it was affected by the interaction between inoculum legacy and test-phase drought treatment, as described earlier. We summarized expression patterns of this subset of genes (normalized as transcripts per million, calculated using the full set of expressed genes) using non-metric multidimensional scaling of the Bray–Curtis distances among all gamagrass plants, which resulted in two axes of variation: MDS1 and MDS2. Next, we used the mediation package in R[85] to compare the direct effects of test-phase drought treatment on each focal plant trait (see 'Plant trait feature selection') to the indirect effects of the drought treatment mediated through MDS1 and MDS2. Each mediation analysis used the linear models:

$$Trait \ value \sim MDS + TestWater$$

$$MDS \sim TestWater$$

where MDS represents the 'site score' of each individual plant on either MDS1 or MDS2. Separate models were fit to test the potential roles of MDS1 and MDS2 as mediator variables; however, a follow-up analysis using both MDS1 and MDS2 as simultaneous mediators, implemented in lavaan[86], yielded equivalent conclusions.

*Gene annotation*. We downloaded maize and gamagrass genome assemblies and annotations from MaizeGDB[82,87]. Functional information for maize genes was taken from the *Z. mays* genome annotation version Zm-B73-REFERENCE-NAM-5.0_Zm00001eb.1 and accessed using MaizeGDB's MaizeMine tool[82,87–89]. For gamagrass genes, we relied on DNA sequence homology with annotated maize genes to infer function. We used BEDtools[90] to extract gene coordinates and

protein-coding sequences from the gamagrass reference genome Td-KS_B6_1-REFERENCE-PanAnd-2.0a[91], and then used OrthoFinder[92] to compare coding sequences from the two species. OrthoFinder identified 32,785 gamagrass genes (71.5% of the total) as homologues of maize genes, grouping them into 21,658 distinct orthogroups. All differential gene expression analyses, however, considered the entire set of expressed genes; those without maize orthologues, or with unannotated maize orthologues, were considered to be of unknown function.

**16S rRNA amplicon sequencing.** *DNA extractions and library preparation*. One crown root from each plant was cut off with a ceramic blade and placed in a 1.7-ml tube and flash frozen in liquid nitrogen for downstream amplicon library preparation. After collection, root subsamples were kept on dry ice and ceramic tweezers were used to transfer the whole root to 1.1-ml cluster tubes (USA Scientific). Tweezers were sterilized with 80% EtOH between samples. Roots were selected at random for placement in cluster tubes and were stored at −80 °C until DNA extraction, at which time roots were freeze dried for 48 h in a FreeZone lyophilizer (Labconco). After freeze drying, roots were flash frozen in liquid nitrogen. To break apart thick roots, sterile forceps and a dissecting needle were used before placing the rack of cluster tubes in an HT Lysing Homogenizer (OHAUS) with 2 clean 5/32 in. steel balls in each tube. Samples were homogenized at 25 Hz for 1 min. Root material was then transferred to 2 ml bead-beating 96-well plates containing sterile 1-mm garnet beads with 850 µl of lysis buffer (1 M Tris pH 8.0, 100 mM NaCl and 10 mM EDTA).

A positive (ZymoBiomics Microbial Community Standard) and negative control (800 µl of lysis buffer) were included on each plate. Bead-beating plates were stored at −20 °C until extraction. After thawing, 10 µl of 20% sodium dodecyl sulfate was added to each well before homogenizing for a total of 20 min at 20 Hz. Plates were incubated in a water bath (55 °C for 90 min) and centrifuged (6 min at 4,500$g$). A 400 µl volume of the resulting supernatant was transferred to new 1-ml 96-well plates containing 120 µl of 5 M potassium acetate in each well and incubated overnight at −20 °C. After thawing the plates, they were centrifuged (6 min at 4,500$g$) and 400 µl of the supernatant was transferred to a new 1-ml 96-well plate containing 600 µl of diluted SPRI-bead solution (protocol derived from ref. 93). These plates were mixed thoroughly for 5 min at 1,000 rpm on an orbital plate shaker. Samples were allowed to incubate for 10 min so DNA could bind to beads, after which the plate was centrifuged (6 min at 4,500$g$) and placed on a magnet rack for 10 min. The supernatant was removed, and the beads were washed 2× with 900 µl of 80% EtOH. After washing, the supernatant was decanted and the beads were air dried. DNA was eluted in 75 µl of pre-heated 1× TE (pH 7.5, 37 °C) and transferred to clean 0.45-ml plates and stored at −20 °C.

16S V4 rRNA gene amplification was performed using the 515F/806R gene primers[94,95] with overhangs for annealing to universal Illumina adaptors (515F, TCGTCGGCAGCGTCAGATGTGTATAAGAGA-CAGNNNNGTGYCAGCMGCCGCGGTAA; 806R, GTCTCGTGGGCTCG-GAGATGTGTATAAGAGACAGNNNNGGACTACNVGGGTWTCTAAT). PCR reactions contained DreamTaq Master Mix (Thermo Fisher Scientific), 10 mg µl⁻¹ bovine serum albumin (BSA), 100 µM peptide nucleic acid (PNA) and PCR-grade water. BSA was used to enhance PCR amplification and PNA was used to suppress primer binding and subsequent amplification of mitochondrial and chloroplast 16S regions[96]. The PCR included an initial denaturation step at 95 °C for 2 min, followed by 27 cycles of an additional denaturation at 95 °C for 20 s, PNA annealing at 78 °C for 5 s, primer annealing at 52 °C for 20 s, and extension at 72 °C for 50 s. This was followed by a final extension step for 10 min at 72 °C. PCR products were purified by incubating at 37 °C for 20 min and then for 15 min at 80 °C after mixing with 0.78 µl of PCR-grade water, 0.02 µl of 10 U µl⁻¹ exonuclease I (Applied Biosystems) and 0.2 µl of 1 U µl⁻¹ shrimp alkaline phosphatase (Applied Biosystems) per 10 µl PCR product. A volume of 2 µl of the purified amplicons was used as

template DNA in an indexing PCR to attach barcoded P5 and P7 Illumina adaptors. The 10-μl reaction included 5 μM each of P5 and P7 adaptors, 1× DreamTaq Master Mix (Thermo Fisher Scientific), 10 mg ml$^{-1}$ BSA, 100 μM PNA and PCR-grade water. An initial denaturation step was performed at 95 °C for 2 min. For 8 cycles, an additional denaturation was carried out at 95 °C for 20 s, PNA annealing at 78 °C for 5 s, primer annealing at 52 °C for 20 s and extension at 72 °C for 50 s. A final extension step was performed at 72 °C for 10 min. PCR products were verified via 2% agarose gel electrophoresis and then pooled by 96-well plate using the Just-a-Plate PCR clean-up and normalization kit (Charm Biotech). Pools were size selected and combined in equimolar concentrations before being sequenced on an Illumina SP flow cell on the NovaSeq 6000 platform (2 × 250 bp reads) by the University of Kansas Medical Center Genome Sequencing Core facility.

*Amplicon data processing.* We removed primer sequences from raw 16S V4 Illumina reads using cutadapt[69], requiring at least five nucleotides of overlap. Additional quality control and processing was performed with the DADA2 software[97]. Forward reads were discarded if they had more than 6 expected errors, otherwise they were truncated at 200 nucleotides; for reverse reads the parameters were 7 expected errors and 170 nucleotides. Error rates were estimated separately for forward and reverse reads based on a sample of $1 \times 10^8$ bases, and then used to denoise and de-replicate reads using the standard DADA2 functions. Chimeric sequences were detected and removed using the 'consensus' procedure in DADA2. Each individual sample was processed in parallel, after which all of the resulting ASV tables were merged. Finally, taxonomy of each ASV was assigned by comparison to the RDP database[98].

*Root bacterial community diversity analysis.* All statistical analyses were performed using R (v.4.4.0). Unfortunately, some samples were lost due to a 96-well plate being dropped during the sample DNA extraction and due to filtering of samples based on sequence quality. Ultimately, 156 gamagrass root microbiome and 276 maize root microbiome samples were included in these analyses. The cleaned and prepared microbiome data, as phyloseq objects, were loaded into R[67]. The phyloseq object sample data table was replaced with an updated metadata file. The mock samples were removed from the dataset and the data were subset by the test-phase host, maize and gamagrass using phyloseq::subset_samples (v.1.48.0). Maize and gamagrass root bacterial microbiome Shannon diversity ($e^{Shannon\ index}$) was fit to a mixed-effects model using the lme4 (v.1.1-35.4) package[99] and lmer() function, using the following model:

$$\exp Shannon \sim TestWater + legacy + (1|DNAplate)$$

where the random effect DNAplate represents the 96-well plate that samples were randomly assigned to for DNA extraction and sequencing library preparation, which helps control for batch effects. A type III ANOVA was performed using stats::anova()[46], to test for significance, followed by pairwise comparisons and significance tests using the emmeans (v.1.10.2) package[100] with FDR adjustments. Data visualizations were generated using ggplot2 (v.3.5.1)[44], the error bars represent standard error and the points are estimated marginal means (EMMs). All plots were saved in PDF format. Composite figures were created in Adobe Illustrator.

Bacterial community β-diversity was accessed with a constrained analysis of principal coordinates, using phyloseq::ordinate, method = 'CAP', and distance = 'Euclidean' with the following formula:

$$\sim TestWater + legacy + Condition(DNAplate + CondHost + CondWater)$$

An ANOVA-like permutation test was used to test for model significance using the vegan::anova.cca() function[45]. The anova.cca by = 'term' option was used to determine the significance of the

TestWater and legacy variables separately. CAP1 and CAP2 axes were plotted using ggplot2, including 95% confidence interval ellipse (stat_ellipse()). To evaluate treatment group β-dispersion, Euclidean distances were calculated for the centred-log-ratio-transformed counts using phyloseq::distance(). Both legacy and TestWater terms were tested for β-dispersion differences using the Euclidean distances matrix and vegan::betadisper() and vegan::permutest() to test for significance[45]. Distances were then extracted from the results output and fit to a linear model. EMMs were calculated and plotted using the same methods as mentioned earlier. The error bars represent standard errors.

Then, an ASV differential abundance analysis was performed on gamagrass- and maize-associated ASVs separately. Centred-log-ratio-transformed counts of ASVs were each fit to a linear model using an iterative for loop. Each ASV was subset using phyloseq:subset_taxa and the phyloseq::psmelt() function was used to reformat the phyloseq object into a data frame. Then the ASV abundance was fit to the following model:

$$Abundance \sim legacy * TestWater$$

Then, stats::anova() was used to test for significance, followed by an FDR *P* value adjustment using stats::p.adjust(). ASVs with significant (≤0.05) adjusted *P* values for the legacy*TestWater interaction terms were extracted and *z*-scores of the centre-log-ratio-transformed counts were plotted in a heat map using pheatmap::pheatmap()[101] with clustering_method = 'complete'.

**Plant trait data analyses.** *Plant trait feature selection.* All plant data were loaded into RStudio. The iWUE was calculated using the ratio of photosynthetic rate (A) and stomatal conductance (gs) or A gs$^{-1}$ ((μmol m$^{-2}$ s$^{-1}$) (mol m$^{-2}$ s$^{-1}$)$^{-1}$). Data were subset by test-phase host treatment groups (gamagrass and maize). For the feature selection, the mock treatment was removed. All 67 plant traits for which we had reliable measurements were formatted into a data frame. Three sample rows were removed from the dataset because they had missing data. The *S*-index was calculated for each of these features using the following formula:

$$S = \frac{trait_{drought}}{mean(trait_{control})}$$

Then, for each host, a random forest model was used to select the top 10 most important features from the 67 total *S*-index measurements using randomForest()[102], with legacy as the predictor. Correlations (stats::cor()) were estimated for the top 10 traits and if any 2 of the top 10 traits were highly correlated ($r \geq 0.7$), the trait that ranked lower in the random forest model was removed. Non-correlated top traits from the TestWater treatment groups (drought versus control) were combined for each test-phase host (maize and gamagrass). Vegan:rda() was used to perform a redundancy analysis on the top traits using the following formula:

$$Top\_traits \sim legacy + Condition(CondGroup), scale = TRUE$$

where the CondGroup variable is a factor with four levels representing the factorial combinations of conditioning-phase drought and host treatments.

*Feature analyses.* An ANOVA-like permutation test was used to test for model significance using the vegan::anova.cca() function. A biplot was created using ggplot2, plotting the redundancy analysis constrained axis (RDA1) and the first unconstrained axis (PC1) with species scores (represented by arrows) and stat_ellipse was used to plot 95% confidence intervals of the site scores.

Finally, *S*-indices for each of the top traits and iWUE were fit to mixed-effects models, when possible, or a fixed-effects model if overfitting of the mixed-effects model occurred. Each feature or trait was visually assessed for outliers, which were removed. Removing the outliers did not impact the interpretation of any of the results. Formulas used for the mixed-effects or linear model, respectively, include:

$$\text{lmer}(S\text{-index} \sim \text{legacy} + (1|\text{CondGroup}))$$

$$\text{lm}(S\text{-index} \sim \text{legacy} + \text{CondGroup})$$

In these models, CondGroup is a factor with four levels representing the factorial combinations of conditioning-phase drought and host treatments. The fit of the model was assessed and, if needed, the *S*-index was transformed using sqrt(), exp() or log() to improve the fit. ANOVA was used to assess significance and EMMs were calculated and plotted as described earlier.

### Reporting summary
Further information on research design is available in the Nature Portfolio Reporting Summary linked to this article.

## Data availability
The 16S rRNA gene amplicon sequencing data, shotgun metagenomic data and metatranscriptome data associated with this study have been deposited in the NCBI Sequence Read Archive under the BioProject identifiers PRJNA1267293, PRJNA1267715, PRJNA1268489 and PRJNA1186942. The raw RNA-seq data from gamagrass and maize have been deposited in the Gene Expression Omnibus under accessions GSE282586 and GSE282587, respectively. Plant phenotype data, soil data and all other source data are freely available in a Zenodo repository at https://doi.org/10.5281/zenodo.13821006 (ref. 103).

## Code availability
All scripts and additional data structures required to reproduce the results of this study are available in a Zenodo repository at https://doi.org/10.5281/zenodo.13821006 (ref. 103).

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

## Acknowledgements

We thank A. Seetharam, T. Kellogg, M. Hufford and the PanAnd consortium for granting early access to the *T. dactyloides* genome and annotation. We thank M. Greer, M. VonLintel, J. Roemer, A. Schlegel, and The Nature Conservancy in Kansas (Smoky Valley Ranch) for permitting us to collect soil samples from these sites, and P. Balint-Kurti for providing maize seeds. We thank C. J. Sturrock and B. S. Atkinson from the Hounsfield Facility at the University of Nottingham for their assistance with the analysis of the computed-tomography images. We thank the undergraduate student researchers and technicians who helped to set up experiments and collect data: F. Tso, H. Reid and C. Rodriguez. We thank J. Swift, C. Kural-Rendon, B. Sikes (University of Kansas), M. Kleiner (North Carolina State University), O. Finkel (The Hebrew University of Jerusalem) and N. Girkin (University of Nottingham) for critical reading of the paper. This work was supported by the US National Science Foundation (numbers IOS-2016351 and DBI-2120153 to M.R.W.), the BBSRC (number BB/V011294/1 to G.C.) and the US National Institute of Food and Agriculture (number 2022-67013-36672 to M.R.W.). Research reported in this publication was made possible in part by the services of the KU Genome Sequencing Core. This laboratory is supported by the National Institute of General Medical Sciences of the National Institutes of Health (NIH) under award number P30GM145499. We thank the KU Medical Center Genomics Core for generating the 16S amplicon sequence datasets. The KUMC Genomics Core is supported by the Kansas Intellectual and Developmental Disabilities Research Center (NIH U54 HD 090216), the Molecular Regulation of Cell Development and Differentiation – COBRE (P30 GM122731), the NIH S10 High-End Instrumentation grant (NIH S10OD021743) and the Frontiers CTSA grant (UL1TR002366).

## Author contributions

Conceptualization: N.A.G., V.C., D.G., G.C. and M.R.W. Data curation: N.A.G., V.C., D.G. and M.R.W. Formal analysis: N.A.G., V.C., D.G., I.S.-G. and M.R.W. Investigation: N.A.G., V.C., D.G., N.F., D.H.J., D.M.W., Â.M., G.C. and M.R.W. Methodology: G.C. and M.R.W. Project administration: G.C. and M.R.W. Software: N.A.G., V.C., D.G. and M.R.W. Supervision: G.C. and M.R.W. Visualization: N.A.G., V.C., D.G. and M.R.W. Writing—original draft: N.A.G., G.C. and M.R.W. Writing—review and editing: V.C. and D.G.

## Competing interests

The authors declare no competing interests.

## Additional information

**Extended data** is available for this paper at https://doi.org/10.1038/s41564-025-02148-8.

**Correspondence and requests for materials** should be addressed to Gabriel Castrillo or Maggie R. Wagner.

**Extended Data Table 1 | Expression patterns of gamagrass (*Tripsacum dactyloides*) genes in relation to microbiome legacy and NMDS axes that mediate drought responses**

**Top 5% of genes with strongest positive loadings on MDS1**

| Td_gene | MDS1 | MDS2 | Gene_set | Zm_orthologs | Zm_gene_symbol |
|---|---|---|---|---|---|
| Td00002ba000754 | 0.282 | 0.182 | I | Zm00001eb057770 | |
| Td00002ba018910 | 0.246 | 0.141 | VII | Zm00001eb112460 | cl31479_3 |
| Td00002ba037703 | 0.241 | -0.057 | % | Zm00001eb237740 | |
| Td00002ba002499 | 0.217 | 0.059 | I | NA | |
| Td00002ba014047 | 0.21 | 0.021 | I | Zm00001eb381580 | |
| Td00002ba022717 | 0.208 | 0.065 | III | Zm00001eb055100 | |
| Td00002ba044330 | 0.205 | 0.103 | VI | Zm00001eb199260 | |
| Td00002ba019383 | 0.19 | 0.148 | I | Zm00001eb115970 | ms33 |
| Td00002ba011993 | 0.181 | 0.078 | I | Zm00001eb307550 | ereb177 |
| Td00002ba024147 | 0.178 | 0.057 | I | Zm00001eb248540 | umc2600a |

**Top 5% of genes with strongest negative loadings on MDS1**

| Td_gene | MDS1 | MDS2 | Gene_set | Zm_orthologs | Zm_gene_symbol |
|---|---|---|---|---|---|
| Td00002ba046162 | -0.203 | 0.116 | V | Zm00001eb199320 | |
| Td00002ba046161 | -0.206 | 0.112 | II | Zm00001eb199320 | |
| Td00002ba036367 | -0.222 | 0.138 | II | Zm00001eb199320 | |
| Td00002ba036368 | -0.224 | 0.132 | II | Zm00001eb199320 | |
| Td00002ba019546 | -0.236 | -0.222 | II | Zm00001eb147080 | |
| Td00002ba039891 | -0.238 | 0.451 | III | NA | |
| Td00002ba007195 | -0.257 | 0.113 | II | Zm00001eb161990 | |
| Td00002ba015525 | -0.259 | 0.46 | IV | Zm00001eb396120 | nas7 |
| Td00002ba015526 | -0.266 | 0.486 | IV | Zm00001eb396120 | nas7 |
| Td00002ba002239 | -0.304 | 0.188 | III | Zm00001eb048620 | |

**Top 5% of genes with strongest positive loadings on MDS2**

| Td_gene | MDS1 | MDS2 | Gene_set | Zm_orthologs | Zm_gene_symbol |
|---|---|---|---|---|---|
| Td00002ba015526 | -0.266 | 0.486 | IV | Zm00001eb396120 | nas7 |
| Td00002ba015525 | -0.259 | 0.46 | IV | Zm00001eb396120 | nas7 |
| Td00002ba039891 | -0.238 | 0.451 | III | NA | |
| Td00002ba001144 | 0.168 | 0.447 | VI | Zm00001eb053700 | |
| Td00002ba001146 | 0.16 | 0.44 | VI | Zm00001eb053700 | |
| Td00002ba041411 | -0.173 | 0.389 | IV | NA | |
| Td00002ba015516 | -0.141 | 0.272 | IV | Zm00001eb396120 | nas7 |
| Td00002ba015515 | -0.141 | 0.272 | IV | Zm00001eb396120 | nas7 |
| Td00002ba033830 | -0.15 | 0.218 | V | Zm00001eb039280 | ga2ox7 |
| Td00002ba019393 | -0.154 | 0.199 | III | Zm00001eb286420 | mads58 |

**Top 5% of genes with strongest negative loadings on MDS2**

| Td_gene | MDS1 | MDS2 | Gene_set | Zm_orthologs | Zm_gene_symbol |
|---|---|---|---|---|---|
| Td00002ba023452 | -0.061 | -0.166 | V | Zm00001eb027750 | pat3 |
| Td00002ba021901 | 0.012 | -0.19 | V | NA | |
| Td00002ba019546 | -0.236 | -0.222 | II | Zm00001eb147080 | |
| Td00002ba019177 | 0.023 | -0.222 | III | Zm00001eb180130 | |
| Td00002ba009640 | -0.077 | -0.222 | II | Zm00001eb210740 | AY110625 |
| Td00002ba008826 | -0.138 | -0.224 | II | Zm00001eb125560 | umc1742 |
| Td00002ba045645 | -0.019 | -0.24 | * | NA | |
| Td00002ba015343 | 0.124 | -0.267 | II | NA | |
| Td00002ba006731 | -0.084 | -0.272 | II | Zm00001eb210740 | AY110625 |
| Td00002ba006546 | -0.029 | -0.435 | VIII | Zm00001eb211300, Zm00001eb211310, Zm00001eb155430, Zm00001eb055690 | pco095801, AY106518, GRMZM2G024958 |

Lists of genes comprising the 5% tails of gene loadings onto both axes of an ordination based on non-metric multidimensional scaling of RNA-seq data (see Fig. 4). Detailed information about the expression responses of these genes is available in Supplementary Table S10. 'Gene_set' refers to the patterns of response to microbiome legacy presented in Fig. 4a, b. % downregulated due to main effect of dry-legacy inoculum (no interaction with drought treatment). * upregulated due to main effect of dry-legacy inoculum (no interaction with acute drought).

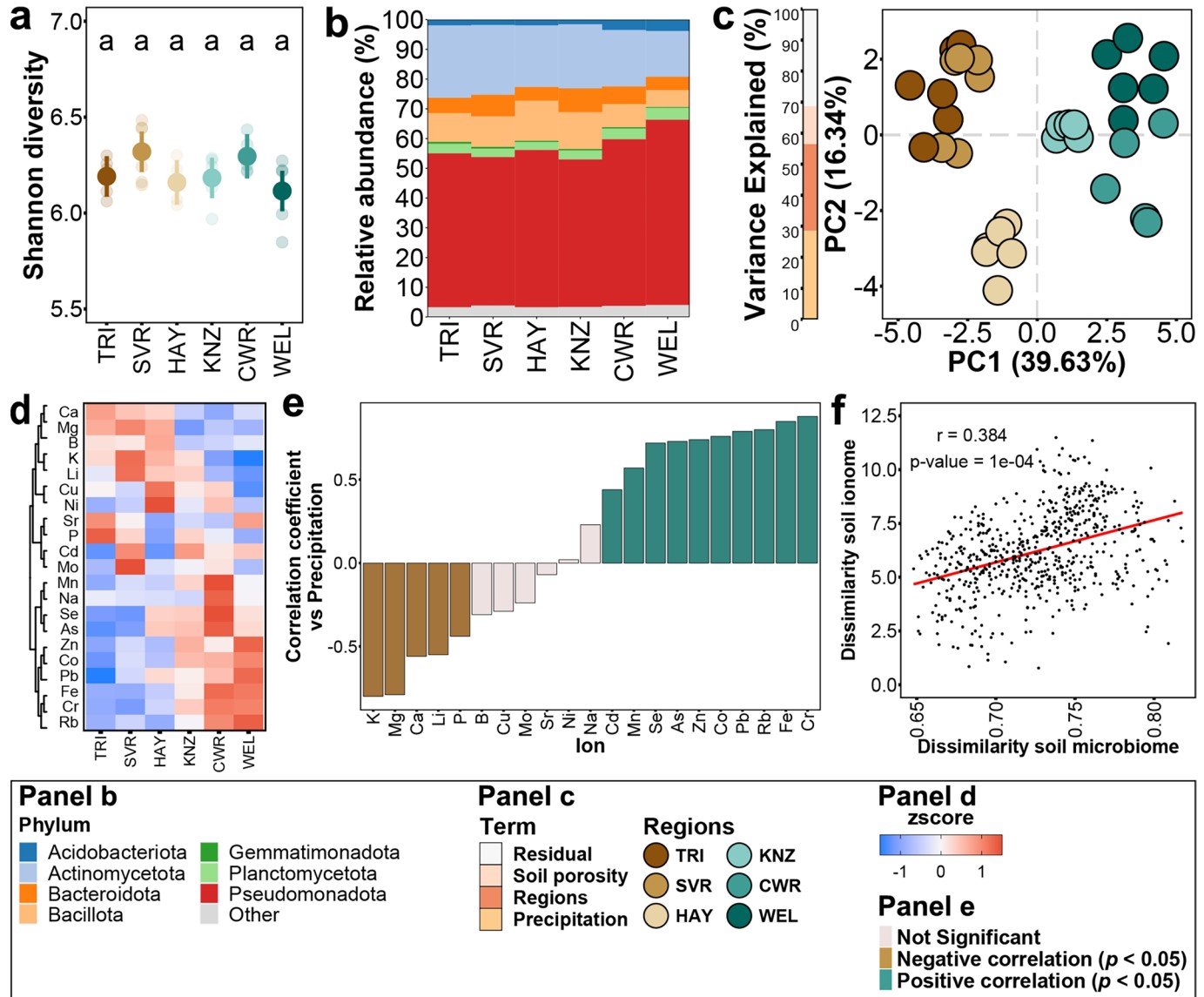

**Extended Data Fig. 1 | Precipitation level and mineral nutrient content correlate with soil microbiota composition. a**. Soil alpha diversity (estimated using the Shannon diversity index) did not differ among sampling locations. Locations are ordered by increasing precipitation (Fig. 1a) (N = 6 independent soil samples per location). Significance was determined using analysis of variance (ANOVA, $F_{5,28}$ = 2.29, p = 0.073); letters correspond to a two-tailed Tukey *post-hoc* test ($\alpha$ = 0.05) of pairwise contrasts. Points represent the estimated marginal means with 95% confidence intervals. **b**. Phylogram showing the relative abundance profiles of the main bacterial phyla across soils exposed to different precipitation levels. **c**. Principal component analysis showing the ionomic profiles of the six focal soils (N = 6 per soil). The bar on the left denotes the percentage of the variance explained by the predictor variables. **d**. Heatmap showing the standardized concentration (z-score) of each mineral nutrient (rows) in the collection of soils exposed to different precipitation levels. The values were clustered according to the ion concentrations and the regions were ordered from low to high precipitation. **e**. Bar graph showing the Pearson correlation coefficient between each mineral nutrient abundance and the level of precipitation across the collection of soils used. Coloured bars indicate statistically significant correlations (q < 0.05). **f**. Pairwise correlation analysis between soil mineral nutrient dissimilarities and soil microbiome composition dissimilarities in soils representing a gradient of precipitation (N = 6 per soil). Each point represents one pair of soil samples. Panel shows the one-tailed Mantel test *r* statistic and its *p*-value (p = 1e-4).

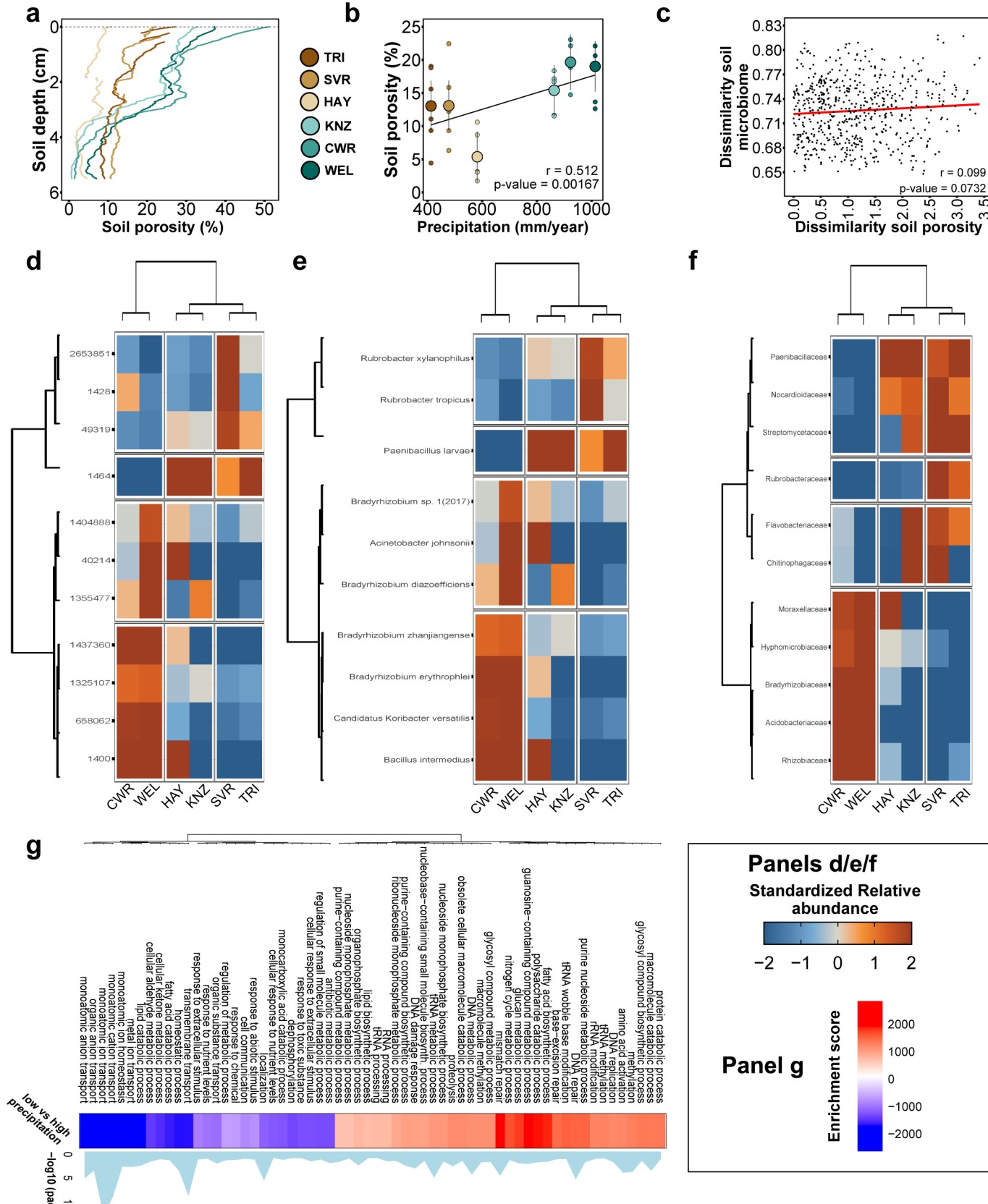

**Extended Data Fig. 2 | See next page for caption.**

**Extended Data Fig. 2 | Precipitation, but not soil porosity, explains microbiota composition across the Kansas soil collection. a**. Percent soil porosity changes with depth in soils exposed to the precipitation gradient. **b**. Two-tailed Pearson correlation analysis between soil porosity (averaged across depths) and mean annual precipitation (N = 6 soil samples per location, except N = 5 in CWR). Points represent the estimated marginal means (EMMs) with 95% confidence intervals. **c**. Pairwise correlation analysis between soil microbiota dissimilarities and soil porosity dissimilarities. The panel shows the one-sided Mantel r statistic and its p-value. **d**–**f**. Heatmaps showing changes in the relative abundances of NCBI TaxIDs (**d**), species (**e**), and families (**f**). In all cases, the values have been clustered according to taxonomic categories and soils. **g**. Numerous biological processes were enriched (red) or depleted (blue) in soils from low-precipitation sites (TRI, SVR, and HAY) relative to high-precipitation sites (WEL, CWR, and KNZ) (q < 0.05). Gene enrichment analysis was conducted using a generalized linear model, followed by Gene Ontology (GO) classification. To evaluate enrichment of each GO category, a two-sided Mann-Whitney U test was used to assess whether genes linked to the category were significantly clustered at either the top or bottom of a globally ranked gene list. p-values were adjusted for multiple comparisons, and categories with an adjusted p-value < 0.05 were considered significant. Adjusted p-values are shown on the plot as -$\log_{10}$ values. Enrichment scores are displayed as square root-transformed delta rank values of the GO categories.

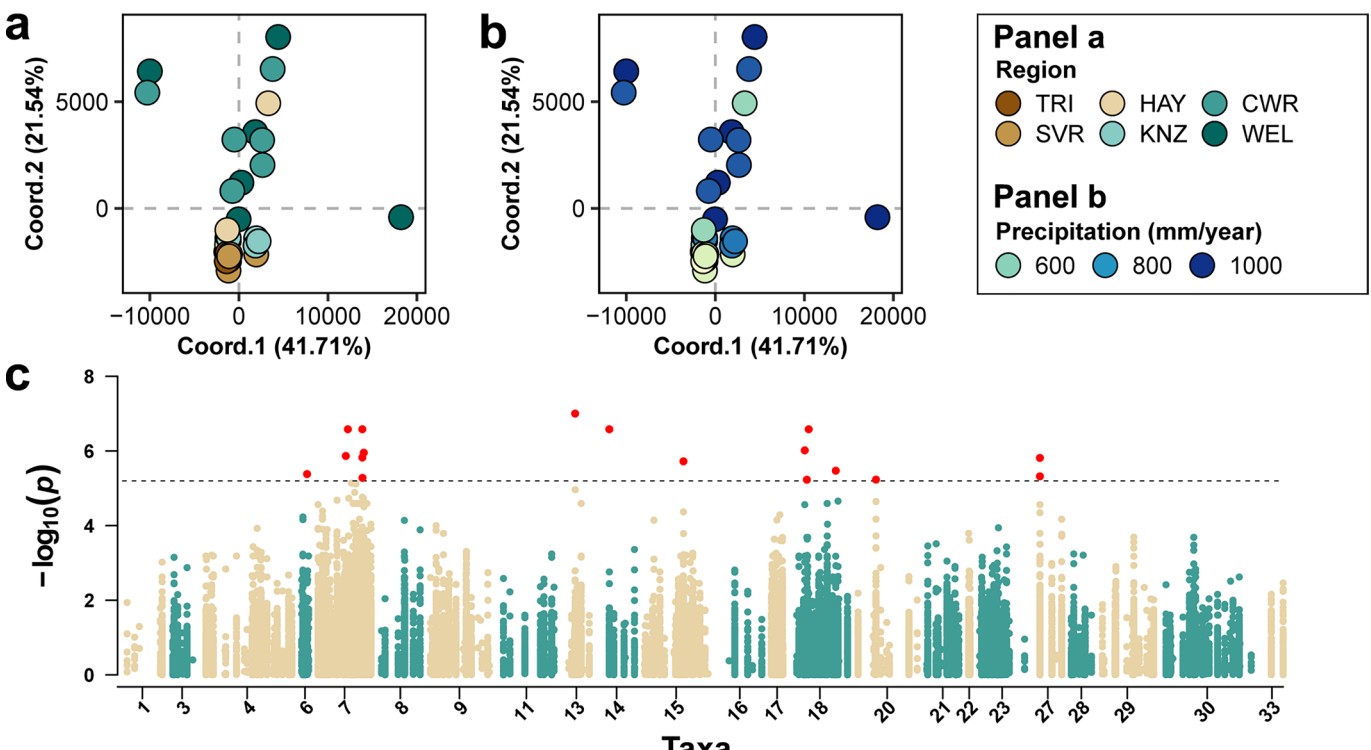

**Extended Data Fig. 3 | Precipitation legacy shapes the genetic differentiation of bacterial lineages within taxa.** To assess genetic differences among bacterial lineages along the precipitation gradient, we selected 33 bacterial species (15 of the bacterial biomarkers of precipitation, plus 18 additional abundant and prevalent taxa) that exhibited high genome coverage in our metagenomic dataset as proxies for the broader bacterial communities. Reference genomes for each species were retrieved from the NCBI Genome database, and filtered shotgun metagenomic reads were mapped to these genomes to identify high-quality biallelic single nucleotide polymorphisms (SNPs). Genetic distances between the bacterial lineages were calculated based on the identified SNPs and PCoA plots were generated and coloured by **a**. soil collection sites and **b**. mean annual precipitation. The variance explained by each axis is indicated. **c**. The Manhattan plot illustrates significant SNPs associated with precipitation, derived from the genetic-environment association (GEA) analysis. The GEA was conducted using a two-sided test of coefficients from a general linear model, with precipitation at each sampling location as the environmental variable. The significance threshold was calculated using the permutation method to account for multiple association testing. The x-axis of the plot represents the SNP positions along the genomes of the selected bacterial species, while the y-axis displays the -$\log_{10}$ p-values from the association model. The horizontal line indicates the statistical significance threshold, as determined by the permutation test. SNPs above this threshold, highlighted in red, were significantly associated with the precipitation gradient. Bacterial taxa with fewer than 1,000 high-quality biallelic SNPs after filtering, and with no significant SNPs detected from the GEA analysis, are not shown in the plot.

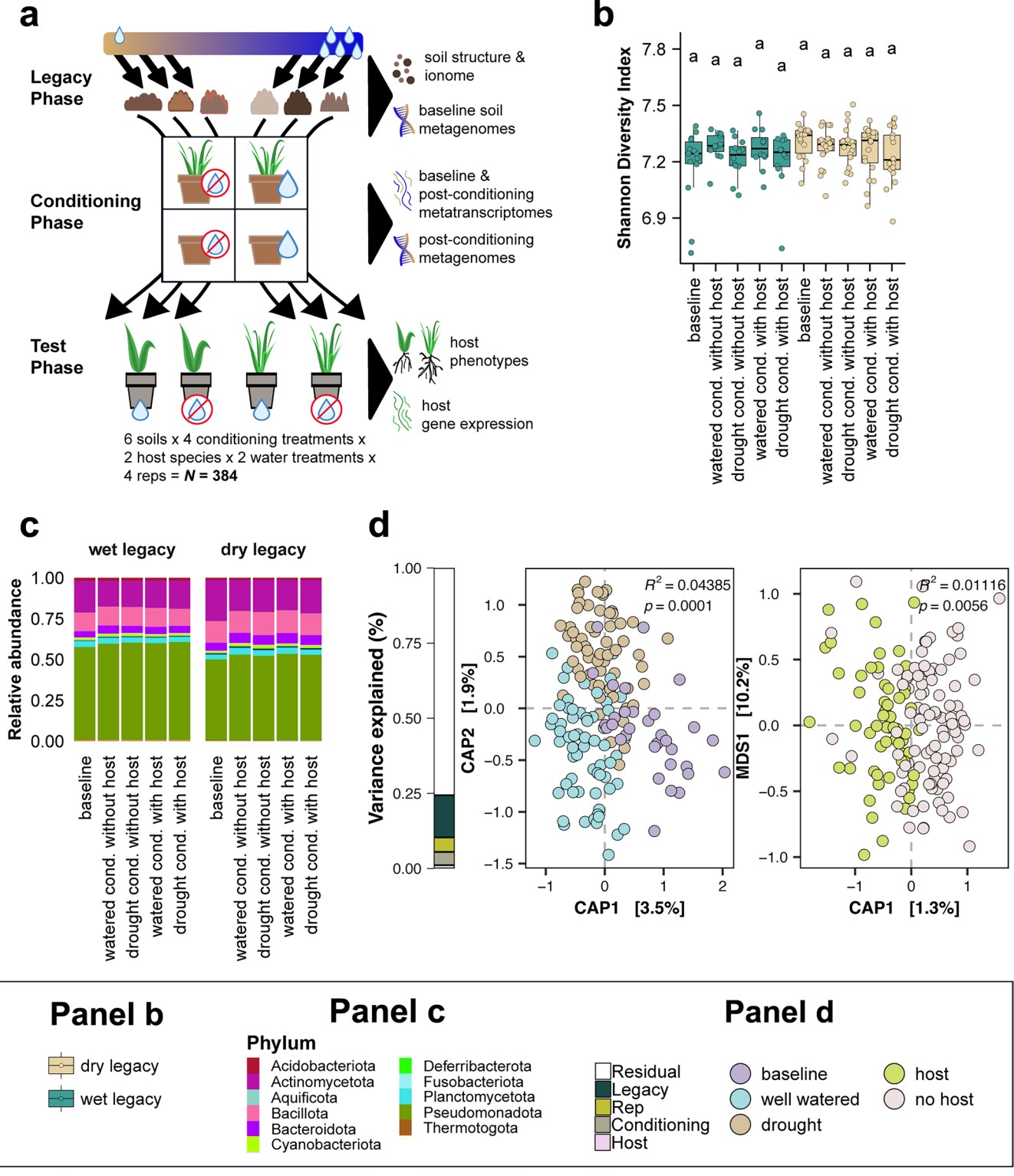

**Extended Data Fig. 4 | See next page for caption.**

**Extended Data Fig. 4 | Precipitation legacy effects in the soil microbiota are resilient to short-term water- and host-related perturbations. a**. Schematic representation of the experimental design used to evaluate the resilience of precipitation legacy effects to perturbations and their functional importance to plant drought response. Six soils spanning the Kansas precipitation gradient ("legacy phase"), were either left unplanted or planted with seedlings of the native grass species *Tripsacum dactyloides* (eastern gamagrass) and subjected to either drought conditions or regular watering in a factorial design ("conditioning phase"). To evaluate how soil precipitation legacy affects plants, and to disentangle the role of the microbiota from possible effects of co-varying abiotic soil properties, we then used the experimentally-conditioned microbial communities to inoculate a new generation of *T. dactyloides* and *Z. mays* plants. These "test phase" plants were divided between water-limited conditions and well-watered control conditions. **b**. Alpha diversity of bacterial communities was not affected by the different conditioning treatments (drought or well-watered, with or without host). For each treatment, at least 12 biological replicates (individual soil pots) were analysed, resulting in a total of 156 bacterial

community profiles generated. Alpha diversity was assessed using the Shannon Diversity Index. In the boxplots, the horizontal line represents the median; box edges indicate the interquartile range (25th-75th percentiles); and whiskers extend to the smallest and largest values within 1.5× the interquartile range. Individual data points, including outliers, are overlaid as dots. Differences among groups were tested with one-way analysis of variance (ANOVA; $p = 0.16$), which revealed no statistically significant variation. Tukey's test further confirmed the absence of significant pairwise differences, as indicated by the shared letters. **c**. Phylograms show that the different conditioning treatments (drought or well-watered, with or without host) did not impact the relative abundance profiles of main bacterial phyla. **d**. Constrained ordination of metagenome taxonomic composition in response to conditioning phase treatments. Statistics are from permutational MANOVA (PERMANOVA) with 9999 permutations. PERMANOVA $R^2$ and $p$-values for the effects of drought/water and host conditioning are shown. The bar on the left indicates the percentage of variance explained by the experimental variables, as estimated by PERMANOVA.

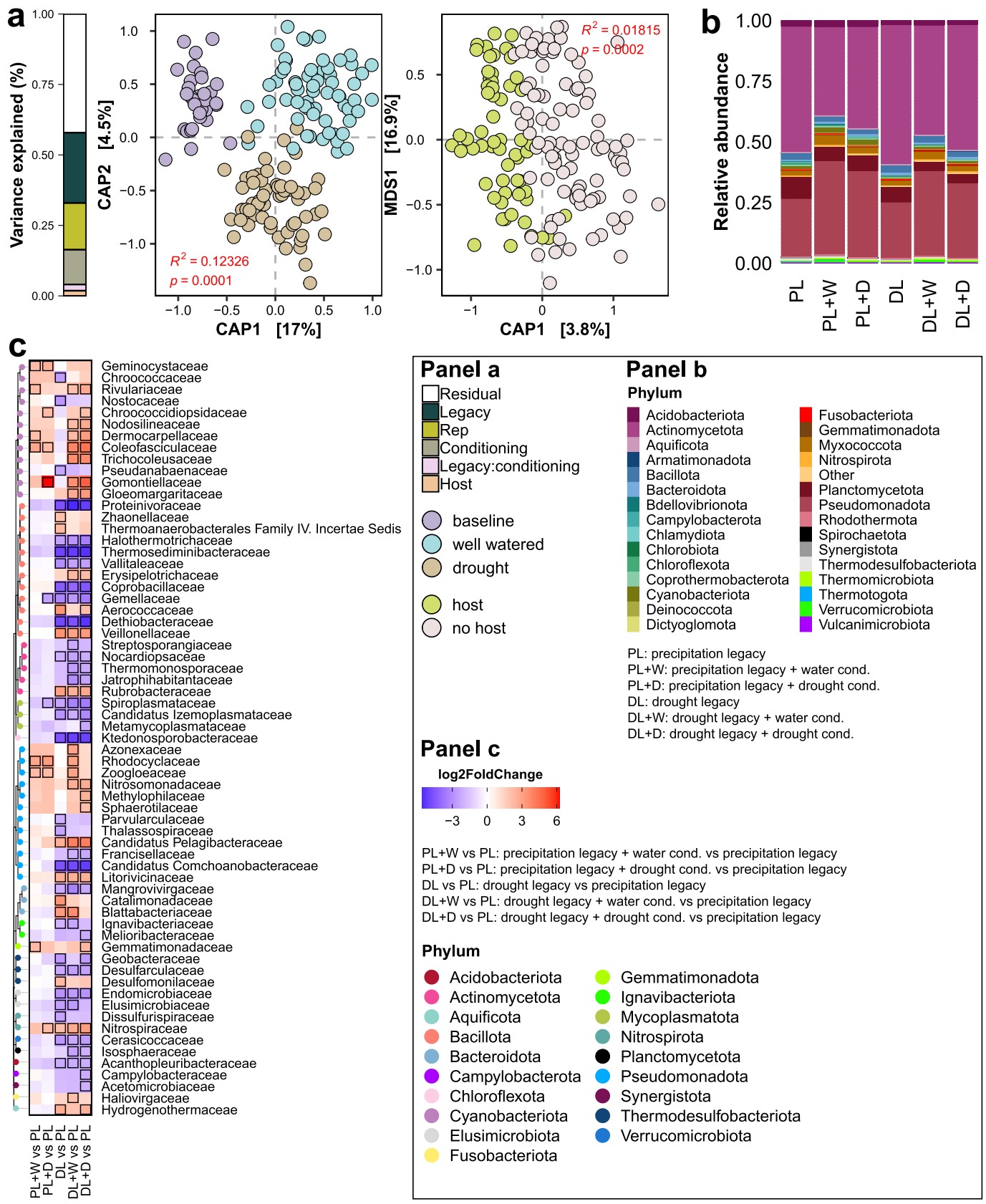

**Extended Data Fig. 5 | See next page for caption.**

**Extended Data Fig. 5 | Precipitation legacy effects shape the transcriptionally-active soil microbiota even after five months of experimental perturbation.**
**a**. Constrained ordination of soil metatranscriptome content. Bray-Curtis dissimilarity matrices were calculated from RNA-based bacterial counts to evaluate transcriptional differences in response to conditioning phase treatments: (left) baseline soils versus soils exposed to well-watered or drought conditions, and (right) soils with or without host plants. Group differences were assessed using PERMANOVA with 9,999 permutations, with the resulting $R^2$ and $p$-values indicated within the plots. The bar on the left describes the percentage of the variance explained by the experimental variables. **b**. Phylogram showing the main bacterial phyla that were transcriptionally active across soils with different precipitation legacies after five months of drought or well-watered conditions, compared to the baseline for each legacy group. **c**. Heatmap showing enrichment or depletion of transcriptionally active bacterial families relative to the baseline high-precipitation-legacy (PL) soil. DL indicates the baseline (pre-conditioning) low-precipitation-legacy soil; +W and +D indicate five months of well-watered or drought conditions, respectively. Heatmap was coloured based on $\log_2$ fold changes derived from a generalised linear model contrasting the abundance of each family in a given treatment against the high-precipitation-legacy baseline soil. Tiles outlined in black denote statistically significant enrichment (red) or depletion (blue) (q < 0.05) with a |$\log_2$ fold change| > 2. Heatmaps were clustered based on taxonomic classification (tree on the left).

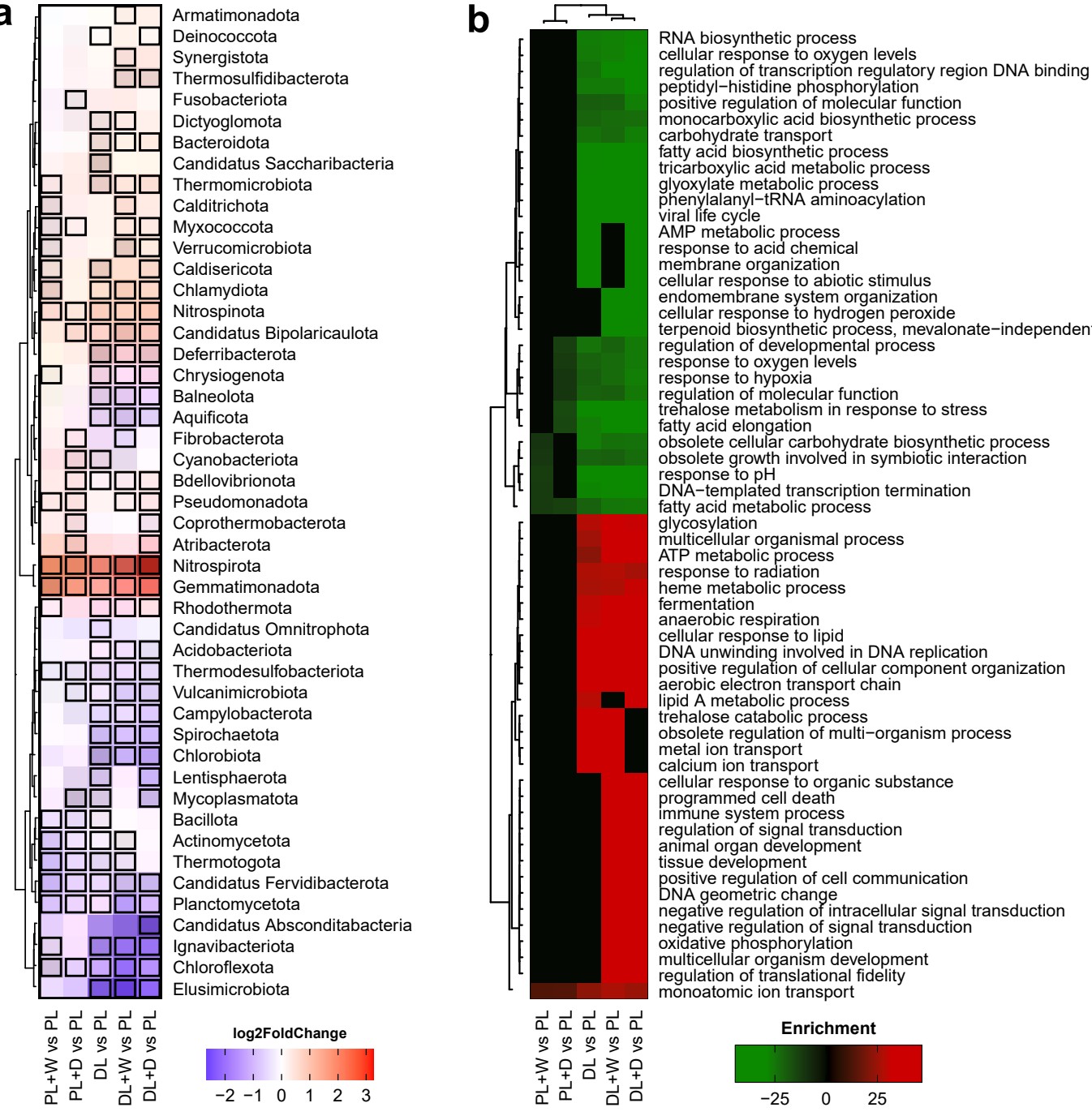

**Panels a/b**

PL+W vs PL: precipitation legacy + water cond. vs precipitation legacy
PL+D vs PL: precipitation legacy + drought cond. vs precipitation legacy
DL vs PL: drought legacy vs precipitation legacy
DL+W vs PL: drought legacy + water cond. vs precipitation legacy
DL+D vs PL: drought legacy + drought cond. vs precipitation legacy

**Extended Data Fig. 6 | See next page for caption.**

**Extended Data Fig. 6 | Transcriptional responses of soil microbiomes to short-term water perturbations are shaped by precipitation legacy.**
**a.** Heatmap showing the enrichment of transcriptionally active bacterial phyla in soils with low-precipitation (DL) or high-precipitation (PL) legacies, exposed to either drought (+D) or well-watered (+W) treatments, relative to the high-precipitation-legacy baseline (PL). DL indicates the baseline (pre-conditioning) low-precipitation-legacy soil. Colours represent $\log_2$ fold changes derived from a generalized linear model comparing each treatment to the high-precipitation-legacy baseline (PL). Tiles outlined in black indicate statistically significant enrichment (red) or depletion (blue) (q < 0.05; |$\log_2$ fold change| > 2). Taxa were hierarchically clustered based on taxonomic classification (dendrogram on the left). **b.** Heatmap depicting enriched or depleted biological processes identified through metatranscriptomic analysis. Soils with high-precipitation legacies (PL) or low-precipitation legacies (DL) were subjected to five-month-long drought or well-watered treatments and then compared to the high-precipitation-legacy baseline. Gene enrichment analysis was conducted using a generalized linear model followed by Gene Ontology (GO) classification. Enrichment of each GO category was evaluated using a two-sided Mann-Whitney U test to determine whether genes associated with the category were significantly clustered at either the top or bottom of a globally ranked gene list. $p$-values were adjusted for multiple comparisons, and categories with an adjusted $p$-value < 0.05 were considered significant. Significantly enriched or depleted GO categories (adjusted p < 0.05) are coloured according to enrichment scores, calculated from square root-transformed delta rank values (red: enrichment; green: depletion). Clustering was performed based on soil treatments and GO terms. The GO categories exhibiting the strongest enrichment or depletion profiles are presented.

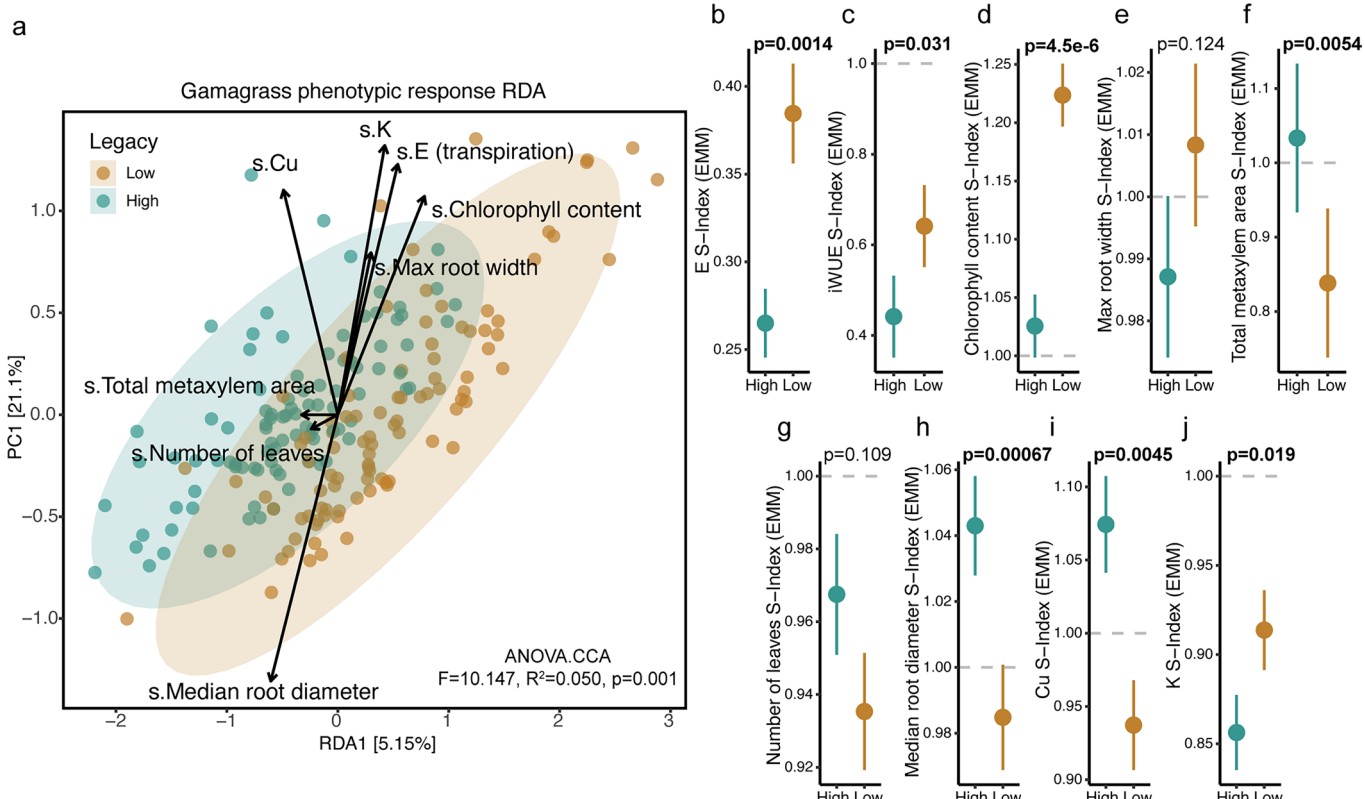

**Extended Data Fig. 7 | Precipitation legacy of microbial inoculum impacts gamagrass phenotypic drought response. a.** A constrained redundancy analysis of the top non-collinear traits found that legacy explains 5.0% of the phenotypic response to acute drought in eastern gamagrass (*Tripsacum dactyloides*; ANOVA-like permutation test, F = 10.15, R² = 0.050, p = 0.001; n = 192). Turquoise points represent plants that were inoculated with high-precipitation-legacy microbiota and brown points represent plants that were inoculated with low-precipitation-legacy microbiota. The ellipses indicate 95% confidence intervals. **b–j.**

Assessment of individual traits indicates that microbiota with a low-precipitation legacy improved gamagrass performance under drought (ANOVA, type III sums of squares; N = 191). Points are estimated marginal means and error bars represent the standard error, and significant p-values (≤0.05 after adjustment for multiple comparisons using the Benjamini-Hochberg false discovery rate) are bolded. All samples are independent biological replicates representing an individual plant.

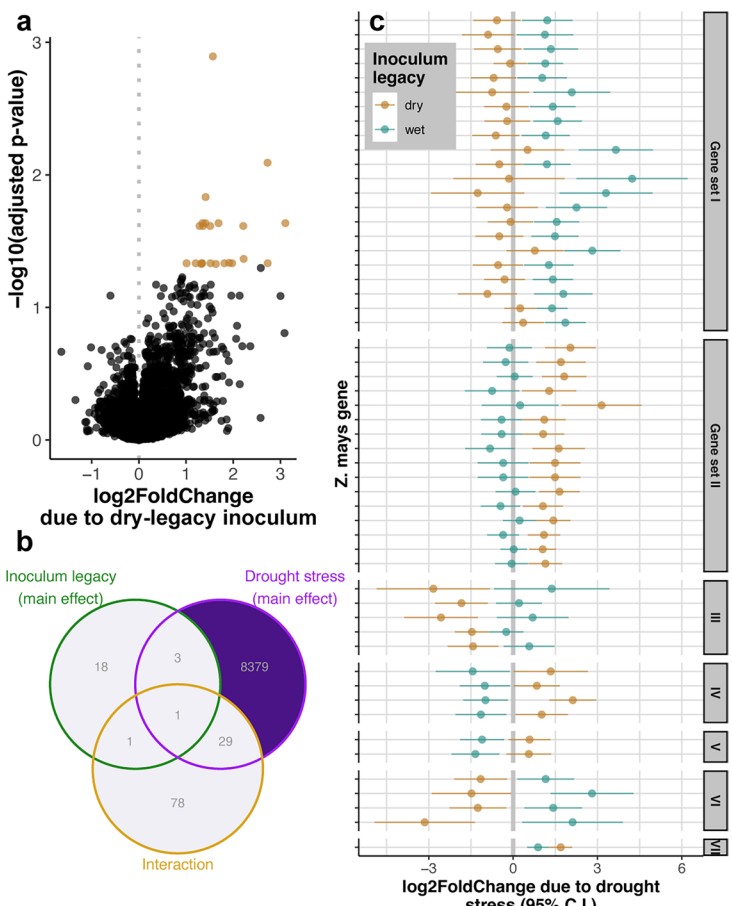

**Extended Data Fig. 8 | Precipitation legacy of microbial inocula alters the transcriptional response to drought in the maize crown root. a**. 23 genes were up-regulated in plants inoculated with soil microbiota from a low-precipitation region, relative to plants inoculated with microbiota from a high-precipitation region. **b**. The sets of genes that responded to the main effects of inoculum legacy and test phase drought treatment had little overlap with each other or with the set of genes that were sensitive to the interaction between the two. **c**. In total, 109 maize genes responded to drought in a manner that was dependent on the drought legacy of the soil microbiota (the inoculum legacy * drought treatment interaction term), regardless of the inoculum's treatment during the conditioning phase. For illustration purposes, only annotated genes with |log₂FoldChange| > 1 in at least one microbial context are shown here; the full list is available in Supplementary Table S10. Each pair of points shows one gene; the position of each point illustrates how the gene's expression changed in response to drought stress during the Test Phase, depending on whether the plant had been inoculated with microbiota derived from a low-precipitation (brown) or dry-precipitation (turquoise) environment. Genes are grouped into sets according to the pattern of how inoculum legacy altered their drought responses. Note: the names of these gene sets are not meant to correspond to the names of the *T. dactyloides* (gamagrass) gene sets shown in Fig. 4b; in each species, Gene set I contains the most genes, Gene set II contains the next most, and so on. For both **a** and **c**, statistical support is derived from a two-sided Wald test of the null hypothesis that the log₂FoldChange = 0, with false discovery rate adjustment of p-values.

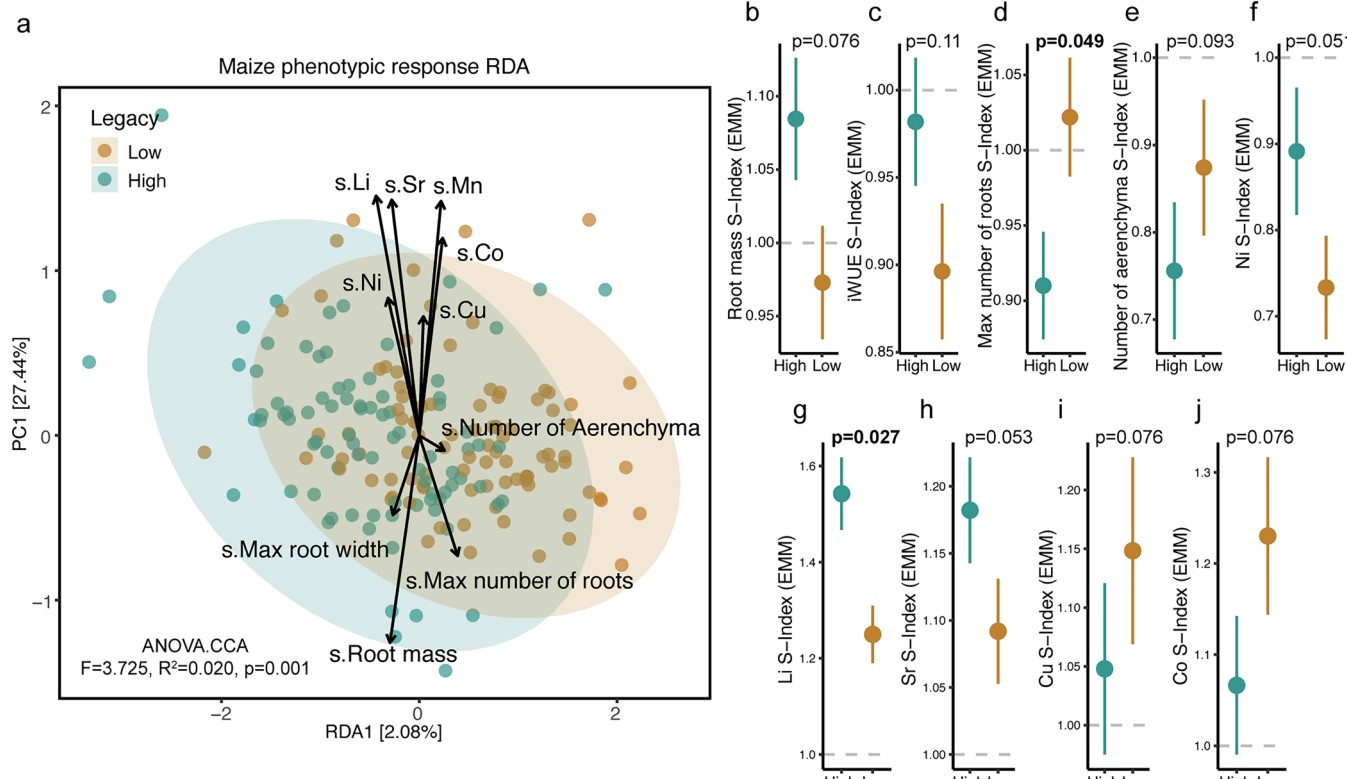

**Extended Data Fig. 9 | Precipitation legacy of the microbiota had only weak impacts on the phenotypic drought response of maize, compared to that of eastern gamagrass.** For comparison, the effects of microbiota precipitation legacy on eastern gamagrass drought responses are shown in Fig. 4 and Extended Data Fig. 7. **a.** A constrained redundancy analysis of the top non-collinear traits found that the precipitation legacy of the microbial inoculum only explains 2.0% of the maize phenotypic response to acute drought (ANOVA-like permutation test, F = 3.725, R² = 0.020, p = 0.001; n = 181 samples). Turquoise points represent root microbiomes with high-precipitation legacy and brown points represent

those with low precipitation legacy. The ellipses indicate 95% confidence intervals. **b–j.** Assessment of individual traits indicates that microbiota with a low-precipitation legacy did not significantly improve maize performance under drought, but did impact several mineral nutrient concentrations (ANOVA, type III sums of squares; n = 180 samples). Points are estimated marginal means and error bars represent the standard error, and significant p-values (≤0.05 after adjustment for multiple comparisons using the Benjamini-Hochberg false discovery rate) are bolded. All samples are independent biological replicates representing an individual plant.

Maggie R. Wagner

# Reporting Summary

## Statistics

For all statistical analyses, confirm that the following items are present in the figure legend, table legend, main text, or Methods section.

| n/a | Confirmed | |
|---|---|---|
| ☐ | ☒ | The exact sample size (*n*) for each experimental group/condition, given as a discrete number and unit of measurement |
| ☐ | ☒ | A statement on whether measurements were taken from distinct samples or whether the same sample was measured repeatedly |
| ☐ | ☒ | The statistical test(s) used AND whether they are one- or two-sided<br>*Only common tests should be described solely by name; describe more complex techniques in the Methods section.* |
| ☐ | ☒ | A description of all covariates tested |
| ☐ | ☒ | A description of any assumptions or corrections, such as tests of normality and adjustment for multiple comparisons |
| ☐ | ☒ | A full description of the statistical parameters including central tendency (e.g. means) or other basic estimates (e.g. regression coefficient) AND variation (e.g. standard deviation) or associated estimates of uncertainty (e.g. confidence intervals) |
| ☐ | ☒ | For null hypothesis testing, the test statistic (e.g. *F*, *t*, *r*) with confidence intervals, effect sizes, degrees of freedom and *P* value noted<br>*Give P values as exact values whenever suitable.* |
| ☒ | ☐ | For Bayesian analysis, information on the choice of priors and Markov chain Monte Carlo settings |
| ☐ | ☒ | For hierarchical and complex designs, identification of the appropriate level for tests and full reporting of outcomes |
| ☐ | ☒ | Estimates of effect sizes (e.g. Cohen's *d*, Pearson's *r*), indicating how they were calculated |

*Our web collection on statistics for biologists contains articles on many of the points above.*

## Software and code

Policy information about availability of computer code

| Data collection | Software packages are detailed in the Methods section. |
|---|---|
| Data analysis | Software packages are detailed in the Methods section. We deposited all scripts and source data required to reproduce the results of this study in the following Zenodo repository: doi: 10.5281/zenodo.13821005. |

For manuscripts utilizing custom algorithms or software that are central to the research but not yet described in published literature, software must be made available to editors and reviewers. We strongly encourage code deposition in a community repository (e.g. GitHub). See the Nature Portfolio guidelines for submitting code & software for further information.

## Data

Policy information about availability of data

All manuscripts must include a data availability statement. This statement should provide the following information, where applicable:
- Accession codes, unique identifiers, or web links for publicly available datasets
- A description of any restrictions on data availability
- For clinical datasets or third party data, please ensure that the statement adheres to our policy

The 16S rRNA gene amplicon sequenng data, shotgun metagenomic data, and metatranscriptome data associated with this study have been deposited in the NCBI Sequence Read Archive under the BioProject IDs PRJNA1267293, PRJNA1267715, PRJNA1268489, and PRJNA1186942. The raw RNA-seq data from gamagrass and

maize have been deposited in the Gene Expression Omnibus under accessions GSE282586 and GSE282587, respectively. Plant phenotype data and soil data are available in a Zenodo repository.

# Research involving human participants, their data, or biological material

Policy information about studies with human participants or human data. See also policy information about sex, gender (identity/presentation), and sexual orientation and race, ethnicity and racism.

| | |
|---|---|
| Reporting on sex and gender | N/A |
| Reporting on race, ethnicity, or other socially relevant groupings | N/A |
| Population characteristics | N/A |
| Recruitment | N/A |
| Ethics oversight | N/A |

Note that full information on the approval of the study protocol must also be provided in the manuscript.

# Field-specific reporting

Please select the one below that is the best fit for your research. If you are not sure, read the appropriate sections before making your selection.

☐ Life sciences    ☐ Behavioural & social sciences    ☒ Ecological, evolutionary & environmental sciences

For a reference copy of the document with all sections, see nature.com/documents/nr-reporting-summary-flat.pdf

# Ecological, evolutionary & environmental sciences study design

All studies must disclose on these points even when the disclosure is negative.

| | |
|---|---|
| Study description | Six soils from Kansas, USA were collected and thoroughly characterized using shotgun metagenome and metatranscriptome sequencing, as well as X-ray CT scanning to quantify porosity. The soils were then exposed to drought treatment or well-watered conditions, with or without a plant host (Tripsacum dactyloides) in a factorial design. In the "Conditioning Phase", a total of 192 sterile 100 mL pots were filled with the six soils and which were then randomly assigned to one of four conditions in a fully-factorial design: with or without a host, and either water-stressed or well-watered. Half the pots were planted with seedlings of the native prairie grass T. dactyloides (Eastern gamagrass, cultivar "Pete"); the rest remained unplanted. Thus, the replication was N=8 per soil per treatment. In the "Test Phase", the microbial extract from each of the 192 Conditioning Phase pots (as well as the 24 uninoculated control pots) was used to inoculate 4 pots for the "Test Phase": one pot per combination of watering treatment (droughted or control) and host species (maize or gamagrass). |
| Research sample | Six soils were collected from never-plowed prairie remnants in Kansas, USA. All of the plants measured in this study were grown from seed in growth chambers. |
| Sampling strategy | The collection sites were selected to be evenly spaced across the precipitation gradient. Six independent sub-samples were randomly selected within each of the six sites. |
| Data collection | Numerous methods of data collection were used and are detailed in the Methods section of the Supplementary Information. Data collection and recording was done by Nichole Ginnan, Valeria Custodio, and several core facilities. |
| Timing and spatial scale | The soils were collected during a single collection trip, i.e., they represent a "snapshot" in time (October 2020). The spatial scale is on the order of kilometers and is detailed in the manuscript. |
| Data exclusions | No data were excluded from the analysis other than a couple of outlier data points that were clear errors, e.g., biologically impossible data points such as a root that was recorded as 100% aerenchyma. |
| Reproducibility | For the "Test Phase" analyses, we grouped together low-precipitation and high-precipitation soils so that our experiment was effectively replicated over three distinct soils representing each precipitation category. |
| Randomization | We used a random number generator within an Excel spreadsheet to randomize the placement of plants within the growth chamber with respect to microbial inoculum and drought treatment. We did the same for the soil mesocosms (randomized with respect to starting soil, drought treatment, and presence or absence of a host plant). |
| Blinding | Soil mesocosms and plants were tracked using a non-descriptive ID number. During the experiments, we were not blinded to which replicates were assigned to the drought treatment vs. the control treatment because we needed that information to water them accordingly. During the Test Phase we also were not blinded to host species because they are visually distinct. However, we were |

blinded to the microbial inoculum treatments during data collection (that information was linked to the replicates' unique ID numbers).

Did the study involve field work? ☒ Yes ☐ No

## Field work, collection and transport

| | |
|---|---|
| Field conditions | We did not record daily weather conditions during our sampling trips. |
| Location | GPS coordinates for all collection sites are provided in Supplemental Table S1. |
| Access & import/export | The Kansas soils were collected from pre-existing research sites that were established as part of the U.S. National Science Foundation grant OIA-1656006. Permission was acquired from the corresponding land managers (The Nature Conservancy in Kansas (Smoky Valley Ranch) and the Konza Prairie Biological Station). |
| Disturbance | A minor amount of soil was removed from each collection site. To minimize disturbance, we collected and pooled several small samples rather than digging a single large hole in any site. |

# Reporting for specific materials, systems and methods

We require information from authors about some types of materials, experimental systems and methods used in many studies. Here, indicate whether each material, system or method listed is relevant to your study. If you are not sure if a list item applies to your research, read the appropriate section before selecting a response.

### Materials & experimental systems

| n/a | Involved in the study |
|---|---|
| ☒ | ☐ Antibodies |
| ☒ | ☐ Eukaryotic cell lines |
| ☒ | ☐ Palaeontology and archaeology |
| ☒ | ☐ Animals and other organisms |
| ☒ | ☐ Clinical data |
| ☒ | ☐ Dual use research of concern |
| ☐ | ☒ Plants |

### Methods

| n/a | Involved in the study |
|---|---|
| ☒ | ☐ ChIP-seq |
| ☒ | ☐ Flow cytometry |
| ☒ | ☐ MRI-based neuroimaging |

## Plants

| | |
|---|---|
| Seed stocks | Maize genotype B73 was acquired courtesy of Dr. Peter Balint-Kurti (USDA-ARS). Tripsacum dactyloides cultivar "Pete" was purchased from the Gamagrass Seed Company (Falls City, NE, USA). |
| Novel plant genotypes | N/A |
| Authentication | N/A |

