## [Peer Review File · Nature Microbiology]

Precipitation legacy effects on soil microbiota facilitate adaptive drought responses in plants

Corresponding Author: Dr Maggie Wagner

Version 0:

Reviewer comments:

Reviewer #1

(Remarks to the Author)

The manuscript "Persistent legacy effects on soil microbiota facilitate plant adaptive responses to drought" is an extremely thorough investigation of the mechanisms by which soil microbial responses to long-term precipitation differences influence plant drought tolerance. The manuscript interrogates this question about plant-microbe interactions from many different perspectives. The findings are clearly presented, and the study is one of the most rigorous I have seen in a long time. I have only a few minor comments aimed at improving this already solid and exciting manuscript.

Lines 63-65: I'm not entirely sure what is meant by "...are particularly salient at the metatranscriptomic level..."

Line 190-196: These conclusions are not immediately apparent from the prior text describing the Kansas GO results or the extended figures. This likely reflects my ignorance of GO terms, but if I read correctly, only one term mentioned in the Santiago soils is the same as those listed in the Kansas results (lipid catabolic process). Additionally, the extended data figures do little to clarify because they are too small to fully understand, and also because extended figure 2g and figure 4i do not parallel one another (different colors used on heat map, unclear if axes are the same as they are so small I cannot read them).

Line 211: It would be useful to remind the reader what proportion of samples showed the same pattern across both gradients (i.e., "We validated eight of XXXX bacterial markers from Kansas and eleven of the XXXX biomarkers from Santiago...")

Line 502-504: I think this conclusion is too strong. The fact that the host with the more stable microbiome was also the one that benefited from soil drought legacy does not necessarily mean that the stable microbiome is the cause of that difference between host species. I recommend revising to "and may be related to the ability of the host to maintain a stable microbiota...". Similarly, I would also downplay the conclusion that root microbiome stability plays a role in adaptive plant responses in lines 528-529. I think this speculation takes away from the other findings that are much more thoroughly supported.

Figures: I realize that space is tight, but the figures are so small that they are not always legible and, therefore, are not particularly useful. In the Extended figures in the supplement, could you simply put fewer panels on a page? In the main text, consider deleting panels so that the interesting data figures are more easily understood. Additionally, I think the support for your conclusions would be much stronger and more apparent to a broad audience if the biological take-home messages were presented in the figure legend. What key point do you want readers to take from each panel?

Figure 1: I would delete panel a and possibly also panel b (panel b is somewhat redundant with the subpanel showing precipitation levels across both gradients in panel g). The figure legend suggests that panel d shows the relationship between each mineral abundance and the precipitation gradient, but the x-axis appears to be microbial taxa (I believe the references to this panel in the main text also suggest it's the association with microbial taxa). Panels e and f: It is hard to differentiate which connecting lines are black.

Figure 2: In panels b, d, and e, could you also indicate the short-term treatment (wet or dry). Basically, I'm wondering if there's any trend that the short-term treatments are moving the communities towards their long-term legacy compositions.

Figure 3: I am not sure if this is possible, but could you merge panels b and c. In other words, the fact that legacy did not affect root bacterial microbiome is not very apparent from panel c (while the stats support that claim, visually the differentiation of treatments is quite similar to the effects in panel b). If you showed all 4 treatments in a single panel, the larger effect of test phase compared to legacy might be more apparent.

Supplement Page 31: I must've missed it, but why were the sample sizes uneven (lower for well-watered test phases)? The text at the top of page 30 implied equal sample sizes (192 for each of the four treatments)

(Remarks on code availability)

Given that I do not often run several of the analyses used in this manuscript, I do not feel that I would be a good judge of the code.

Reviewer #2

(Remarks to the Author)

Ginnan et al., describes a series of experiments looking the relationship between soil moisture and the phylogenetic and

functional composition of the soil microbiome and rhizosphere. The manuscript is split into two sections. The first concerns the microbial functional potential across a ~600 mm precipitation gradient in Kansas. The second provides a mechanistic look at whether the dry-adapted rhizosphere improves plant health under disturbance (soil drying).

The manuscript starts by introducing the Kansas precipitation gradient, and identifying the factors structuring functional differences within the microbiome. This is not an extensive gradient (only ~600 mm difference between the wet and dry sites), so it is interesting that the enrichment or depletion of certain traits appear robust. From here the authors attempt a rather odd 'normalization' of their data to a drier gradient in Cape Verde (range ~150 to 450 mm). I was thrown by the inclusion of these sites, and given that the rest of the manuscript concerns rhizosphere processes from the Kansas gradient I do not think it belongs in the main text. Next the manuscript establishes short-term drought experiments, both with and without plants. This is probably the most interesting section of the manuscript. In particular, the plant-microbe findings, while not necessarily new - we know that a dry-adapted rhizosphere mitigates drought stress - add much needed mechanistic information to our broader knowledge. I was particularly intrigued by the restriction of benefits to the host, whereby maize appeared to not benefit at all from a dry-adapted microbiome.

There is a lot to admire in this manuscript, the motivating questions are critical unknowns, and addressed through field measurements, laboratory incubations, sequencing, and statistically correlative approaches. There is a lot packed into one manuscript (as evidenced by the extensive supplemental materials). As such, the manuscript is quite complex, and I found myself getting a little lost in the narrative, which is frequently composed of lists of taxa and genes, and few contrasts to other studies. I have several specific comments that I detail below, however, I would recommend that the authors try to simplify the flow of the manuscript, concentrate more on truly novel outcomes of the plant-microbe experiment, compare and contrast the benefits afforded to gamagrass absent in maize, and try to improve the figures, which are remarkably busy. I would also recommend the authors pay close attention to the quality of the figures in the supplemental. In several of these figures (i.e., Fig. S2, S3, etc.) axes, and labels are illegible.

Specific comments:

Ln. 63: I think 'controlling for' as written here, refers to the statistical filtering of the data. Is that the case? Not that these soil properties are similar across the precipitation gradient? I can't find data on soil porosity, bulk density, conductance, etc. so can't say for sure how distinct they are.

Ln. 67: Pedantic point, but 'reveal' is a little sensationalist - maybe 'show', or 'demonstrate'?

Ln. 68: This sentence is quite awkward. What do the authors mean by 'buffer'? Mitigate? Or improve fitness?

Ln. 78: A similar comment to above, this word choice is a little too dramatic. I think the authors could point out mortality induced by an increasing severity and frequency of drought.

Ln. 79/80: I think this is only remarkable from our perspective, and I would avoid terms like 'their' and 'remarkable', it's anthropomorphic.

Ln. 104: What is the width of precipitation gradient?

Ln. 129: Are mineral weathering rates very different across this gradient? Seems like nutrient distribution would undoubtedly follow?

Ln. 136: To what depth?

Ln. 142: Couldn't the legacy effect hide such a relationship?

Ln. 159: What's the justification for the cut-off between low and high precipitation sites? HAY and KNZ have strong overlap in MAP distribution - what role does inter annual variability play? Why not low, medium, and high ppt distributions?

Ln. 164: Maybe rephrase as 'metabolic processes related to nitrogen cycling', also, what does this mean? Fixation, Denitrification? Or deamination and amino acid synthesis?

Ln. 169: I think it confirms that the community has different functional potential. Translating that into function is quite difficult. Are there any data on process or decomposition rates? Can the authors tease apart the role the functional potential plays from other controls on metabolism (temperature, soil moisture, etc.)?

Ln. 188: I'm confused by this section. I don't know what the motivation for it is. I am satisfied that functional potential changes with MAP from the Kansas precipitation gradient. This whole section seems unnecessary.

Ln. 202: Do these differences undermine the comparison?

Ln. 209: Strange way to write a negative result.

Ln. 259: Where do these reference genomes come from? Are the isolates from the same environment? Or from databases?

Ln. 359: I think this is a very interesting result. Functional resilience is important, but hard to demonstrate. However, the significance gets lost in the narrative style of the manuscript. Can it be drawn out a little better?

Ln. 373: By experimentally conditioned, these are the communities selected for by the fluctuating wet-dry experiment without plants?

Ln. 403: I don't understand this sentence.

Ln. 409-412: This is all quite confusing.

Ln. 438: How was transpiration measured here?

Ln. 448: I'm not sure I understand this - are these correlations between transpiration and rhizosphere gene expression? There are a lot of gates between the rhizosphere and stomatal control, so what does this mean?

Ln. 495: Signaling = signalling.

Ln. 564: What is a 'less stable' microbiome? Large changes in microbiome function or phylogenetic composition? And on temporal or spatial scales?

(Remarks on code availability)

I'm unable to access the zenodo site. The link redirects to the home page, and finding datasets that have yet to be officially released is not possible.

Reviewer #3

(Remarks to the Author)

The article titled "Persistent legacy effects on soil microbiota facilitate plant adaptive responses to drought" seeks to demonstrate that legacy soil moisture treatments have similar effects on soil microbiota across soil types as well as on the ability of a native grass to withstand drought treatments. It is overall well written but could use some improvement before its suitable for publication with Nature Microbiology.

The biggest issue was in the supplementary (extended data) figures which were of poor resolution, and I couldn't read/understand them. This hindered my ability to adequately review those figures and the results associated with them.

Furthermore, in my opinion it's important to distinguish between functional capacity represented by metagenomic data and active functions represented by metatranscriptomics data. Often the author refers to the metagenomic data as representing function, however, this is more accurate to describe this as functional capacity. Also, in discussing metagenomic data and it's not changing during the 5 months of drought or control treatment when plants are introduced, I think it's important to mention that these data are captured through DNA which is more persistent than RNA and could represent microbes that are no longer active or even relic DNA.

The authors used a vast array of statistical analyses, some of which I'm not as familiar with, but I am familiar with DESeq2 which requires count data. In the methods, the authors mention they use TPM as input which would be incorrect.

Overall, I also think that the results need to be discussed more and integrated with current literature. For example, when mentioned which genes are associated with the precipitation gradients, there is minimal discussion of why this would be or how they would be associated with higher or lower moisture. As it is, the overall traits associated with low or high moisture legacy are not clear or integrated in with the current literature.

Some additional comments by line:

Line 80: The authors could mention how much variation is represented by PC1.

Line 81: The darkest colors of each precipitation regime are very similar to each other and hard to distinguish.

Line 89: Again, add percentage explained by PC1.

Line 92: Can you list the individual mineral nutrients that were correlated with annual precipitation since you say there are just several (plus I couldn't read the extended data at all to see which ones they were).

Line 120-121: "water availability" is used twice in this sentence; maybe you could reduce to once and alter terminology.

Line 136: As mentioned above, I would say these communities have different functional capacities.

Line 138: What are the specific functions that are specifically related to osmotic stress that were different? Or do you mean the differences listed above are due to adaptations to the different moisture levels? If so, I wouldn't state "osmotic stress" here. (Maybe this is shown in the figure, but I can't read.)

Line 170: While I can't read extended data fig 4 due to the resolution, I can also tell that the font size is too small.

Line 170: Again, say what these nutrients are. I think if there are just several, it's nice to state in the manuscript for the readers.

Lines 190-195: Could you make these comparisons more clear? Perhaps either listing all the similar and different ones in the text, making a table or in an easy to read figure? As it is, I'm not convinced of the extent of similarity vs. difference.

Line 224: It's not clear what you are referring to as the focal species.

Line 225: I don't think this is referring to the correct figure.

Line 237-240: Can you discuss why these genes would be associated with "drought tolerance"?

Line 265-266: To me it was unclear what the treatment groups were, do they only include precipitation or rhizosphere vs bulk too?

Line 167: Does the PCA of metagenomics here refer to the taxonomy or genes? You switch back and forth above between both ways, so would be good to specify here.

Lines 278-279: Extended data fig 6i is mentioned twice in this sentence, but only need to refer to once.

Line 297: Again, refer to as bacterial functional capacity.

Line 300: Enrichment of GO terms in what treatment?

Line 303: Which categories were similar? Could you discuss why you think this is?

Lines 314-328: These are some big claims, but its hard for me to evaluate because I can't read the figures. Also, was there a comparison made between drought and control treatments regardless of legacy? Here it only states the functions that didn't change, but what did change? Just so we get the whole picture.

Line 368: I don't see a figure 2 or 2g.

Methods:

Section 1.1

How long were soils stored at 4C until DNA extractions? The length of time sitting at 4C could impact communities.

Do you think air-drying the soils could impact microbial communities?

Section 1.8.7

Why do you filter to just bacterial species? What about archaea?

DESeq2 should be used on count data, not coverage (or TPM). Can you justify why you used these other more normalized data as input? Do you get similar results when using raw counts?

Section 2.1

You introduce the grass without spelling out the genus name. Just introduce it once as genus species, then use the genus abbreviation throughout the rest of the manuscript. Also, in figure legends be consistent and follow journals requirements for how to mention species names because its not consistent currently.

Section 2.8.2

How were the sequences assembled into contigs?

Section 2.8.4

Make sure to cite all packages/programs used (TransDecoder has no reference)

Extended data Fig. 6: There are too many panels here. Even if the resolution was okay, it would still be difficult to read. Maybe split into two?

(Remarks on code availability)

Decision Letter:

28th March 2025

Dear Dr Wagner,

Thank you for your patience while your manuscript "Persistent legacy effects on soil microbiota facilitate plant adaptive responses to drought" was under peer-review at Nature Microbiology. It has now been seen by 3 referees, whose expertise and comments you will find at the end of this email. Although they find your work of some potential interest, they have raised a number of concerns that will need to be addressed before we can consider publication of the work in Nature Microbiology.

The main issue here is the presentation of the storyline of the paper, and the organization of the data as it pertains to the narrative you're trying to tell. (Hopefully this is easier to address than technical concerns--of which there were only a few, and all seem to be minor!) Our editorial advice is to hone in on a simpler, clearer main message, and to really focus the manuscript on the most novel parts: take out the gradient comparisons and focus instead on how microbial climate legacies impact plant fitness, and the importance of host specificity. If you need any editorial assistance as you reframe, please don't hesitate to drop me a line. I am happy to lend my expertise!

Another important point: please be sure that code is made available for the reviewers to assess upon submission of your revised manuscript!

Should further experimental data and editing allow you to address these criticisms, we would be happy to look at a revised manuscript.

Please include a data availability statement as a separate section after Methods but before references, under the heading "Data Availability". This section should inform readers about the availability of the data used to support the conclusions of your study. This information includes accession codes to public repositories (data banks for protein, DNA or RNA sequences, microarray, proteomics data etc...), references to source data published alongside the paper, unique identifiers such as URLs to data repository entries, or data set DOIs, and any other statement about data availability. At a minimum, you should include the following statement: "The data that support the findings of this study are available from the corresponding author upon request", mentioning any restrictions on availability. If DOIs are provided, we also strongly encourage including these in the Reference list (authors, title, publisher (repository name), identifier, year). For more guidance on how to write this section please see: <http://www.nature.com/authors/policies/data/data-availability-statements-data-citations.pdf>

* If you have not done so already we suggest that you begin to revise your manuscript so that it conforms to our Article format instructions at <http://www.nature.com/nmicrobiol/info/final-submission>. Refer also to any guidelines provided in this letter.

When submitting the revised version of your manuscript, please pay close attention to our [href="https://www.nature.com/nature-portfolio/editorial-policies/image-integrity">Digital Image Integrity Guidelines.](https://www.nature.com/nature-portfolio/editorial-policies/image-integrity) and to the following points below:

EXTENDED DATA FIGURES

Link Redacted

Note: This url links to your confidential homepage and associated information about manuscripts you may have submitted or be reviewing for us. If you wish to forward this e-mail to co-authors, please delete this link to your homepage first.

Nature Microbiology is committed to improving transparency in authorship. As part of our efforts in this direction, we are now requesting that all authors identified as 'corresponding author' on published papers create and link their Open Researcher and Contributor Identifier (ORCID) with their account on the Manuscript Tracking System (MTS), prior to acceptance. This applies to primary research papers only. ORCID helps the scientific community achieve unambiguous attribution of all scholarly contributions. You can create and link your ORCID from the home page of the MTS by clicking on 'Modify my Springer Nature account'. For more information please visit www.springernature.com/orcid.

If you wish to submit a suitably revised manuscript we would hope to receive it within 6 months. If you cannot send it within this time, please let us know. We will be happy to consider your revision, even if a similar study has been accepted for publication at Nature Microbiology or published elsewhere (up to a maximum of 6 months).

Yours sincerely,

Reviewer Expertise:

Referee #1: soil biogeochemistry, ecological memory

Referee #2: drought impacts, soil microbiology, biogeochem (mass spec), modeling

Referee #3: drought impacts, soil microbiology, 'omics

Reviewer Comments:

Reviewer #1 (Remarks to the Author):

The manuscript "Persistent legacy effects on soil microbiota facilitate plant adaptive responses to drought" is an extremely thorough investigation of the mechanisms by which soil microbial responses to long-term precipitation differences influence plant drought tolerance. The manuscript interrogates this question about plant-microbe interactions from many different perspectives. The findings are clearly presented, and the study is one of the most rigorous I have seen in a long time. I have only a few minor comments aimed at improving this already solid and exciting manuscript.

Lines 63-65: I'm not entirely sure what is meant by "...are particularly salient at the metatranscriptomic level..."

Line 190-196: These conclusions are not immediately apparent from the prior text describing the Kansas GO results or the extended figures. This likely reflects my ignorance of GO terms, but if I read correctly, only one term mentioned in the Santiago soils is the same as those listed in the Kansas results (lipid catabolic process). Additionally, the extended data figures do little to clarify because they are too small to fully understand, and also because extended figure 2g and figure 4i do not parallel one another (different colors used on heat map, unclear if axes are the same as they are so small I cannot read them).

Line 211: It would be useful to remind the reader what proportion of samples showed the same pattern across both gradients (i.e., "We validated eight of XXXX bacterial markers from Kansas and eleven of the XXXX biomarkers from Santiago...")

Line 502-504: I think this conclusion is too strong. The fact that the host with the more stable microbiome was also the one that benefited from soil drought legacy does not necessarily mean that the stable microbiome is the cause of that difference between host species. I recommend revising to "and may be related to the ability of the host to maintain a stable microbiota...". Similarly, I would also downplay the conclusion that root microbiome stability plays a role in adaptive plant responses in lines 528-529. I think this speculation takes away from the other findings that are much more thoroughly supported.

Figures: I realize that space is tight, but the figures are so small that they are not always legible and, therefore, are not particularly useful. In the Extended figures in the supplement, could you simply put fewer panels on a page? In the main text, consider deleting panels so that the interesting data figures are more easily understood. Additionally, I think the support for your conclusions would be much stronger and more apparent to a broad audience if the biological take-home messages were presented in the figure legend. What key point do you want readers to take from each panel?

Figure 1: I would delete panel a and possibly also panel b (panel b is somewhat redundant with the subpanel showing precipitation levels across both gradients in panel g). The figure legend suggests that panel d shows the relationship between each mineral abundance and the precipitation gradient, but the x-axis appears to be microbial taxa (I believe the references to this panel in the main text also suggest it's the association with microbial taxa). Panels e and f: It is hard to differentiate which connecting lines are black.

Figure 2: In panels b, d, and e, could you also indicate the short-term treatment (wet or dry). Basically, I'm wondering if there's any trend that the short-term treatments are moving the communities towards their long-term legacy compositions.

Figure 3: I am not sure if this is possible, but could you merge panels b and c. In other words, the fact that legacy did not affect root bacterial microbiome is not very apparent from panel c (while the stats support that claim, visually the differentiation of treatments is quite similar to the effects in panel b). If you showed all 4 treatments in a single panel, the larger effect of test phase compared to legacy might be more apparent.

Supplement Page 31: I must've missed it, but why were the sample sizes uneven (lower for well-watered test phases)? The text at the top of page 30 implied equal sample sizes (192 for each of the four treatments)

Reviewer #1 (Remarks on code availability):

Given that I do not often run several of the analyses used in this manuscript, I do not feel that I would be a good judge of the code.

Reviewer #2 (Remarks to the Author):

Ginnan et al., describes a series of experiments looking the relationship between soil moisture and the phylogenetic and functional composition of the soil microbiome and rhizosphere. The manuscript is split into two sections. The first concerns the microbial functional potential across a ~600 mm precipitation gradient in Kansas. The second provides a mechanistic look at whether the dry-adapted rhizosphere improves plant health under disturbance (soil drying).

The manuscript starts by introducing the Kansas precipitation gradient, and identifying the factors structuring functional differences within the microbiome. This is not an extensive gradient (only ~600 mm difference between the wet and dry sites), so it is interesting that the enrichment or depletion of certain traits appear robust. From here the authors attempt a rather odd 'normalization' of their data to a drier gradient in Cape Verde (range ~150 to 450 mm). I was thrown by the inclusion of these sites, and given that the rest of the manuscript concerns rhizosphere processes from the Kansas gradient I do not think it belongs in the main text. Next the manuscript establishes short-term drought experiments, both with and without plants. This is probably the most interesting section of the manuscript. In particular, the plant-microbe findings, while not necessarily new - we know that a dry-adapted rhizosphere mitigates drought stress - add much needed mechanistic information to our broader knowledge. I was particularly intrigued by the restriction of benefits to the host, whereby maize appeared to not benefit at all from a dry-adapted microbiome.

There is a lot to admire in this manuscript, the motivating questions are critical unknowns, and addressed through field measurements, laboratory incubations, sequencing, and statistically correlative approaches. There is a lot packed into one manuscript (as evidenced by the extensive supplemental materials). As such, the manuscript is quite complex, and I found myself getting a little lost in the narrative, which is frequently composed of lists of taxa and genes, and few contrasts to other studies. I have several specific comments that I detail below, however, I would recommend that the authors try to simplify the flow of the manuscript, concentrate more on truly novel outcomes of the plant-microbe experiment, compare and contrast the benefits afforded to gamagrass absent in maize, and try to improve the figures, which are remarkably busy. I would also recommend the authors pay close attention to the quality of the figures in the supplemental. In several of these figures (i.e., Fig. S2, S3, etc.) axes, and labels are illegible.

Specific comments:

Ln. 63: I think 'controlling for' as written here, refers to the statistical filtering of the data. Is that the case? Not that these soil properties are similar across the precipitation gradient? I can't find data on soil porosity, bulk density, conductance, etc. so can't say for sure how distinct they are.

Ln. 67: Pedantic point, but 'reveal' is a little sensationalist - maybe 'show', or 'demonstrate'?

Ln. 68: This sentence is quite awkward. What do the authors mean by 'buffer'? Mitigate? Or improve fitness?

Ln. 78: A similar comment to above, this word choice is a little too dramatic. I think the authors could point out mortality induced by an increasing severity and frequency of drought.

Ln. 79/ 80: I think this is only remarkable from our perspective, and I would avoid terms like 'their' and 'remarkable', it's anthropomorphic.

Ln. 104: What is the width of precipitation gradient?

Ln. 129: Are mineral weathering rates very different across this gradient? Seems like nutrient distribution would undoubtedly follow?

Ln. 136: To what depth?

Ln. 142: Couldn't the legacy effect hide such a relationship?

Ln. 159: What's the justification for the cut-off between low and high precipitation sites? HAY and KNZ have strong overlap in MAP distribution - what role does inter annual variability play? Why not low, medium, and high ppt distributions?

Ln. 164: Maybe rephrase as 'metabolic processes related to nitrogen cycling', also, what does this mean? Fixation, Denitrification? Or deamination and amino acid synthesis?

Ln. 169: I think it confirms that the community has different functional potential. Translating that into function is quite difficult. Are there any data on process or decomposition rates? Can the authors tease apart the role the functional potential plays from other controls on metabolism (temperature, soil moisture, etc.)?

Ln. 188: I'm confused by this section. I don't know what the motivation for it is. I am satisfied that functional potential changes with MAP from the Kansas precipitation gradient. This whole section seems unnecessary.

Ln. 202: Do these differences undermine the comparison?

Ln. 209: Strange way to write a negative result.

Ln. 259: Where do these reference genomes come from? Are the isolates from the same environment? Or from databases?

Ln. 359: I think this is a very interesting result. Functional resilience is important, but hard to demonstrate. However, the significance gets lost in the narrative style of the manuscript. Can it be drawn out a little better?

Ln. 373: By experimentally conditioned, these are the communities selected for by the fluctuating wet-dry experiment without plants?

Ln. 403: I don't understand this sentence.

Ln. 409-412: This is all quite confusing.

Ln. 438: How was transpiration measured here?

Ln. 448: I'm not sure I understand this - are these correlations between transpiration and rhizosphere gene expression? There are a lot of gates between the rhizosphere and stomatal control, so what does this mean?

Ln. 495: Signaling = signalling.

Ln. 564: What is a 'less stable' microbiome? Large changes in microbiome function or phylogenetic composition? And on

temporal or spatial scales?

Reviewer #2 (Remarks on code availability):

I'm unable to access the zenodo site. The link redirects to the home page, and finding datasets that have yet to be officially released is not possible.

Reviewer #3 (Remarks to the Author):

The article titled "Persistent legacy effects on soil microbiota facilitate plant adaptive responses to drought" seeks to demonstrate that legacy soil moisture treatments have similar effects on soil microbiota across soil types as well as on the ability of a native grass to withstand drought treatments. It is overall well written but could use some improvement before its suitable for publication with Nature Microbiology.

The biggest issue was in the supplementary (extended data) figures which were of poor resolution, and I couldn't read/understand them. This hindered my ability to adequately review those figures and the results associated with them.

Furthermore, in my opinion its important to distinguish between functional capacity represented by metagenomic data and active functions represented by metatranscriptomics data. Often the author refers to the metagenomic data as representing function, however, this its more accurate to describe this as functional capacity. Also, in discussing metagenomic data and it's not changing during the 5 months of drought or control treatment when plants are introduced, I think its important to mention that these data are captured through DNA which is more persistent than RNA and could represent microbes that are no longer active or even relic DNA.

The authors used a vast array of statistical analyses, some of which I'm not as familiar with, but I am familiar with DESeq2 which requires count data. In the methods, the authors mention they use TPM as input which would be incorrect.

Overall, I also think that the results need to be discussed more and integrated with current literature. For example, when mentioned which genes are associated with the precipitation gradients, there is minimal discussion of why this would be or how they would be associated with higher or lower moisture. As it is, the overall traits associated with low or high moisture legacy are not clear or integrated in with the current literature.

Some additional comments by line:

Line 80: The authors could mention how much variation is represented by PC1.

Line 81: The darkest colors of each precipitation regime are very similar to each other and hard to distinguish.

Line 89: Again, add percentage explained by PC1.

Line 92: Can you list the individual mineral nutrients that were correlated with annual precipitation since you say there are just several (plus I couldn't read the extended data at all to see which ones they were).

Line 120-121: "water availability" is used twice in this sentence; maybe you could reduce to once and alter terminology.

Line 136: As mentioned above, I would say this communities have different functional capacities.

Line 138: What are the specific functions that are specifically related to osmotic stress that were different? Or do you mean the differences listed above are due to adaptations to the different moisture levels? If so, I wouldn't state "osmotic stress" here. (Maybe this is shown in the figure, but I can't read.)

Line 170: While I can't read extended data fig 4 due to the resolution, I can also tell that the font size is too small.

Line 170: Again, say what these nutrients are. I think if there are just several, it's nice to state in the manuscript for the readers.

Lines 190-195: Could you make these comparison more clear? Perhaps either listing all the similar and different ones in the text, making a table or in a easy to read figure? As it is, I'm not convinced of the extent of similarity vs. difference.

Line 224: Its not clear what you are referring to as the focal species.

Line 225: I don't this this is referring to the correct figure.

Line 237-240: Can you discuss why these gens would be associated with "drought tolerance"?

Line 265-266: To me it was unclear what the treatment groups were, do they only include precipitation or rhizosphere vs bulk too?

Line 167: Does the PCA of metagenomics here refer to the taxonomy or genes? You switch back and forth above between both ways, so would be good to specify here.

Lines 278-279: Extended data fig 6i is mentioned twice in this sentence, but only need to refer to once.

Line 297: Again, refer to as bacterial functional capacity.

Line 300: Enrichment of GO terms in what treatment?

Line 303: Which categories were similar? Could you discuss why you think this is?

Lines 314-328: These are some big claims, but its hard for me to evaluate because I can't read the figures. Also, was there a comparison made between drought and control treatments regardless of legacy? Here it only states the functions that didn't change, but what did change? Just so we get the whole picture.

Line 368: I don't see a figure 2 or 2g.

Methods:

Section 1.1

How long were soils stored at 4C until DNA extractions? The length of time sitting at 4C could impact communities.

Do you think air-drying the soils could impact microbial communities?

Section 1.8.7

Why do you filter to just bacterial species? What about archaea?

DESeq2 should be used on count data, not coverage (or TPM). Can you justify why you used these other more normalized data as input? Do you get similar results when using raw counts?

Section 2.1

You introduce the grass without spelling out the genus name. Just introduce it once as genus species, then use the genus abbreviation throughout the rest of the manuscript. Also, in figure legends be consistent and follow journals requirements for how to mention species names because its not consistent currently.

Section 2.8.2

How were the sequences assembled into contigs?

Section 2.8.4

Make sure to cite all packages/programs used (TransDecoder has no reference)

Extended data Fig. 6: There are too many panels here. Even if the resolution was okay, it would still be difficult to read. Maybe split into two?

Version 1:

Reviewer comments:

Reviewer #1

(Remarks to the Author)

The authors have done a good job responding to my earlier comments. I continue to be enthusiastic about the rigor of the study and the breadth of approaches the authors bring together to make a convincing story. I have only a few additional comments for them to consider.

Line 69: Does this paragraph really “disentangle” the influence of precipitation from co-varying edaphic properties (additionally, the influence on what?)? It seems like this is just associating soil properties with precipitation, and the causal direction is unclear. To really disentangle the effects of precip from edaphic properties, I think you would want to do some sort of multiple regression to control for edaphic factors when testing the effects of precipitation or perhaps something like structural equation modeling given that you suggest causal directions (lines 79-83).

Line 260: The manuscript alternates between using both “gamagrass” and *T. dactyloides*. I’d stick with one.

Line 394: should this read: “blinded to treatment allocation”?

Line 406-407: I think there might be some inconsistencies in language here. In line 406, should “site” be “subplot”? And in line 407, should “geographical location” be “site”?

Line 808: Here and elsewhere. 225°C would be exceptionally hot for drying plant biomass. Are you sure that temperature wasn’t in Fahrenheit?

Lines 902-904; Lines 1258 vs 1279 and elsewhere: I’m having trouble interpreting the statistical models, in part because of vague or inconsistent model terms. For example in line 903, what is the difference between “conditioning” and “Condition”?

Similarly, what is “CondGroup) in lines 1433 and 1446? In line 1258, “TestPhaseWater” is used but in line 1279, “Test phase drought treatment” is used to refer to the same thing. Consistency in language would make it much easier for the reader to follow the analyses.

Line 911: Here and elsewhere, shouldn’t Biological Replicate be a random factor, rather than fixed?

Line 1063: I recommend changing to “...(plus 8 sterile buffer-only...). This would help make the total N clear.

Line 1420: What is the “mock” treatment?

Line 1420-1421: How were these 67 features selected? Also, remind the reader what these features were (traits?).

Data availability: Where will the plant phenotypic data be archived?

(Remarks on code availability)

Reviewer #3

(Remarks to the Author)

Ginnan et al. did an excellent job addressing all the reviewers’ comments and the manuscript reads much more smoothly now with a improved straight forward story.

I just have a couple additional comments regarding the data analyses.

1. In the methods, there are two approaches to metatranscriptomics read taxonomy classification. First, in section 2.7.3 the author mentions using kraken v. 2.1.2. Then, in section 2.8.4 they mention using CAT v.8.22. Could the authors clarify which

method they use to assign taxonomy to metatranscriptomics reads?

2. (line 639) Did you use a multiple comparison correction for these p-values (or in all cases you did multiple comparisons?)

(Remarks on code availability)

I looked through the code and it was well organized and interpretable for reproducibility.

Decision Letter:

Our ref: NMICROBIOL-25010354A

31st July 2025

Dear Maggie,

Thank you for submitting your revised manuscript "Persistent legacy effects on soil microbiota facilitate plant adaptive responses to drought" (NMICROBIOL-25010354A). It has now been seen by the original referees and their comments are below. Some good news to share! The reviewers find that the paper has improved in revision, and therefore we'll be happy in principle to publish it in Nature Microbiology, pending minor revisions to satisfy the referees' final requests and to comply with our editorial and formatting guidelines.

Thank you again for your interest in Nature Microbiology Please do not hesitate to contact me if you have any questions.

Sincerely,

Reviewer #1 (Remarks to the Author):

The authors have done a good job responding to my earlier comments. I continue to be enthusiastic about the rigor of the study and the breadth of approaches the authors bring together to make a convincing story. I have only a few additional comments for them to consider.

Line 69: Does this paragraph really "disentangle" the influence of precipitation from co-varying edaphic properties (additionally, the influence on what?)? It seems like this is just associating soil properties with precipitation, and the causal direction is unclear. To really disentangle the effects of precip from edaphic properties, I think you would want to do some sort of multiple regression to control for edaphic factors when testing the effects of precipitation or perhaps something like structural equation modeling given that you suggest causal directions (lines 79-83).

Line 260: The manuscript alternates between using both "gamagrass" and *T. dactyloides*. I'd stick with one.

Line 394: should this read: "blinded to treatment allocation"?

Line 406-407: I think there might be some inconsistencies in language here. In line 406, should "site" be "subplot"? And in line 407, should "geographical location" be "site"?

Line 808: Here and elsewhere. 225°C would be exceptionally hot for drying plant biomass. Are you sure that temperature wasn't in Fahrenheit?

Lines 902-904; Lines 1258 vs 1279 and elsewhere: I'm having trouble interpreting the statistical models, in part because of vague or inconsistent model terms. For example in line 903, what is the difference between "conditioning" and "Condition"? Similarly, what is "CondGroup" in lines 1433 and 1446? In line 1258, "TestPhaseWater" is used but in line 1279, "Test phase drought treatment" is used to refer to the same thing. Consistency in language would make it much easier for the reader to follow the analyses.

Line 911: Here and elsewhere, shouldn't Biological Replicate be a random factor, rather than fixed?

Line 1063: I recommend changing to "... (plus 8 sterile buffer-only...). This would help make the total N clear.

Line 1420: What is the "mock" treatment?

Line 1420-1421: How were these 67 features selected? Also, remind the reader what these features were (traits?).

Data availability: Where will the plant phenotypic data be archived?

Reviewer #3 (Remarks to the Author):

Ginnan et al. did an excellent job addressing all the reviewers' comments and the manuscript reads much more smoothly now with a improved straight forward story.

I just have a couple additional comments regarding the data analyses.

1. In the methods, there are two approaches to metatranscriptomics read taxonomy classification. First, in section 2.7.3 the author mentions using kraken v. 2.1.2. Then, in section 2.8.4 they mention using CAT v.8.22. Could the authors clarify which method they use to assign taxonomy to metatranscriptomics reads?

2. (line 639) Did you use a multiple comparison correction for these p-values (or in all cases you did multiple comparisons?)

Reviewer #3 (Remarks on code availability):

I looked through the code and it was well organized and interpretable for reproducibility.

Version 2:

Decision Letter:

15th September 2025

Hello again, Maggie,

I am pleased to accept your Article "Precipitation legacy effects on soil microbiota facilitate adaptive drought responses in plants" for publication in Nature Microbiology. Thank you for having chosen to submit your work to us and many congratulations.

Authors may need to take specific actions to achieve compliance with funder and institutional open access mandates. If your research is supported by a funder that requires immediate open access (e.g. according to [Plan S principles](https://www.springernature.com/gp/open-science/plan-s-compliance) or the [NIH public access policy](https://www.springernature.com/gp/open-science/us-federal-agency-compliance)) then you should select the gold OA route, and we will direct you to the compliant route where possible. Because authors warrant under our subscription licensing terms that they haven't committed to licensing any version of their article under a licence inconsistent with the terms of our agreement – including the applicable embargo period – publication under the subscription model isn't suitable for authors whose funders require no embargo.

We welcome the submission of potential cover material (including a short caption of around 40 words) related to your manuscript; suggestions should be sent to Nature Microbiology as electronic files (the image should be 300 dpi at 210 x 297 mm

in either TIFF or JPEG format). Please note that such pictures should be selected more for their aesthetic appeal than for their scientific content, and that colour images work better than black and white or grayscale images. Please do not try to design a cover with the Nature Microbiology logo etc., and please do not submit composites of images related to your work. I am sure you will understand that we cannot make any promise as to whether any of your suggestions might be selected for the cover of the journal.

With kind regards,

P.S. Click on the following link if you would like to recommend Nature Microbiology to your librarian
<http://www.nature.com/subscriptions/recommend.html#forms>

** Visit the Springer Nature Editorial and Publishing website at http://editorial-jobs.springernature.com?utm_source=ejP_NMicro_email&utm_medium=ejP_NMicro_email&utm_campaign=ejp_NMicro for more information about our career opportunities. If you have any questions please click [here](mailto:editorial.publishing.jobs@springernature.com).

Reviewer #1 (Remarks to the Author):

The manuscript “Persistent legacy effects on soil microbiota facilitate plant adaptive responses to drought” is an extremely thorough investigation of the mechanisms by which soil microbial responses to long-term precipitation differences influence plant drought tolerance. The manuscript interrogates this question about plant-microbe interactions from many different perspectives. The findings are clearly presented, and the study is one of the most rigorous I have seen in a long time. I have only a few minor comments aimed at improving this already solid and exciting manuscript.

Response: Thank you for these supportive comments and for your time spent improving our manuscript.

Lines 63-65: I'm not entirely sure what is meant by "...are particularly salient at the metatranscriptomic level..."

Response: Thanks for pointing out that this wasn't clear. We have revised this sentence to read: "Our results demonstrate that legacy effects of historical exposure to dry conditions are more salient at the metatranscriptomic level than at the taxonomic or metagenomic levels, and that they trigger transcriptional changes in plant roots that improve resistance to subsequent acute droughts, at least in some plant species." (lines 49-53)

Line 190-196: These conclusions are not immediately apparent from the prior text describing the Kansas GO results or the extended figures. This likely reflects my ignorance of GO terms, but if I read correctly, only one term mentioned in the Santiago soils is the same as those listed in the Kansas results (lipid catabolic process). Additionally, the extended data figures do little to clarify because they are too small to fully understand, and also because extended figure 2g and figure 4i do not parallel one another (different colors used on heat map, unclear if axes are the same as they they are so small I cannot read them).

Response: We removed the comparison to the Santiago gradient at the request of the editor and another reviewer, so this comment is now moot. In general, however, we have re-made the Extended Data figures to be more readable.

Line 211: It would be useful to remind the reader what proportion of samples showed the same pattern across both gradients (i.e., "We validated eight of XXXX bacterial markers from Kansas and eleven of the XXXX biomarkers from Santiago...")

Response: We removed the comparison to the Santiago gradient at the request of the editor and another reviewer, so this comment is now moot.

Line 502-504: I think this conclusion is too strong. The fact that the host with the more stable microbiome was also the one that benefited from soil drought legacy does not necessarily mean that the stable microbiome is the cause of that difference between host species. I recommend revising to "and may be related to the ability of the host to maintain a stable microbiota..."). Similarly, I would also downplay the conclusion that root microbiome stability plays a role in adaptive plant responses in lines 528-529. I think this speculation takes away from the other findings that are much more thoroughly supported.

Response: This is a fair criticism, we have made this revision exactly as recommended (now line 360). The latter conclusion (now line numbers 378-381) now reads: “These benefits largely did not extend to maize, which also had a relatively unstable root microbiome and reduced physiological and morphological sensitivity to soil microbiome legacy during drought. However, further research is needed to confirm whether root microbiome stability is a mechanism of adaptive plant responses to drought.”

Figures: I realize that space is tight, but the figures are so small that they are not always legible and, therefore, are not particularly useful. In the Extended figures in the supplement, could you simply put fewer panels on a page? In the main text, consider deleting panels so that the interesting data figures are more easily understood. Additionally, I think the support for your conclusions would be much stronger and more apparent to a broad audience if the biological take-home messages were presented in the figure legend. What key point do you want readers to take from each panel?

Response: Yes, and we apologize for not catching this problem before finalizing the submission. The supplemental figures have been re-drawn for clarity and in many cases, divided into several figures to simplify each one and ensure proper sizing.

Your suggestion to convey important take-home messages in the figure legends is an excellent one that we have adopted. The first sentence of each caption is now a statement summarizing the point of the figure.

Figure 1: I would delete panel a and possibly also panel b (panel b is somewhat redundant with the subpanel showing precipitation levels across both gradients in panel g).

Response: We ended up removing panel g along with the rest of the Cape Verde-Kansas gradient comparison, so we opted to retain panel b as it was no longer redundant. After considering your suggestion we ultimately opted to also retain panel a, in part because another reviewer requested more detail on the width of the precipitation gradient, and we feel the map adds useful context on that subject. We did enhance panel a by adding more detail on the precipitation contours, so that it hopefully is now helpful enough to merit inclusion in the paper.

The figure legend suggests that panel d shows the relationship between each mineral abundance and the precipitation gradient, but the x-axis appears to be microbial taxa (I believe the references to this panel in the main text also suggest it's the association with microbial taxa).

Response: Good catch, this was an oversight in the caption. We have corrected the caption of Fig. 1d to refer to microbial taxa.

Panels e and f: It is hard to differentiate which connecting lines are black.

Response: Thanks for this feedback, we have re-drawn the figure using thicker lines to fix this problem.

Figure 2: In panels b, d, and e, could you also indicate the short-term treatment (wet or dry). Basically, I'm wondering if there's any trend that the short-term treatments are moving the communities towards their long-term legacy compositions.

Response: We thank the reviewer for this insightful comment. We looked into this question of whether the effects of the short-term watering treatment were congruent with the legacy effects, and found mixed evidence.

The below plot is a re-drawing of what is now Fig. 2c, showing the metatranscriptome content of the bacterial communities before and after the conditioning phase. The positions of the points are slightly different from the original Fig. 2c because to visualize the effect of the experimental treatment, we had to adjust the formula for the constrained ordination so that the treatment variable was not partialled out, but our original finding (the dominance of the legacy effect) still holds. This re-drawn figure shows that at least for the low-precipitation-legacy communities, the short-term drought treatment moved metatranscriptomes in the positive direction on CAP1, which is the same as the legacy effect of long-term drought exposure. However, this trend (squares to the left of triangles) is weaker or absent in the high-precipitation-legacy communities.

Ultimately, however, we decided to retain the original version of this figure, on the grounds that the paper is already complex and benefits from a singular focus on the legacy effect. We do not have room to give this interesting trend the exploration that it deserves. A thorough and detailed analysis of the microbial communities' responses to the conditioning phase treatments, including the influence of historical legacy, is in our plans for the near future. But we feel the main message of this paper is communicated most effectively by focusing only on the legacy effect. We hope the reviewer will support this decision, but if you or the editor feel it is a deal-breaker then we can revisit it.

Figure 3: I am not sure if this is possible, but could you merge panels b and c. In other words, the fact that legacy did not affect root bacterial microbiome is not very apparent from panel c (while the stats support that claim, visually the differentiation of treatments is quite similar to the effects in panel b). If you showed all 4 treatments in a single panel, the larger effect of test phase compared to legacy might be more apparent.

Response: Thank you for this suggestion. We have updated Figure 3 to show the test phase water treatment and the legacy impacts on microbiome root composition in a single plot for maize and gamagrass. Because we used a constrained ordination, rather than an unconstrained, the plot still emphasizes the variation in community composition due to both covariates (legacy and treatment), which might be visually misleading at a quick glance. However, if the reader looks at the percent variance explained by the CAP2 axis, it becomes clear why Legacy was not a statistically significant predictor of gamagrass microbiome composition.

Supplement Page 31: I must've missed it, but why were the sample sizes uneven (lower for well-watered test phases)? The text at the top of page 30 implied equal sample sizes (192 for each of the four treatments)

Response: Thank you for catching this discrepancy! Resource and space limitations prevented us from conducting a full factorial experiment as we incorrectly implied at the top of page 30. We have made the following edits to clarify:

1. "As stated in section 2.1, the conditioning phase had 24 treatment groups with eight replicates of each treatment (192 pots total). For the test phase, microbial extracts from all eight replicates of each group (plus sterile buffer-only control inoculums) were each inoculated into a pot planted with gamagrass (N=200) and a pot planted with maize (N=200) that were then maintained under watered-stress (drought) conditions. Furthermore, four of the eight replicate extracted microbial inoculants (as well as four sterile buffer-only control inoculums) were each inoculated into an additional gamagrass planted (N=100) and maize planted (N=100) pots, which were then maintained under

well watered control conditions. This makes for a total of N=600 plants at the start of the test phase. Throughout the experiment, nine water-stressed maize and five well-watered maize were lost (no gamagrass died). Therefore, phenotype measurements were completed on a total of N=586 plants. We chose this design because resource and space limitations prevented us from testing all 192 inocula under both drought and control conditions, and we were primarily interested in microbial effects on plant function under drought; we therefore opted to maximize our power to test for differences among the inocula under water limitations.” (lines 1046-1059)

2. “Thus, each microbial community extracted from one of the 192 conditioning phase pots was used to inoculate four plants in the Test Phase” now reads “Thus, each microbial community extracted from one of the 192 conditioning phase pots was used to inoculate either two or four plants in the Test Phase.” (lines 1065-1066)

Reviewer #2 (Remarks to the Author):

Ginnan et al., describes a series of experiments looking the relationship between soil moisture and the phylogenetic and functional composition of the soil microbiome and rhizosphere. The manuscript is split into two sections. The first concerns the microbial functional potential across a ~600 mm precipitation gradient in Kansas. The second provides a mechanistic look at whether the dry-adapted rhizosphere improves plant health under disturbance (soil drying).

The manuscript starts by introducing the Kansas precipitation gradient, and identifying the factors structuring functional differences within the microbiome. This is not an extensive gradient (only ~600 mm difference between the wet and dry sites), so it is interesting that the enrichment or depletion of certain traits appear robust. From here the authors attempt a rather odd 'normalization' of their data to a drier gradient in Cape Verde (range ~150 to 450 mm). I was thrown by the inclusion of these sites, and given that the rest of the manuscript concerns rhizosphere processes from the Kansas gradient I do not think it belongs in the main text.

Response: Thank you for this suggestion, to streamline the paper we have removed all reference to the Cape Verde gradient including the formal comparisons to the Kansas soils.

Next the manuscript establishes short-term drought experiments, both with and without plants. This is probably the most interesting section of the manuscript. In particular, the plant-microbe findings, while not necessarily new - we know that a dry-adapted rhizosphere mitigates drought stress - add much needed mechanistic information to our broader

knowledge. I was particularly intrigued by the restriction of benefits to the host, whereby maize appeared to not benefit at all from a dry-adapted microbiome. There is a lot to admire in this manuscript, the motivating questions are critical unknowns, and addressed through field measurements, laboratory incubations, sequencing, and statistically correlative approaches.

Response: Thanks for these supportive comments, we are glad you found it interesting.

There is a lot packed into one manuscript (as evidenced by the extensive supplemental materials). As such, the manuscript is quite complex, and I found myself getting a little lost in the narrative, which is frequently composed of lists of taxa and genes, and few contrasts to other studies. I have several specific comments that I detail below, however, I would recommend that the authors try to simplify the flow of the manuscript, concentrate more on truly novel outcomes of the plant-microbe experiment, compare and contrast the benefits afforded to gamagrass absent in maize, and try to improve the figures, which are remarkably busy.

Response: We believe that by removing the Cape Verde gradient analysis and comparisons to the Kansas gradient we have simplified the narrative considerably. This edit also reduced the size of Fig. 1 and eliminated several supplemental figures. We also did another round of general editing to simplify the language and shorten the text wherever possible.

I would also recommend the authors pay close attention to the quality of the figures in the supplemental. In several of these figures (i.e., Fig. S2, S3, etc.) axes, and labels are illegible.

Response: Yes, and we apologize for not catching this problem before finalizing the submission. The supplemental figures have been re-drawn for clarity and in many cases, divided into several figures to simplify each one and ensure proper sizing.

Specific comments:

Ln. 63: I think 'controlling for' as written here, refers to the statistical filtering of the data. Is that the case? Not that these soil properties are similar across the precipitation gradient? I can't find data on soil porosity, bulk density, conductance, etc. so can't say for sure how distinct they are.

Response: Yes, you interpreted this correctly. To clarify, we have revised this phrase to "statistically controlling for" (lines 25 and 1707)

Ln. 67: Pedantic point, but 'reveal' is a little sensationalist - maybe 'show', or 'demonstrate'?

Response: We have substituted “show”, “describe”, or “demonstrate” for “reveal” here and elsewhere in the text.

Ln. 68: This sentence is quite awkward. What do the authors mean by 'buffer'? Mitigate? Or improve fitness?

Response: Thanks for pointing out that this was unclear. We have re-written this sentence as: “This microbial precipitation legacy persisted through a 5-month-long experimental drought and mitigated the negative physiological effects of acute drought for a wild grass species that is native to the precipitation gradient, but not for the domesticated crop species maize.” (lines 26-29)

Ln. 78: A similar comment to above, this word choice is a little too dramatic. I think the authors could point out mortality induced by an increasing severity and frequency of drought.

Response: We have re-written this sentence and the previous one to read: “The increasing frequency and intensity of droughts associated with global climate change are threatening plant health and survival in both natural and agricultural ecosystems.” (lines 35-36)

Ln. 79/ 80: I think this is only remarkable from our perspective, and I would avoid terms like 'their' and 'remarkable', it's anthropomorphic.

Response: Point taken, the sentence now reads: “However, the ability of soil microbial communities to quickly adapt to environmental shifts¹ may bolster the resilience of plants and ecosystems to drought stress².” (lines 36-38)

Ln. 104: What is the width of precipitation gradient?

Response: We have added this information to the text: “... six never-irrigated remnant prairies spanning ~568 km of a steep precipitation gradient...” (line 59)

Ln. 129: Are mineral weathering rates very different across this gradient? Seems like nutrient distribution would undoubtedly follow?

Response: This is an interesting question. Precipitation introduces acids into a system, and thus plays a key role in weathering. However, the relationship is not straightforward; the weathering process is complex and varies among soil ages, rock

types, vegetation, temperatures, etc. Although we did not directly measure mineral weathering rates, other work has confirmed that precipitation patterns partly govern soil mineral content. The following sentences have been updated in the text to clarify this: “This suggests that precipitation patterns might influence the accumulation of mineral nutrients in these soils, and both precipitation and nutrients may impact microbial communities. For example, precipitation can drive mineral weathering and solute production in soils⁸, although this process also depends on many other geochemical and biological factors⁹.” (lines 79-83)

Ln. 136: To what depth?

Response: We now specify that porosity decreased with depth, down to approximately 3.5 cm and then stabilized (lines 88-89)

Ln. 142: Couldn't the legacy effect hide such a relationship?

Response: Yes, that is possible. We have revised this sentence to read: “...suggesting either that soil porosity is not a key element controlling the overall microbial community composition in these soils, or that the influence of porosity is masked by precipitation legacy.” (lines 93-95).

Ln. 159: What's the justification for the cut-off between low and high precipitation sites? HAY and KNZ have strong overlap in MAP distribution - what role does inter annual variability play? Why not low, medium, and high ppt distributions?

Response: Indeed, this was a large part of our reasoning for including only the 2 westernmost and 2 easternmost soils for the plant RNA-seq analysis later in the paper. In reality, this is a continuous gradient and any discrete categories will be somewhat artificial. We are comfortable with these groupings because they preserve the monotonic pattern of precipitation across the gradient, and are accurate geographic clusters as well.

For this particular analysis, by comparing only two groups we obtained a more straightforward picture of how metagenomic content of soils differed according to precipitation patterns. For example, it enabled us to express the abundances of metagenomic contigs as simple \log_2 -fold increases or decreases between wetter/drier environments. The addition of a third, intermediate grouping would have considerably complicated how we could present and report these patterns, which we wanted to avoid in an already-complex paper. But more importantly, the inclusion of one intermediate-precipitation site in each group was a conservative choice that

should minimize false positives and the influence of borderline/ambiguous cases. It gives us higher confidence in the results we report.

As for the role of inter-annual precipitation variability in the formation of legacy effects, it is an open question but our results from the “conditioning phase” experiment begin to address it. The fact that we still observed strong legacy effects after 5 months of high or low water exposure suggests that these legacies are built up over longer periods of time and reflect the chronic or average precipitation regime more than the most recent year’s precipitation.

Ln. 164: Maybe rephrase as 'metabolic processes related to nitrogen cycling', also, what does this mean? Fixation, Denitrification? Or deamination and amino acid synthesis?

Response: Thank you for your helpful suggestion. We have revised the sentence to read: “Biological processes that were enriched in low-precipitation soils included metabolic processes related to nitrogen cycling, fatty acid biosynthesis, DNA repair, and glucan metabolism...”

To identify the nitrogen cycling processes represented, we examined KEGG reactions enriched across the precipitation gradient. The data revealed that genes associated with amino acid synthesis and deamination were predominant in the microbial community (e.g., L-glutamine amido-ligase [AMP-forming], L-glutamine amidohydrolase, L-glutamate:NAD⁺ oxidoreductase, L-asparagine amidohydrolase, and L-ornithine ammonia-lyase). However, reactions linked to other nitrogen cycling pathways were also detected, including nitrogen fixation (e.g., L-glutamine amidohydrolase, L-glutamate:ammonia ligase [ADP-forming]) and denitrification (e.g., nitrous-oxide:ferricytochrome-c oxidoreductase). These results are available in Supplementary Table S2, “Enriched KEGG Reactions Across the Precipitation Gradient (Low vs. High) in Kansas.csv”.

Ln. 169: I think it confirms that the community has different functional potential. Translating that into function is quite difficult. Are there any data on process or decomposition rates? Can the authors tease apart the role the functional potential plays from other controls on metabolism (temperature, soil moisture, etc.)?

Response: That’s a good point, we have reworded this to refer to “functional potential”.

As for decomposition rates and other readouts of microbial metabolism, we did not measure these because we were primarily interested in how legacy effects impact

host plants, rather than ecosystem-level processes. Even if we had measured microbial metabolism, we would have little ability to disentangle the causal influences because we conducted our experiment in controlled conditions, particularly holding temperature constant for all mesocosms. However, two recent studies conducted within this same Kansas precipitation gradient both found that soil respiration is governed by soil moisture (Podzikowski, L.Y., Billings, S.A. & Bever, J.D. 2025. Plant functional diversity shapes soil respiration response to soil moisture availability. *Ecosystems* 28, 15 <https://doi.org/10.1007/s10021-024-00946-5>; Widanagamage, N., Santos, E., Rice, C.W., Patignani, A. 2025. Study of soil heterotrophic respiration as a function of soil moisture under different land covers. *Soil Biology and Biochemistry* 200, <https://doi.org/10.1016/j.soilbio.2024.109593>). Podzikowski et al. also demonstrated that functional trait diversity within the plant community influences microbial activity, while Widanagamage et al. showed that water-filled porosity, gas diffusivity, pH, organic matter content, and total microbial biomass were also correlated with respiration. Neither study included a detailed analysis of microbial gene content/functional potential, so although it is outside the scope of our paper, the interactions among all of these controls remain an interesting area for future work.

Ln. 188: I'm confused by this section. I don't know what the motivation for it is. I am satisfied that functional potential changes with MAP from the Kansas precipitation gradient. This whole section seems unnecessary.

Response: Thanks for confirming that the data from the Kansas gradient are sufficiently strong to show that a precipitation legacy exists; we have removed the comparison to the Santiago gradient to streamline the paper.

Ln. 202: Do these differences undermine the comparison?

Response: We removed the comparison to the Santiago gradient, so this comment is now moot.

Ln. 209: Strange way to write a negative result.

Response: We removed the comparison to the Santiago gradient, so this comment is now moot.

Ln. 259: Where do these reference genomes comes from? Are the isolates from the same environment? Or from databases?

Response: We now specify that the reference genomes came from the NCBI Genome database (lines 127-128)

Ln. 359: I think this is a very interesting result. Functional resilience is important, but hard to demonstrate. However, the significance gets lost in the narrative style of the manuscript. Can it be drawn out a little better?

Response: We agree that this is an exciting result. As requested we have strengthened the wording of this result to make it more apparent in the text. It now reads, "These results confirm that precipitation legacy strongly shapes both functional potential and gene expression in soil microbiota, and remains robust to perturbation (e.g., five-month-long acute drought). The functional resilience of the soil microbiota creates the potential for microbial legacy effects to influence host responses to environmental changes in natural ecosystems, conceivably enhancing plant resilience to future droughts." (lines 214-218)

Ln. 373: By experimentally conditioned, these are the communities selected for by the fluctuating wet-dry experiment without plants?

Response: That is correct. To make this more clear we updated the text to say, "...we extracted the microbial communities from the conditioning phase pots and used them to inoculate..." (lines 225-226) and also refer the reader to Extended Data Fig. 4a which provides a schematic of the experiment.

Ln. 403: I don't understand this sentence.

Response: This sentence has been updated to provide a clearer explanation:

"Indeed, the gene expression profiles in crown roots of plants inoculated with dry-legacy microbiota were distinct from those of plants inoculated with wet-legacy microbiota." (lines 253-255)

Ln. 409-412: This is all quite confusing.

Response: These lines refer to an interaction effect between our experimental drought treatment and the microbial inoculum's precipitation legacy, which both were independent variables that we manipulated in the Test Phase experiment. Specifically, we were discussing how the transcriptional response to the drought treatment was different in plants inoculated with dry-legacy vs. wet-legacy inoculum.

To improve clarity, we have modified the text to say, “This strongly suggests that soil microbiota from low-precipitation sites tend to dampen the transcriptional response of gamagrass to acute drought. For instance, 50 *T. dactyloides* genes were downregulated due to the drought treatment, but only in plants that had been inoculated with high-precipitation-legacy microbiota (Fig. 4b gene set I). These included five orthologs of maize genes predicted to be involved in ethylene- or ABA-mediated signalling of water stress (*Td00002ba004498*, *Td00002ba024351*, *Td00002ba011993*, *Td00002ba005402*, *Td00002ba000033*), and a heat shock protein linked to temperature stress (*Td00002ba042486*) (Supplementary Table S10).” (lines 260-268)

Ln. 438: How was transpiration measured here?

Response: Transpiration measurements were completed using a LI-COR LI-6800 Portable Photosynthesis System. The detailed settings used to obtain these measurements are provided in Supplementary Materials section 3.1, “Experimental design and non-destructive phenotypic measurements”.

Ln. 448: I'm not sure i understand this - are these are correlations between transpiration and rhizosphere gene expression? There are a lot of gates between the rhizosphere and stomatal control, so what does this mean?

Response: The evidence is stronger than a simple correlation analysis. A mediation analysis helps identify potential pathways by which one factor influences another, similar to path analysis and structural equation modeling. While it doesn't prove causality on its own, mediation analysis can reveal plausible mechanisms consistent with a causal model, especially when supported by biological knowledge and experimental design.

In our analysis, the independent variable was drought treatment. The dependent variables were iWUE and transpiration. It was already clear that drought treatment affected iWUE and transpiration (Fig. 4e), and because of our experimental design, we know that it was a cause-and-effect relationship.

We then wanted to test the hypothesis that the 198 plant genes that were differentially expressed in response to microbial inocula (dry-legacy vs. wet-legacy; (Fig. 4a-b) were part of the mechanism linking drought treatment to iWUE and transpiration.

We had already established that the independent variable (drought) causally altered these genes' expression (Fig. 4b), which we summarized as MDS1 and MDS2. A simple correlation between MDS1/MDS2 and iWUE/transpiration would not have

allowed us to rule out the possibility that drought altered the genes' expression and that drought altered iWUE/transpiration, but that these two processes were unrelated. **In contrast, the mediation analysis allowed us to demonstrate that these processes were not independent:** accounting for the role of MDS1 and MDS2 alters the strength of the direct effect of drought on iWUE/transpiration. In other words, drought alters iWUE through at least two pathways, one of which involves MDS1 and MDS2, the other of which does not. Thus, this set of 198 plant genes is collectively one of the “gates” between soil microbiota precipitation legacy and stomatal control.

It is true that a lot more work will be needed to fully describe the root-to-shoot axis mechanism, and unfortunately, our current datasets do not provide the resolution needed to do this. However, our analysis provides a strong foundation to guide follow-up research on this, including a promising candidate gene (nicotianamine synthase). Additional candidate mechanisms based on the current state of the field include microbial signaling molecules (ABA mimics, etc.) and microbe-dependent plant immune signalling.

Ln. 495: Signaling = signalling.

Response: Corrected.

Ln. 564: What is a 'less stable' microbiome? Large changes in microbiome function or phylogenetic composition? And on temporal or spatial scales?

Response: We explore and discuss microbiome stability in terms of consistently assembling a taxonomically similar microbiome regardless of difference in the input community, as well as consistency in community alpha diversity regardless of drought treatment. This has been clarified in multiple places in the text, for example, “...gamagrass root microbiomes are more stable—*i.e.*, they experience less change in microbiome taxonomic composition and assembly—than those of maize, in response to varying water availability and inoculation with different starting communities.” (lines 322-324). The language surrounding this result also has been softened, as suggested in our reviewer comments (“However, further research is needed to confirm whether root microbiome stability is a mechanism of adaptive plant responses to drought.”; lines 380-381)

Reviewer #2 (Remarks on code availability):

I'm unable to access the zenodo site. The link redirects to the home page, and finding datasets that have yet to be officially released is not possible.

Response: We apologize for this mistake. We have updated the permissions on the Zenodo package so that you should now be able to access our code and other files using the provided link.

Reviewer #3 (Remarks to the Author):

The article titled “Persistent legacy effects on soil microbiota facilitate plant adaptive responses to drought” seeks to demonstrate that legacy soil moisture treatments have similar effects on soil microbiota across soil types as well as on the ability of a native grass to withstand drought treatments. It is overall well written but could use some improvement before its suitable for publication with Nature Microbiology.

The biggest issue was in the supplementary (extended data) figures which were of poor resolution, and I couldn’t read/understand them. This hindered my ability to adequately review those figures and the results associated with them.

Response: Yes, and we apologize for not catching this problem before finalizing the submission. The supplemental figures have been re-drawn for clarity and in many cases, divided into several figures to simplify each one and ensure proper sizing.

Furthermore, in my opinion its important to distinguish between functional capacity represented by metagenomic data and active functions represented by metatranscriptomics data. Often the author refers to the metagenomic data as representing function, however, this its more accurate to describe this as functional capacity. Also, in discussing metagenomic data and it’s not changing during the 5 months of drought or control treatment when plants are introduced, I think its important to mention that these data are captured through DNA which is more persistent than RNA and could represent microbes that are no longer active or even relic DNA.

Response: These are very valid points, thanks for bringing this issue up. Throughout the manuscript when discussing the metagenomics data, we now refer to “functional potential” or “functional capacity” rather than simply “function”.

Additionally, to emphasize this important distinction, we added the following text: “Metagenome data often includes unexpressed genes and sequences from dormant or dead organisms, which could exaggerate the robustness of soil legacy effects. Therefore, we also quantified metatranscriptomes from the same samples to focus on biologically active processes across the treatments and soils.” (lines 195-198)

The authors used a vast array of statistical analyses, some of which I'm not as familiar with, but I am familiar with DESeq2 which requires count data. In the methods, the authors mention they use TPM as input which would be incorrect.

Response: We sincerely thank the reviewer for raising this important point. Upon re-examining our analysis scripts, we confirm that we used raw estimated counts as input for DESeq2. The reference to TPM in the Methods section (Section 2.8.4: Metatranscriptome sequence analysis) was an oversight and has now been corrected.

We have updated the Methods section of the manuscript accordingly. The original text: "Transcript quantification analysis was performed using Salmon v1.10.043 in the mapping-based mode with the de novo assembled reference metatranscriptome. Subsequently, the transcript-level abundance estimates from salmon were extracted as transcripts per million (TPM) for the identified transcripts using the R package tximport v1.28.044."

has been revised to:

"Transcript quantification analysis was performed using Salmon v1.10.0 in the mapping-based mode with the de novo assembled reference metatranscriptome. Subsequently, the transcript-level abundance estimates from salmon were extracted for the identified transcripts using the R package tximport v1.28.044 as raw counts in default setting."

Additionally, we double-checked our code for the two other analyses that involved DESeq (analysis of taxonomic composition across the precipitation gradient, and differential plant gene expression) and confirmed that both used raw counts as the input. We thank the reviewer again for helping us improve the accuracy of our methods description.

Overall, I also think that the results need to be discussed more and integrated with current literature. For example, when mentioned which genes are associated with the precipitation gradients, there is minimal discussion of why this would be or how they would be associated with higher or lower moisture. As it is, the overall traits associated with low or high moisture legacy are not clear or integrated in with the current literature.

Response: Thank you for this recommendation. We have added discussion of several bacterial traits to strengthen our argument that they are signatures of precipitation legacy.

In the section describing the differences in functional potential between low- and high-precipitation soils, we have added text and citations to clarify how the highlighted biological processes are relevant to drought tolerance in bacteria: “Biological processes enriched in low-precipitation soils included nitrogen cycling, fatty acid biosynthesis, DNA repair, and glucan metabolism, all of which have been linked to drought or stress tolerance^{13–15}. Additional processes linked to stress responses were depleted in low-precipitation soils, including ion transport (involved in osmotic adjustment), lipid catabolism (relevant for membrane integrity), and metabolism of cellular aldehydes and ketones (involved in oxidative stress)^{16,17} (Extended Data Fig. 2g).” (lines 116-121)

Additionally, in the section describing the bacterial genotype-environment association analysis, we now highlight several interesting genes and briefly explain, with citations, why they are likely to be involved in adaptation to precipitation: “Notably, some of the corresponding genes have known adaptive functions such as the phenolic acid stress response (PadR family transcriptional regulator)^{18,19}, maintenance of cellular functions under iron starvation and oxidative stress (Fe-S cluster assembly protein SufD and SufB)²⁰, and fatty acid synthesis (acetyl-CoA carboxylase biotin carboxylase subunit)²¹, which impacts membrane composition and stress tolerance¹⁷ (Extended Data Fig. 3c; Supplementary Table S3). These results indicate that precipitation legacy effects manifest through genetic differentiation within bacterial species, not just variation in community composition. ” (lines 135-142)

Some additional comments by line:

Line 80: The authors could mention how much variation is represented by PC1.

Response: Done. It now reads, “The first axis of the principal coordinate analysis, which explained 10.6% of the total variation, separated the soil bacterial communities of the two highest-precipitation sites” (line 63)

Line 81: The darkest colors of each precipitation regime are very similar to each other and hard to distinguish.

Response: We have re-drawn the figures with a different color scheme to eliminate this problem.

Line 89: Again, add percentage explained by PC1.

Response: Done: “The first principal coordinate axis of the soil mineral nutrient profiles, which explained 39.6% of the total variation, separated the three low-precipitation sites...” (line 74)

Line 92: Can you list the individual mineral nutrients that were correlated with annual precipitation since you say there are just several (plus I couldn't read the extended data at all to see which ones they were).

Response: Certainly - the sentence now reads “Concentrations of K, Mg, Ca, Li, and P were negatively correlated with mean annual precipitation, while Cd, Mn, Se, As, Zn, Co, Pb, Rb, Fe, and Cr were positively correlated” (lines 75-77)

Line 120-121: “water availability” is used twice in this sentence; maybe you could reduce to once and alter terminology.

Response: Thanks for pointing this out, we have revised this sentence to: “...these results indicate that water availability shapes soil bacterial communities, possibly by selecting for functions necessary to adapt to dry conditions and/or subsequent re-wetting¹².” (lines 103-105)

Line 136: As mentioned above, I would say this communities have different functional capacities.

Response: Agreed, we have revised the sentence to state that the communities have different functional potentials (and have done similarly throughout the paper to reinforce that metagenome data describe functional potential rather than demonstrated function).

Line 138: What are the specific functions that are specifically related to osmotic stress that were different? Or do you mean the differences listed above are due to adaptations to the different moisture levels? If so, I wouldn't state “osmotic stress” here. (Maybe this is shown in the figure, but I can't read.)

Response: This is a good point, we have modified this claim as follows: “From this, we conclude that even after controlling for multiple other soil factors, the bacterial communities from low-precipitation sites have different functional potential than those from high-precipitation sites. This result suggests that these soil bacterial communities are functionally adapted to local precipitation levels, making them excellent candidates for exploring how precipitation legacy of the soil microbiome affects drought tolerance in plants.”

Line 170: While I can't read extended data fig 4 due to the resolution, I can also tell that the font size is too small.

Response: Yes, we apologize for not catching this problem before finalizing the submission. We ended up removing this figure along with others related to the Cape Verde gradient, but we also made efforts to improve the resolution and size of all of the extended data figures.

Line 170: Again, say what these nutrients are. I think if there are just several, it's nice to state in the manuscript for the readers.

Response: We removed the comparison to the Cape Verde gradient at the request of the editor and another reviewer, so this comment is now moot.

Lines 190-195: Could you make these comparison more clear? Perhaps either listing all the similar and different ones in the text, making a table or in a easy to read figure? As it is, I'm not convinced of the extent of similarity vs. difference.

Response: We removed the comparison to the Cape Verde gradient, so this comment is now moot.

Line 224: Its not clear what you are referring to as the focal species.

Response: Thanks for pointing out this was not clear. We have revised this paragraph to better explain: "... we investigated precipitation-associated genetic variation within 33 focal bacterial species, including the previously-identified bacterial biomarkers and other highly abundant and prevalent taxa." (lines 125-127)

Line 225: I don't this this is referring to the correct figure.

Response: Thank you for pointing this out. We have removed this phantom figure callout, and double-checked that all other in-text figure callouts are accurate.

Line 237-240: Can you discuss why these gens would be associated with "drought tolerance"?

Response: We have updated the text and added several citations to support our statement. This section now says: "Notably, some of the corresponding genes have known adaptive functions such as the phenolic acid stress response (PadR family transcriptional regulator)^{18,19}, maintenance of cellular functions under iron starvation and

oxidative stress (Fe-S cluster assembly protein SufD and SufB)²⁰, and fatty acid synthesis (acetyl-CoA carboxylase biotin carboxylase subunit)²¹, which impacts membrane composition and stress tolerance¹⁷ (Extended Data Fig. 3c; Supplementary Table S3). These results indicate that precipitation legacy effects manifest through genetic differentiation within bacterial species, not just variation in community composition.” (lines 125-142)

Line 265-266: To me it was unclear what the treatment groups were, do they only include precipitation or rhizosphere vs bulk too?

Response: We agree this was vague and we have clarified this by replacing “across treatment groups” with “water availability did not affect bacterial community alpha diversity regardless of whether a plant was present” (lines 158-159).

Line 167: Does the PCA of metagenomics here refer to the taxonomy or genes? You switch back and forth above between both ways, so would be good to specify here.

Response: This entire section has been removed to streamline the paper, so the comment is moot.

Lines 278-279: Extended data fig 6i is mentioned twice in this sentence, but only need to refer to once.

Response: Thanks for catching this, we have deleted the first mention.

Line 297: Again, refer to as bacterial functional capacity.

Response: Done.

Line 300: Enrichment of GO terms in what treatment?

Response: Thank you for this helpful comment. We have clarified the text to specify the treatment conditions. The sentence has been revised to:

“GO terms related to the nitrogen cycle metabolic process, including purine-containing compound metabolic process and pyrimidine-containing compound metabolic process, were enriched in dry-legacy soils, while GO terms related to ion transport and amino acid catabolic process were depleted in the dry-legacy soils, regardless of conditioning treatment (Supplementary Fig. 4a, Supplementary Table S6). ” (lines 183-187)

Line 303: Which categories were similar? Could you discuss why you think this is?

Response: Thank you for this insightful comment. We have now included the overlapping enriched GO categories across the precipitation gradient and conditioning treatments in Supplementary Table S7, "Overlapping Enriched GO Categories Across the Precipitation Gradient and Conditioning Treatments.csv".

The results reveal that the precipitation legacy effect on the functional composition of soil metagenomes was not erased by the experimental perturbations applied during the conditioning phase. This suggests that certain bacterial functions may be resilient to acute environmental variations, highlighting the robustness of the precipitation legacy in shaping the microbial community's functional structure. Our response to your next comment discusses this issue in more detail.

Lines 314-328: These are some big claims, but its hard for me to evaluate because I can't read the figures. Also, was there a comparison made between drought and control treatments regardless of legacy? Here it only states the functions that didn't change, but what did change? Just so we get the whole picture.

Response: Yes, we apologize for not catching the problem with the figures before finalizing the submission. We have fixed the figures for the revised version, which should clarify our evidence for these claims.

We feel strongly that a full exploration of the metagenome response to the drought & control treatments could easily be its own paper, and is outside the scope of this manuscript which is already on the dense side. However, you make a fair point of needing more information to put the robustness of the legacy effect in context. To make this point clearer, we have added a new table showing the overlapping categories, Supplementary Table S7, and the following text to this discussion:

"These enrichment patterns mirrored the original field soil observations: of the 62 GO categories that were associated with precipitation legacy in the pre-conditioning soils, 49 retained the same pattern after five months of experimental drought, and 50 did so after five months of ample watering (Extended Data Fig. 2g, Supplementary Table S7)." (lines 187-191).

Line 368: I don't see a figure 2 or 2g.

Response: Thank you for catching this. We have removed references to Figs. 2f and 2g, and double-checked that all in-text figure callouts are accurate.

Methods:

Section 1.1

How long were soils stored at 4C until DNA extractions? The length of time sitting at 4C could impact communities.

Response: The answer is about 3 days. We have updated the Materials and Methods section of the manuscript to provide this detail. The original text has been revised to: "The Kansas sub-samples were shipped to the University of Nottingham for further processing, and all samples were stored at 4°C until use (~3 days)." (lines 407-409). Thank you for helping us clarify this aspect of our materials and methods.

We agree that presumably, some microbial activity continued during the storage period and the measured community composition of these soils may have been slightly different than the composition at time of collection. However, 4 degrees C is well within the annual temperature range at all of the collection sites; in fact, the average daily low air temperature in Kansas in October (when soils were collected) is around 8 degrees C, and well below that for the entire winter.

Therefore we have no reason to think that the storage period significantly altered the natural dynamics of these soils.

Do you think air-drying the soils could impact microbial communities?

Response: Presumably, the evaporation of water from the soils imparted a drought stress, which our data show does indeed have an impact on the microbial community. Perhaps this caused an increase in the relative abundance of dry-adapted organisms. However, because all of our samples were treated equally, we have no reason to think that this would compromise our results and the clear differences among the six soils that we observed. It is also not clear to us that a couple days of air-drying at room temperature is more stressful than the weeks-long droughts that these soils often encounter in late summer at 90+ degrees F. In any case, the air-dried sub-samples were only used for X-ray CT scanning, ionomics, and metagenome sequencing; presumably, any organisms that could not survive the air-drying would still be detectable via their DNA.

As far as the risk of contamination from exposure to air, in our experience the introduction of microbes from the air (e.g., into open pots in the greenhouse) usually

has negligible impact on soil communities. The very large population sizes and very diverse communities within soils seem to be quite robust to invasion via the air.

Section 1.8.7

Why do you filter to just bacterial species? What about archaea?

Response: Thank you for this insightful question. Our decision to focus exclusively on bacterial sequences was motivated by both ecological and technical considerations. First, prior studies have shown that root-associated fungi in these soils exhibit limited sensitivity to drought, and a similar pattern appears to apply to other non-bacterial taxa. Second, our metagenomic data were overwhelmingly dominated by bacterial sequences, accounting for 93.63% of the reads in soils collected across the precipitation gradient and 89.99% in the conditioning experiment (see Supplementary Fig. 5a and b). In contrast, archaeal sequences constituted only a small fraction of the community (0.97% and 1.03%, respectively). Given their low relative abundance, we focused our analyses on the dominant bacterial taxa to maximize statistical power and ensure clear interpretation of the results. We have revised the manuscript to reflect this:

Revised text: “Our focus only on bacteria was motivated by past work showing that root-colonizing fungi from these soils were insensitive to drought¹⁸, and by our observation that bacterial sequences accounted for the vast majority of metagenomic reads, 93.63% in soils collected across the precipitation gradient and 89.99% in the conditioning experiment (Supplementary Fig. 5a and b).”

DESeq2 should be used on count data, not coverage (or TPM). Can you justify why you used these other more normalized data as input? Do you get similar results when using raw counts?

Response: As described earlier in our response, we double-checked our code and can confirm that our mention of TPM was an error. We revised this text to clarify that raw counts were used as the input. Thank you again for helping us improve the accuracy of our Methods section!

Section 2.1

You introduce the grass without spelling out the genus name. Just introduce it once as genus species, then use the genus abbreviation throughout the rest of the manuscript.

Response: Thanks for catching this, we have changed this to *Tripsacum dactyloides*.

Also, in figure legends be consistent and follow journals requirements for how to mention species names because its not consistent currently.

Response: Thank you for bringing this to our attention. While we could not find any specific requirements on how to mention species names in figure legends, for consistency we modified the captions to include the full scientific name(s) of the relevant species. We also verified the other legend requirements and updated our figure legends accordingly.

Section 2.8.2

How were the sequences assembled into contigs?

Response: We have revised the text in Section 2.8.2 to clarify this:

“2.8.2 Gene Ontology (GO) term enrichment analysis: To identify enriched biological processes within the microbial communities, sequence reads from individual samples were assembled into contigs using metaFlye from the Flye v2.9 package with default parameters, as described in Methods Section 1.8.7. Relative abundance counts were then determined, and the resulting contigs were subjected to taxonomic classification and filtering, also as outlined in Methods Section 1.8.7.”

Section 2.8.4

Make sure to cite all packages/programs used (TransDecoder has no reference)

Response: Thank you for catching that we missed this citation. It has been added.

Extended data Fig. 6: There are too many panels here. Even if the resolution was okay, it would still be difficult to read. Maybe split into two?

Response: We apologize for not catching the issues with the resolution before the first submission, thank you for pointing this out. This figure has been edited to reduce the complexity and to make it easier to read.

Reviewer #1 (Remarks to the Author):

The authors have done a good job responding to my earlier comments. I continue to be enthusiastic about the rigor of the study and the breadth of approaches the authors bring together to make a convincing story. I have only a few additional comments for them to consider.

Line 69: Does this paragraph really “disentangle” the influence of precipitation from co-varying edaphic properties (additionally, the influence on what?)? It seems like this is just associating soil properties with precipitation, and the causal direction is unclear. To really disentangle the effects of precip from edaphic properties, I think you would want to do some sort of multiple regression to control for edaphic factors when testing the effects of precipitation or perhaps something like structural equation modeling given that you suggest causal directions (lines 79-83).

This is a fair criticism, we agree. In fact, we did use the multiple regression approach that you suggest, but this is not described until a few paragraphs later (beginning Line 96 in the revised version). So we have re-written this line as: “To investigate co-variation between precipitation and edaphic properties, ...”

Line 260: The manuscript alternates between using both “gamagrass” and *T. dactyloides*. I’d stick with one.

Thanks for this suggestion, we went with “gamagrass” as the default name.

Line 394: should this read: “blinded to treatment allocation”?

Yes, thank you, we have added “treatment” back into this sentence.

Line 406-407: I think there might be some inconsistencies in language here. In line 406, should “site” be “subplot”? And in line 407, should “geographical location” be “site”?

Yes indeed, we have corrected the terminology to be more consistent.

Line 808: Here and elsewhere. 225°C would be exceptionally hot for drying plant biomass. Are you sure that temperature wasn’t in Fahrenheit?

You are correct, thank you for catching that error! We have fixed all 5 occurrences.

Lines 902-904; Lines 1258 vs 1279 and elsewhere: I’m having trouble interpreting the statistical models, in part because of vague or inconsistent model terms. For example in line 903, what is the difference between “conditioning” and “Condition”? Similarly, what is “CondGroup) in lines

1433 and 1446? In line 1258, “TestPhaseWater” is used but in line 1279, “Test phase drought treatment” is used to refer to the same thing. Consistency in language would make it much easier for the reader to follow the analyses.

Yes, we see how this is confusing. This is a great suggestion. We have revised the models to consistently use only the following terms:

- Legacy
- CondWater
- CondHost
- TestWater

After these are used for the first time (lines 902-904 in the revised version) we add the clarifying text: “In these models and others described later in the text, the variables have the following meanings: *Legacy* refers to the precipitation regime of the inoculum’s original collection site (a factor with two levels, dry and wet). *CondWater* is our shorthand for “Conditioning phase watering treatment”, a factor with two levels (drought or control). *CondHost* is our shorthand for “Conditioning phase host treatment”, a factor with two levels (gamagrass or none).”

And later, in section 3.5.3 where the fourth term is used for the first time, we added: “*TestWater* is our shorthand for “Test phase watering treatment”, a factor with two levels (drought or control).”

In the two sections that refer to “CondGroup” we have added the text: “CondGroup is a factor with four levels representing the factorical combinations of conditioning phase drought and host treatments”.

Finally, you asked about the meaning of “Condition” in these models. This is an unfortunate complication arising from the syntax of the tool that we used for these analyses. Immediately following the models where Condition() appears, we have added the clarification: “Note that in the above models, “Condition()” is not a variable but part of the syntax of the CAP functions in the vegan package in R. These analyses were *partial* constrained ordinations, meaning that variation attributed to the factors within the Condition() function was removed or “partialled out” prior to the constrained ordination being conducted. In other words, the constrained ordination was performed using the residuals from the partialling-out process. This approach is useful for clearer visualizations of the patterns associated with variable(s) of interest (the factors outside of the Condition() function).”

Line 911: Here and elsewhere, shouldn’t Biological Replicate be a random factor, rather than fixed?

Thank you for catching this. In general the answer is yes. But in this case, because these were not repeated-measures analyses, Biological Replicate actually should not be in these models at all. We went through the relevant code and confirmed that our DESeq2 models did not include a separate term for Biological Replicate (in fact,

because each replicate only had one data point, including that term would have broken the model). We have therefore removed “Biological Replicate” from the statistical models in which it appeared.

Line 1063: I recommend changing to “...(plus 8 sterile buffer-only...). This would help make the total N clear.

Done, thanks for the suggestion. We also changed “control” to “mock” in this sentence to link it more clearly to the mock treatment you reference in your next comment.

Line 1420: What is the “mock” treatment?

The mock treatment consisted of sterile 1 mL PBS without any soil added. Its creation and application are described in lines 1079 to 1084 in the revised version (4th paragraph of section 3.1)

Line 1420-1421: How were these 67 features selected? Also, remind the reader what these features were (traits?).

Thanks for pointing out this was unclear. We have re-written the sentence to clarify both points: “All 67 plant traits for which we had reliable measurements were formatted into a data frame.”

Data availability: Where will the plant phenotypic data be archived?

The plant phenotype data are already archived in our Zenodo repository:
<http://doi.org/10.5281/zenodo.13821005>

Reviewer #3 (Remarks to the Author):

Ginnan et al. did an excellent job addressing all the reviewers’ comments and the manuscript reads much more smoothly now with a improved straight forward story.

I just have a couple additional comments regarding the data analyses.

1. In the methods, there are two approaches to metatranscriptomics read taxonomy classification. First, in section 2.7.3 the author mentions using kraken v. 2.1.2. Then, in section 2.8.4 they mention using CAT v.8.22. Could the authors clarify which method they use to assign taxonomy to metatranscriptomics reads?

We are grateful to the reviewer’s attention to this point. Both Kraken2 and CAT were used in our analysis, with each applied at different stages to leverage their respective strengths.

In Section 2.7.3, Kraken2 (with Bracken) was used to perform direct taxonomic classification of raw metatranscriptomic reads. We have added the following justification

to section 2.7.3: “This approach enabled a rapid and comprehensive overview of taxonomic profiles based on read-level assignments, and was primarily used for estimating taxonomic abundance and diversity directly from the sequence data.”

And we have added the following clarification to section 2.8.4: “The CAT v8.22 taxonomic classification pipeline was used to assign taxonomy to the predicted protein-coding transcripts following de novo assembly and functional annotation (as opposed to assigning taxonomy to the raw reads as done in Section 2.7.3). These taxonomic assignments were used in downstream analyses involving gene expression and functional enrichment. ”

2. (line 639) Did you use a multiple comparison correction for these p-values (or in all cases you did multiple comparisons?)

Yes. We have clarified this as follows: “A taxID, species, or family was considered statistically significant if it had a p -value < 0.05 after adjusting for multiple comparisons using the Benjamini-Hochberg method”. While double-checking this we noticed a coding error that caused us to subset the taxa using the uncorrected p-values; we fixed this issue and re-drew Extended Data Figure 2d-f using the corrected subset of taxa.

Reviewer #3 (Remarks on code availability):

I looked through the code and it was well organized and interpretable for reproducibility.